# Spinal premotor interneurons controlling antagonistic muscles are spatially intermingled

Remi Ronzano[1†], Sophie Skarlatou[2†], Bianca K Barriga[3,4†], B Anne Bannatyne[5†], Gardave Singh Bhumbra[6†], Joshua D Foster[6], Jeffrey D Moore[7], Camille Lancelin[1], Amanda M Pocratsky[1], Mustafa Görkem Özyurt[1], Calvin Chad Smith[1], Andrew J Todd[5], David J Maxwell[5], Andrew J Murray[8], Samuel L Pfaff[3*], Robert M Brownstone[1*], Niccolò Zampieri[2*], Marco Beato[6*]

[1]Department of Neuromuscular Diseases, University College London, London, United Kingdom; [2]Max Delbruck Center for Molecular Medicine, Berlin, Germany; [3]Gene Expression Laboratory, Salk Institute for Biological Studies, La Jolla, United States; [4]Biological Sciences Graduate Program, University of California, San Diego, San Diego, United States; [5]Institute of Neuroscience and Psychology, College of Medical, Veterinary and Life Sciences, University of Glasgow, Glasgow, United Kingdom; [6]Department of Neuroscience Physiology and Pharmacology, University College London, London, United Kingdom; [7]Howard Hughes Medical Institute and Department of Molecular and Cellular Biology, Center for Brain Science, Harvard University, Cambridge, United States; [8]Sainsbury Wellcome Centre for Neural Circuits and Behaviour, University College London, London, United Kingdom

*For correspondence:
pfaff@salk.edu (SLP);
r.brownstone@ucl.ac.uk (RMB);
niccolo.zampieri@mdc-berlin.
de (NZ);
m.beato@ucl.ac.uk (MB)

†These authors contributed equally to this work

**Abstract** Elaborate behaviours are produced by tightly controlled flexor-extensor motor neuron activation patterns. Motor neurons are regulated by a network of interneurons within the spinal cord, but the computational processes involved in motor control are not fully understood. The neuroanatomical arrangement of motor and premotor neurons into topographic patterns related to their controlled muscles is thought to facilitate how information is processed by spinal circuits. Rabies retrograde monosynaptic tracing has been used to label premotor interneurons innervating specific motor neuron pools, with previous studies reporting topographic mediolateral positional biases in flexor and extensor premotor interneurons. To more precisely define how premotor inter-neurons contacting specific motor pools are organized, we used multiple complementary viral-tracing approaches in mice to minimize systematic biases associated with each method. Contrary to expectations, we found that premotor interneurons contacting motor pools controlling flexion and extension of the ankle are highly intermingled rather than segregated into specific domains like motor neurons. Thus, premotor spinal neurons controlling different muscles process motor instructions in the absence of clear spatial patterns among the flexor-extensor circuit components.

## Editor's evaluation

This is a tour-de-force fundamental study of the spatial organization of flexor and extensor premotor interneurons in the mouse spinal cord by comprehensively employing most of the available premotor circuit tracing strategies involving genetically modified mouse strains and rabies virus. The important results are consistent with rigorous positional reconstructions of the premotor neuron labeling from the multiple circuit mapping approaches employed, convincingly demonstrating over-lapping spatial distributions of these premotor neurons, regardless of their putative excitatory and

inhibitory neurotransmitter identity. These new results compellingly revise our understanding of the spatial organization of spinal premotor circuits.

## Introduction

Precise regulation in the timing and pattern of activation of muscle groups across a joint is at the basis of motor control. In limbed vertebrates, the activity of flexor and extensor muscles is directed by dedicated pools of motor neurons that receive inputs from different subtypes of excitatory and inhibitory interneurons. Several of these classes of interneurons have been described in electrophysiological, anatomical, and genetic studies (*Hultborn et al., 1971*; *Jankowska, 2001*; *Goulding, 2009*); however, the incomplete knowledge of the composition of spinal circuitry that control the activity of flexor and extensor motor neurons limits progress toward a full understanding of motor circuits.

Viral trans-synaptic tracing techniques have been used for several decades to map motor circuits (*Ugolini, 2020*). Given that rabies virus (RabV) jumps across synapses in the retrograde direction to infect presynaptic neurons, its use via intramuscular injections led to the identification of polysynaptic pathways (*Rathelot and Strick, 2009*). But the identification of monosynaptic connectivity relied on the timing of transsynaptic jumps, thus leading to a degree of uncertainty about the number of synapses between a labelled neuron and the motor neurons innervating the injected muscle.

The introduction of RabV monosynaptic tracing provided a high-throughput method for mapping presynaptic connectivity of selected neuronal populations (*Callaway and Luo, 2015*). Monosynaptic restriction is achieved by using a mutant virus lacking the gene encoding the rabies glycoprotein (G; ΔG-RabV), which is necessary for transsynaptic transfer, combined with selective complementation of G expression in neurons of choice (*Wickersham et al., 2007*). Various methodologies have been used to restrict G expression to the target neuronal population, with these G-expressing neurons becoming 'starter cells' from which infecting ΔG-RabV can jump only one synapse and selectively label presynaptic neurons (*Wall et al., 2010*; *Callaway and Luo, 2015*).

Shortly after its introduction, monosynaptic rabies tracing was applied to the study of premotor interneurons in the spinal cord. To obtain selective complementation of G and subsequent rabies monosynaptic transfer from a single motor neuron pool, an elegant approach based on intramuscular co-injection of an AAV expressing G (AAV-G) and ΔG-RabV, both of which can infect motor neurons retrogradely, was described (*Stepien et al., 2010*). Thus, starter cells are generated in one fell swoop by taking advantage of the stringent anatomical specificity of motor neuron to muscle connectivity (*Figure 1A*). When this method was applied to study the distribution of premotor interneurons controlling the activity of extensor and flexor muscles in the hindlimb, a prominent spatial segregation along the medio-lateral axis of the dorsal ipsilateral spinal cord was observed, with extensor premotor interneurons found in more medial positions than flexors (*Tripodi et al., 2011*). The authors proposed that this organisation led to 'private' disynaptic pathways from proprioceptive afferents to appropriate motor neurons, and that this might offer some circuit organisational advantages.

More recently, in order to address concerns that this method could also lead to rabies infection and transsynaptic transfer from the sensory route (*Figure 1A*; *Zampieri et al., 2014*), G expression was further restricted to motor neurons by combining the use of a mouse line expressing Cre recombinase under the control of choline acetyltransferase (ChAT) and intramuscular injection of an AAV driving expression of G in a conditional manner (AAV-flex-G; *Figure 1B*). Under these conditions, segregation of flexor and extensor premotor interneurons was also shown at forelimb level (*Wang et al., 2017*). Finally, a further modification to the original AAV complementation strategy was introduced: stereotactic injection of AAV-flex-G in the spinal cord of *Chat^{Cre/+}* mice was used to target G expression to cholinergic neurons, with restriction of starter cells to a motor pool achieved by ΔG-RabV muscle injection. These experiments also showed medio-lateral segregation in the distribution of flexor and extensor premotor interneurons (*Figure 1C*; *Takeoka and Arber, 2019*). Thus, these different experiments that all used AAV for complementing G expression in motor neurons demonstrated similar segregation of extensor and flexor premotor interneurons.

Given the importance that neuronal position may play in circuit organization and function, we sought to identify premotor interneurons for further investigation. However, we elected to achieve G complementation by using a mouse genetic approach that takes advantage of a conditional mouse line that drives G expression under control of Cre recombinase (Rosa26^{RΦGT}, otherwise known as *RΦGT*

**eLife digest** The spinal cord contains circuits of nerve cells that control how the body moves. Within these networks are interneurons that project to motor neurons, which innervate different types of muscle to contract: flexors (such as the biceps), which bend, or 'flex', the body's joints, and extensors (such as the triceps), which lead to joint extension. These motor signals must be carefully coordinated to allow precise and stable control of the body's movements.

Previous studies suggest that where interneurons are placed in the spinal cord depends on whether they activate the motor neurons responsible for flexion or extension. To test if these findings were reproducible, Ronzano, Skarlatou, Barriga, Bannatyne, Bhumbra et al. studied interneurons which flex and extend the ankle joint in mice. In collaboration with several laboratories, the team used a combination of techniques to trace how interneurons and motor neurons were connected in the mouse spinal cord. This revealed that regardless of the method used or the laboratory in which the experiments were performed, the distribution of interneurons associated with flexion and extension overlapped one another.

This finding contradicts previously published results and suggests that interneurons in the spinal cord are not segregated based on their outputs. Instead, they may be positioned based on the signals they receive, similar to motor neurons.

Understanding where interneurons in the spinal cord are placed will provide new insights on how movement is controlled and how it is impacted by injuries and disease. In the future, this knowledge could benefit work on how neural circuits in the spinal cord are formed and how they can be regenerated.

mice; *Figure 1D and E*; *Takatoh et al., 2013*). This method has been previously used to trace premotor circuitry of the vibrissal and orofacial motor systems, in combination with *Chat^{Cre/+}* mice (*Takatoh et al., 2013*; *Stanek et al., 2014*), as well as forelimb muscles in combination with *Olig2^{Cre/+}* mice (*Skarlatou et al., 2020*). We reasoned that using this approach, G should be available at high levels in all motor neurons, thereby leading to efficient monosynaptic transfer from all the cells infected by ΔG-RabV, which was supplied via intramuscular injection. In contrast with previous studies, we did not observe any difference in the distribution of flexor and extensor premotor interneurons. Thus, we decided to replicate the experiments using AAV complementation strategies: AAV-G injection in wild-type mice (*Tripodi et al., 2011*) and AAV-flex-G in *Chat^{Cre/+}* mice (*Wang et al., 2017*). Surprisingly, we did not observe segregation in the spatial organization of flexor and extensor premotor interneurons. Finally, we resorted to a different viral tracing method and used timed infection with pseudorabies virus (PRV; *Strack and Loewy, 1990*; *Jovanovic et al., 2010*), which also resulted in overlapping distributions of flexor and extensor premotor neurons. Altogether, these experiments conducted in different laboratories and using most of the available methods described in the literature for viral transsynaptic tracing of premotor circuits do not show segregation of flexor and extensor premotor neuron distributions.

## Results

### Flexor and extensor premotor interneurons in Chat^{Cre/+}; Rosa26^{RΦGT} mice

In order to determine the spatial distribution of premotor interneurons controlling flexion and extension of the ankle, we injected ΔG-RabV/mCherry and ΔG-RabV/eGFP in the TA (tibialis anterior; ankle flexor) and LG (lateral gastrocnemius; ankle extensor) muscles of postnatal day (P) 1–2 *Chat^{Cre/+}*; *Rosa26^{RΦGT}* mice. Analysis of lumbar level (L) 2 and L5 sections 8–9 days after injection revealed two main clusters of premotor interneurons located in the dorsal ipsilateral and ventral contralateral spinal cord (*Figure 2A and B*). Next, we obtained coordinates for the labelled cells in each section of the lumbar spinal cord and mapped premotor interneuron positions in three dimensions. The projection of x-y coordinates along the rostro-caudal axis of the spinal cord showed no difference in medio-lateral and dorso-ventral positions of flexor and extensor premotor interneurons (*Figure 2C*, left panel). Convolved distributions fully overlapped for the two groups in all four quadrants. Similarly, projection

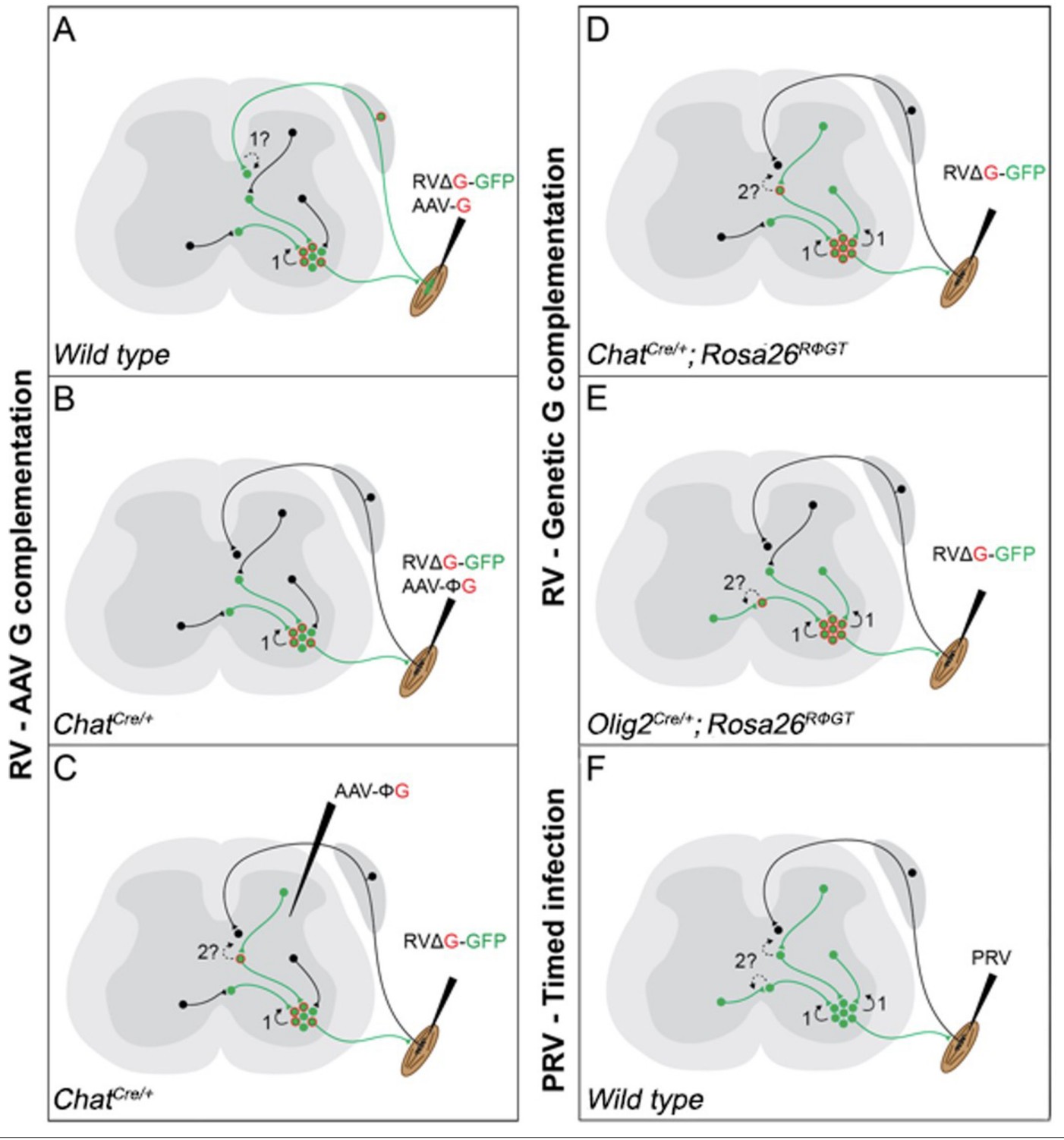

**Figure 1.** Schematic diagram of the currently available techniques for labelling premotor interneurons. (**A**) Simultaneous injection of RabV and AAV-G into muscles. Rabies transfer pathways that could potentially contaminate the distribution of premotor interneurons are labelled by a question mark and indicated by dashed lines. A: Afferent labelling could lead to anterograde labelling of sensory related interneurons. (**B**) Simultaneous muscle injection of RabV and a Cre-dependent AAV-G into mice expressing Cre in motor neurons eliminates the risk of anterograde transfer from afferents. (**C**) Intraspinal injection of a flexed AAV-G in mice expressing Cre in motor neurons is followed by intramuscular rabies injection. (**D and E**) RabV muscle injection is performed on mice selectively expressing the rabies glycoprotein in cholinergic neurons (**D**) or neuron expressing the Olig2 transcription factor (**E**). (**F**) PRV Bartha is injected in muscles and retrogradely spreads through synapses. Restriction to first order interneurons can be achieved by extracting the tissue early (~48 hr) after injection.

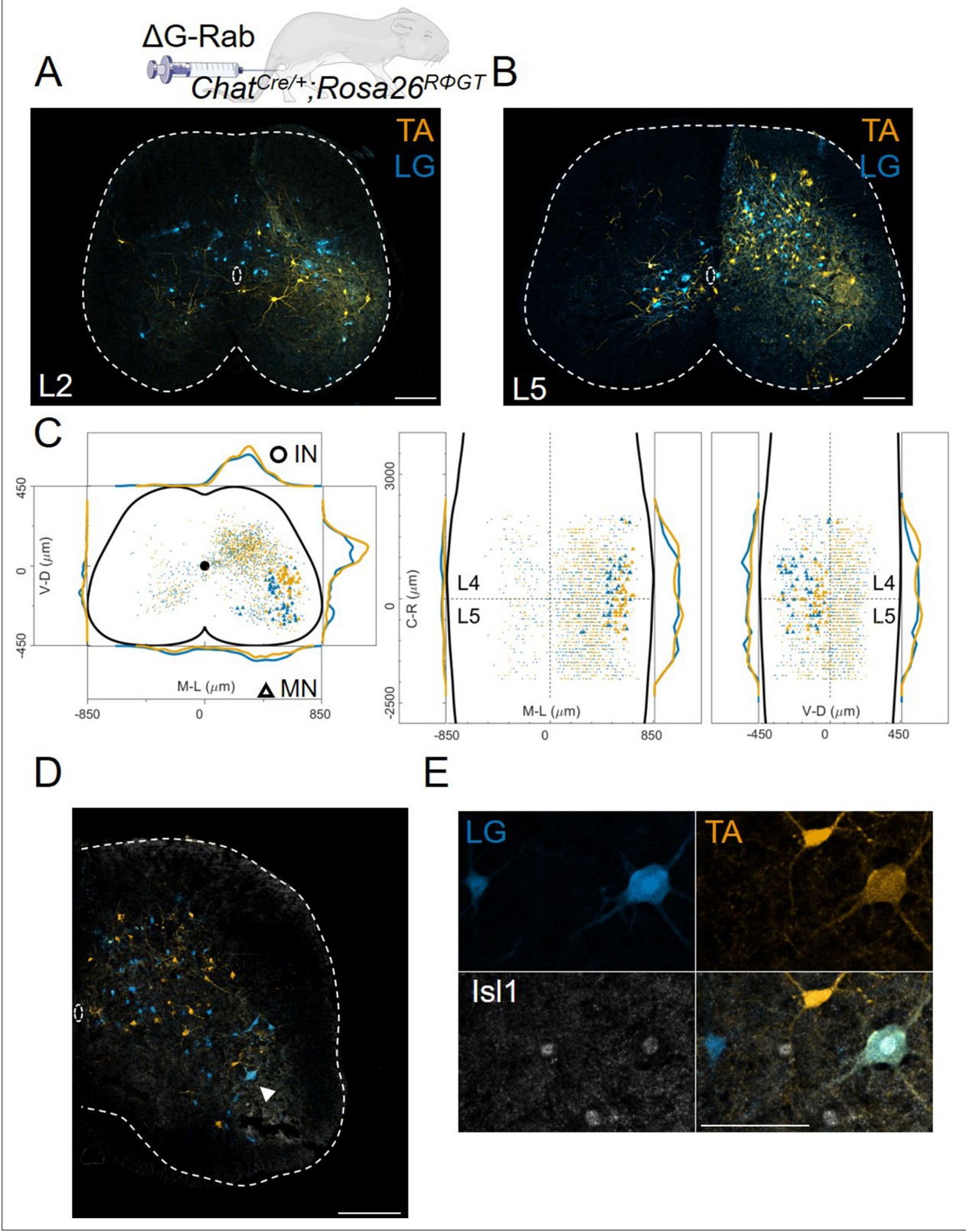

**Figure 2.** Double injections of flexor and extensor muscles shows no segregation of premotor interneurons. (**A**) Maximum intensity projection of a single 30 μm section taken from the L2 segment of a P10 cord infected with ΔG-Rab-mCherry in the LG and ΔG-Rab-EGFP in the TA in Chat^Cre/+; *Rosa26^RΦGT* mice. (**B**) Same sample as A, showing a L5 section (scale bars: 200 μm). (**C**) Projections along the transverse (left), horizontal (middle) and sagittal (right) plane throughout the lumbar region of the sample showed in A and B (170427 n2, UCL). Dots denote individual premotor interneurons, triangles denote

*Figure 2 continued on next page*

*Figure 2 continued*

infected motor neurons. Convolved density along each axe are shown to the sides of the raw data (top-bottom and left-right distributions in all panels sum to 1). For each section the data are scaled to the reference points indicated in the methods in order to account for size differences along the segments. (**D**) Half section of a cord on the side of a double injection of LG and TA in the L4 segment. Some isl1 +motor neurons are labelled in the dorsal nuclei and one (indicated by arrowhead and enlarged in (**E**)) is labelled by both fluorescent proteins, indicating a potential transsynaptic jump between antagonist motor neurons (scale bars, 250 µm and 50 µm in D and E respectively).

along the horizontal plane (i.e. along the long axis of the spinal cord in the left-right plane, *Figure 2C*, middle panel) or sagittal plane (i.e. along the long axis of the spinal cord in the anterior-posterior plane, parallel to the mid-sagittal section, *Figure 2C*, right panel) revealed no obvious differences in the rostro-caudal, dorso-ventral and medio-lateral distribution of flexor and extensor premotor interneurons.

To study in detail the positional organization of premotor interneurons controlling the activity of the ankle joint, we analysed 13 animals in which we had performed simultaneous ΔG-RabV-eGFP and ΔG-RabV-mCherry injections in three different pairs of antagonist and synergist muscles in P1-2 *Chat^Cre/+*; *Rosa26^RΦGT* mice: TA and LG, LG, and MG (medial gastrocnemius; ankle extensor and LG synergist) and TA and PL (peroneus longus; ankle flexor and TA synergist). We started by analysing the results from the antagonist pairs, LG (n=11) and TA (n=7), pooling experiments from single and double injections (*Figure 3A–B*). All LG and TA experiments are overlaid, with different shades of blue (LG) and orange (TA) representing different animals, showing the reproducibility of premotor interneurons distributions across single experiments (pooled distributions shown in *Figure 3C*, individual experiments are shown in *Figure 12—figure supplements 1–7*). The Hedges' G coefficients of the distributions in the ipsilateral dorsal quadrants for all pairs of experiments had a median of –0.06 (IQR –0.26, 0.14), showing homogeneity between experiments and no differences in the positional organization of flexor and extensor premotor interneurons. The reproducibility of the results is confirmed by analysis of the coordinates across all experiments, showing similar correlation values within or across muscles (*Figure 3D*; r≥0.78). The values of the medians of individual experiments for LG and TA injections were 329 and 315 µm respectively (Hedges' G=−0.06, *Figure 3E–F*).

Since it was previously reported that the medio-lateral segregation in the distribution of flexor and extensor premotor interneurons is more pronounced in spinal segments rostral to the infected motor nucleus (*Tripodi et al., 2011*), we analysed the organization of premotor interneurons at different lumbar levels. Positional coordinates were pooled and divided into 800 µm rostro-caudal bins and distributions were plotted for each bin from L1 to L6 (*Figure 3—figure supplement 1*). No differences in the medio-lateral distributions of LG and TA premotor interneurons were observed in any segment analysed (median positions on the medio-lateral axis for L1, the segment with the largest visible medio-lateral segregation: LG = 309 µm and TA = 327 µm, hierarchical bootstrapped Hedges'G=–0.17 (IQR –0.12, –0.22)). We further tested whether the normalization procedure might have affected the relative position of LG and TA premotor interneurons by plotting the raw coordinates split across segments (*Figure 3—figure supplement 2*) and confirmed that even in L1 there was no medio-lateral segregation (non-scaled medians were LG = 275 µm and TA = 285 µm, hierarchical bootstrapped Hedges'G=0.14 [IQR –0.20, –0.09]). Moreover, the relative density of LG and TA interneurons was similar throughout the lumbar segments (*Figure 3—figure supplement 3*).

## The identity of infected motor neurons

The identity of starter cells represents a critical element for the interpretation of rabies tracing experiments. For the rabies tracing approaches discussed here (*Figure 1*), it is difficult to determine unambiguously the number of starter motor neurons because of rabies toxicity, that kills many neurons shortly after infection (*Reardon et al., 2016*). Nevertheless, we took advantage of the topographic organization of motor neuron to muscle connectivity to evaluate the pool identity and number of infected motor neurons that survived until the end of the experiment (*Romanes, 1964*; *McHanwell and Biscoe, 1981*; *Bácskai et al., 2014* ). As predicted by the known position of the TA and LG motor pools in the spinal cord, we found that the majority of infected motor neurons were localized in the dorsal part of the ventral horn (*Figure 2C*; *Sürmeli et al., 2011*). Surprisingly, we have also found some putative motor neurons (23 out of 1174, see example in *Figure 2C*) in positions consistent with medial motor column identity and motor neurons in more ventral, 'ectopic' positions (*Figure 2C* and

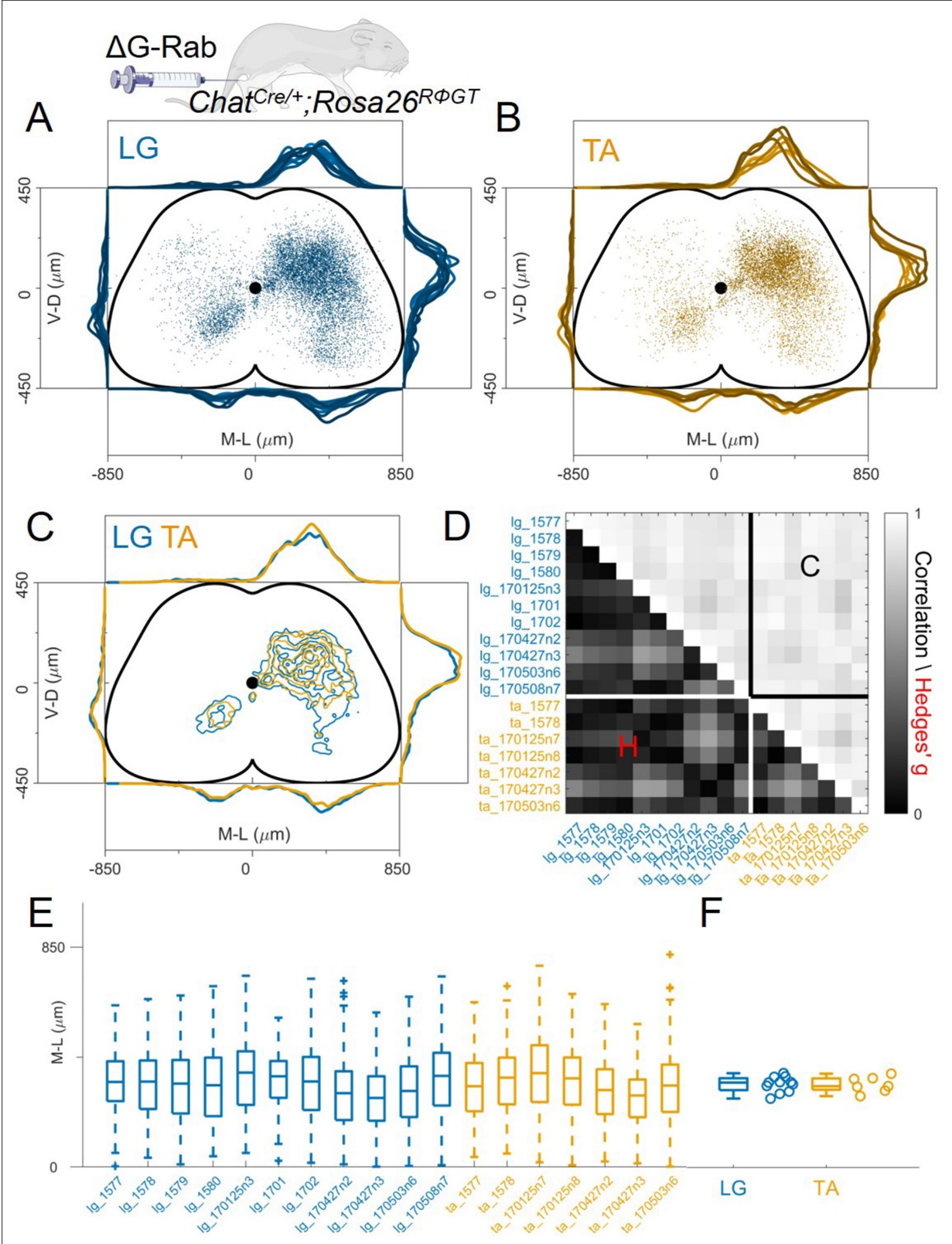

**Figure 3.** Consistent distribution of flexor and extensor premotor interneurons across all individual experiments. (**A**, **B**) Distribution of premotor interneurons of LG (**A**) and TA (**B**) for all the injections. Distributions for each individual experiment are represented with different shades of blue and orange. (**C**) All experiments (single or double ΔG-RabV injections) pooled, showing an overlap of the flexor and extensor related distribution in all quadrants of the spinal cord, with individual dots replaced by contours. For each section the data are scaled to the reference points indicated in the

*Figure 3 continued on next page*

*Figure 3 continued*

methods in order to account for size differences along the segments. (**D**) Correlation (top right) and absolute value of Hedges' G coefficient (bottom left) across all pairs of experiments, indicating a high degree of consistency and small effect sizes across all experiments, independent of the muscle injected. (**E**) Box and whisker plot of the mediolateral position of dorsal ipsilateral premotor interneurons in each experiment. (**F**) Values of the dorsal ipsilateral interneuron distribution median for each LG and TA experiment. Pooled distributions split into lumbar segments are shown in *Figure 3—figure supplement 1*, normalized and 2, raw data before normalization.

The online version of this article includes the following source data and figure supplement(s) for figure 3:

**Source data 1.** Hedges'G and correlation across experiments in the lower and upper triangular matrix respectively shown in panel D.

**Source data 2.** Median of mediolateral coordinates in the ipsilateral dorsal quadrant for each experiment shown in panel F, LG (n=11 experiments) and TA (n=7 experiments).

**Figure supplement 1.** The distribution of premotor interneurons is similar throughout the lumbar spinal cord.

**Figure supplement 1—source data 1.** Cartesian x-y-z coordinates pooled within lumbar segments L1 to L6 for LG (source data 1–6, n=11 experiments) and TA (source data 7–12, n=7 experiments).

**Figure supplement 1—source data 2.** Cartesian x-y-z coordinates pooled within lumbar segment L2 for LG.

**Figure supplement 1—source data 3.** Cartesian x-y-z coordinates pooled within lumbar segment L3 for LG.

**Figure supplement 1—source data 4.** Cartesian x-y-z coordinates pooled within lumbar segment L4 for LG.

**Figure supplement 1—source data 5.** Cartesian x-y-z coordinates pooled within lumbar segment L5 for LG.

**Figure supplement 1—source data 6.** Cartesian x-y-z coordinates pooled within lumbar segment L6 for LG.

**Figure supplement 1—source data 7.** Cartesian x-y-z coordinates pooled within lumbar segment L1 for TA.

**Figure supplement 1—source data 8.** Cartesian x-y-z coordinates pooled within lumbar segment L2 for TA.

**Figure supplement 1—source data 9.** Cartesian x-y-z coordinates pooled within lumbar segment L3 for TA.

**Figure supplement 1—source data 10.** Cartesian x-y-z coordinates pooled within lumbar segment L4 for TA.

**Figure supplement 1—source data 11.** Cartesian x-y-z coordinates pooled within lumbar segment L5 for TA.

**Figure supplement 1—source data 12.** Cartesian x-y-z coordinates pooled within lumbar segment L6 for TA.

**Figure supplement 2.** Same data as in *Figure 3—figure supplement 1* shown before normalization procedures, with idealized spinal cord section scaled to the average size of each segment.

**Figure supplement 2—source data 1.** Cartesian x-y-z raw coordinates pooled within lumbar segments L1 to L6 for LG (source data 1–6, n=11 experiments) and TA (source data 7–12, n=7 experiments).

**Figure supplement 2—source data 2.** Cartesian x-y-z raw coordinates pooled within lumbar segment L2 for LG.

**Figure supplement 2—source data 3.** Cartesian x-y-z raw coordinates pooled within lumbar segment L3 for LG.

**Figure supplement 2—source data 4.** Cartesian x-y-z raw coordinates pooled within lumbar segment L4 for LG.

**Figure supplement 2—source data 5.** Cartesian x-y-z raw coordinates pooled within lumbar segment L5 for LG.

**Figure supplement 2—source data 6.** Cartesian x-y-z raw coordinates pooled within lumbar segment L6 for LG.

**Figure supplement 2—source data 7.** Cartesian x-y-z raw coordinates pooled within lumbar segment L1 for TA.

**Figure supplement 2—source data 8.** Cartesian x-y-z raw coordinates pooled within lumbar segment L2 for TA.

**Figure supplement 2—source data 9.** Cartesian x-y-z raw coordinates pooled within lumbar segment L3 for TA.

**Figure supplement 2—source data 10.** Cartesian x-y-z raw coordinates pooled within lumbar segment L4 for TA.

**Figure supplement 2—source data 11.** Cartesian x-y-z raw coordinates pooled within lumbar segment L5 for TA.

**Figure supplement 2—source data 12.** Cartesian x-y-z raw coordinates pooled within lumbar segment L6 for TA.

**Figure supplement 3.** distribution of flexor and extensor premotor interneurons pooled across all LG and TA injections shown in the transverse plane (left) and as front (middle) and lateral (right) view along the rostrocaudal axis.

*Figure 12—figure supplements 1–7*), where pools that innervate muscles controlling the function of the knee and hip joints reside (*Sürmeli et al., 2011*). Motor neuron labelling occasionally extended outside the lower lumbar segments where most of the ankle flexors and extensor pools are located (*Figure 12—figure supplements 1–7*). Moreover, in double TA and LG injections we found instances of motor neurons infected with both viruses (*Figure 2D and E*). In 5 experiments, a total of 200 LG and 150 TA motor neurons were labelled, of which 13 were infected with both ΔG-RabV-eGFP and ΔG-RabV-mCherry (see *Table 1*).

The presence of ectopic and double labelled motor neurons could be explained by either secondary labelling due to rabies transsynaptic transfer from starter cells through recurrent connections from

**Table 1.** Details of individual experiments performed in the four different laboratories (University College London, UCL, University of Glasgow, UoG, Max Delbrück Center for Molecular Medicine, MDC, Salk Institute), with individual cell count and virus concentration.
Same experimental code as in *Figures 3D-E, 4D-E, 5B-C, F-G, 6B, E, H, Figures 8E and 9E* and *Figure 11E* and figure supplements. Experiments performed at the Salk Institute (except the PRV-Bartha experiments) were co-injections of either AAV6-B19G ($3\times10^{12}$ titre) or of AAV6 - CAG-Flex-optimizedG (oG, $1\times10^{12}$ titre). AAV and RV were injected in a 3:1 ratio. For all labs and muscles the total volume injected was 1 µl, with the exception of PRV-Bartha experiments, where 0.5 µl were injected. In the Salk Institute experiments, no attempt was made to identify primarily infected motor neurons, therefore the cell count refers to both motor neurons and interneurons.

| Code | Lab | Injection | Perfusion | Muscle | Titre I.U. | MNs | Double labelled MNs | Ipsi dorsal median (µm) | Ipsi premotor INs | Contra premotor INs | Total premotor INs | premotor INs/MNs ratio | Section sampling |
|---|---|---|---|---|---|---|---|---|---|---|---|---|---|
| 170427 n2 | UCL | P2 | P11 | LG | $1\times10^{10}$ | 46 | | 285 | 741 | 116 | 857 | 19 | |
| | | | | TA | $5\times10^{9}$ | 41 | 3 | 298 | 912 | 88 | 1000 | 24 | 1/2 (30 µm) |
| 170427 n3 | UCL | P2 | P11 | LG | $1\times10^{10}$ | 32 | | 267 | 620 | 87 | 707 | 22 | |
| | | | | TA | $5\times10^{9}$ | 6 | 2 | 276 | 386 | 34 | 420 | 70 | 1/2 (30 µm) |
| 170503 n6 | UCL | P2 | P11 | LG | $1\times10^{10}$ | 83 | | 294 | 1935 | 639 | 2574 | 31 | |
| | | | | TA | $5\times10^{9}$ | 55 | 1 | 315 | 1887 | 315 | 2202 | 40 | 1/2 (30 µm) |
| 170125 n3 | UCL | P1 | P10 | LG | $5\times10^{9}$ | 39 | | 365 | 670 | 107 | 777 | 20 | |
| | | | | MG | $5\times10^{9}$ | 39 | 0 | 353 | 819 | 307 | 1126 | 29 | 1/2 (30 µm) |
| 170508 n7 | UCL | P2 | P11 | LG | $1\times10^{10}$ | 110 | | 352 | 1955 | 382 | 2337 | 21 | |
| | | | | MG | $5\times10^{9}$ | 67 | 3 | 322 | 1497 | 429 | 1926 | 29 | 1/2 (30 µm) |
| 170125 n7 | UCL | P1 | P10 | TA | $5\times10^{9}$ | 47 | | 363 | 907 | 308 | 1215 | 26 | |
| | | | | PL | $5\times10^{9}$ | 39 | 0 | 380 | 1044 | 195 | 1239 | 32 | 1/2 (30 µm) |
| 170125 n8 | UCL | P1 | P10 | TA | $5\times10^{9}$ | 22 | | 343 | 920 | 157 | 1077 | 49 | |
| | | | | PL | $5\times10^{9}$ | 22 | 2 | 330 | 741 | 83 | 824 | 37 | 1/2 (30 µm) |
| 1570 | UoG | P1 | P10 | LG | $2\times10^{8}$ | 11 | - | 322 | 1111 | 404 | 1515 | 138 | 1/8 (60 µm) |
| 1571 | UoG | P1 | P10 | LG | $2\times10^{8}$ | 12 | - | 340 | 760 | 196 | 956 | 80 | 1/8 (60 µm) |
| 1573 | UoG | P1 | P10 | TA | $5\times10^{8}$ | 10 | - | 332 | 447 | 68 | 515 | 52 | 1/8 (60 µm) |

*Table 1 continued on next page*

*Table 1 continued*

| Code | Lab | Injection | Perfusion | Muscle | Titre I.U. | MNs | Double labelled MNs | Ipsi dorsal median (µm) | Ipsi premotor INs | Contra premotor INs | Total premotor INs | premotor INs/MNs ratio | Section sampling |
|---|---|---|---|---|---|---|---|---|---|---|---|---|---|
| 1574 | UoG | P1 | P10 | TA | $5\times10^8$ | 14 | - | 365 | 297 | 26 | 323 | 23 | 1/8 (60 µm) |
|  | UoG |  |  | LG | $2\times10^9$ | 18 |  | 329 | 313 | 43 | 356 | 20 |  |
| 1577 |  | P2 | P10 | TA | $5\times10^9$ | 26 | 2 | 312 | 688 | 105 | 793 | 31 | 1/8 (60 µm) |
|  | UoG |  |  | LG | $2\times10^9$ | 21 |  | 330 | 292 | 34 | 326 | 16 |  |
| 1578 |  | P2 | P10 | TA | $5\times10^9$ | 22 | 5 | 346 | 790 | 130 | 920 | 42 | 1/8 (60 µm) |
|  | UoG |  |  | LG | $2\times10^9$ | 30 |  | 322 | 1023 | 194 | 1217 | 41 |  |
| 1579 |  | P2 | P10 | MG | $5\times10^8$ | 7 | 1 | 306 | 169 | 19 | 188 | 27 | 1/8 (60 µm) |
|  | UoG |  |  | LG | $2\times10^9$ | 14 |  | 316 | 414 | 48 | 462 | 33 |  |
| 1580 |  | P2 | P10 | MG | $5\times10^8$ | 8 | 0 | 348 | 470 | 87 | 557 | 70 | 1/8 (60 µm) |
| 1605 | UoG | P1 | P10 | MG | $1\times10^8$ | 6 | - | 340 | 412 | 110 | 522 | 87 | 1/8 (60 µm) |
| 1611 | UoG | P1 | P10 | PL | $1\times10^8$ | 2 | - | 328 | 167 | 24 | 191 | 96 | 1/8 (60 µm) |
| 1613 | UoG | P2 | P10 | PL | $1\times10^8$ | 1 | - | 340 | 164 | 16 | 180 | 180 | 1/8 (60 µm) |
| 1639 | UoG | P2 | P10 | TA | $2\times10^8$ | 15 | - | 341 | 591 | 94 | 685 | 46 | 1/8 (60 µm) |
| 1640 | UoG | P2 | P10 | PL | $2\times10^8$ | 20 | - | 322 | 629 | 122 | 751 | 38 | 1/8 (60 µm) |
|  |  |  |  | LG | $1\times10^8$ | 1 | - | 344 | 142 | 32 | 174 | 174 |  |
| 1644 | UoG | P2 | P10 | TA | $2\times10^8$ | - | - | 296 | 57 | 11 | 68 | - | 1/8 (60 µm) |
|  |  |  |  | LG | $1\times10^8$ | 1 |  | 261 | 90 | 16 | 106 | 106 |  |
| 1646 | UoG | P2 | P10 | TA | $2\times10^8$ | 3 | - | 305 | 76 | 13 | 89 | 30 | 1/8 (60 µm) |
|  |  |  |  | LG | $1\times10^8$ | 2 |  | 307 | 60 | 6 | 66 | 33 |  |
| 1653 | UoG | P2 | P10 | TA | $2\times10^8$ | 2 | - | 312 | 58 | 8 | 66 | 33 | 1/8 (60 µm) |
| 1656 | UoG | P2 | P10 | LG | $1\times10^8$ | - | - | 311 | 563 | 145 | 708 | - | 1/8 (60 µm) |

Table 1 continued

| Code | Lab | Injection | Perfusion | Muscle | Titre I.U. | MNs | Double labelled MNs | Ipsi dorsal median (µm) | Ipsi premotor INs | Contra premotor INs | Total premotor INs | premotor INs/MNs ratio | Section sampling |
|---|---|---|---|---|---|---|---|---|---|---|---|---|---|
| 1657 | UoG | P2 | P10 | LG | $1\times10^8$ | 1 | - | 321 | 323 | 51 | 374 | 374 | 1/8 (60 µm) |
| 1660 | UoG | P2 | P10 | MG | $2\times10^8$ | 7 | - | 324 | 509 | 3 | 512 | 73 | 1/8 (60 µm) |
| 1661 | UoG | P2 | P10 | MG | $2\times10^8$ | 10 | - | 338 | 175 | 63 | 238 | 24 | 1/8 (60 µm) |
| 1662 | UoG | P2 | P10 | MG | $2\times10^8$ | 10 | - | 313 | 375 | 230 | 605 | 61 | 1/8 (60 µm) |
|  |  |  |  | LG | $2\times10^9$ | 8 |  | 351 | 169 | 26 | 195 | 24 |  |
| 1701 | UoG | P2 | P10 | MG | $5\times10^9$ | 34 | 2 | 329 | 594 | 190 | 784 | 23 | 1/8 (60 µm) |
|  |  |  |  | LG | $2\times10^9$ | 14 |  | 331 | 561 | 107 | 668 | 48 |  |
| 1702 | UoG | P2 | P10 | MG | $5\times10^9$ | 2 | 2 | 322 | 76 | 11 | 87 | 44 | 1/8 (60 µm) |
| 353 | MDC | P4 | P10 | GS | $1\times10^9$ | 31 | - | 283 | 1542 | 431 | 1973 | 64 | All (40 µm) |
| 399 | MDC | P4 | P10 | GS | $1\times10^9$ | 41 | - | 286 | 569 | 77 | 646 | 16 | All (40 µm) |
| 1332 | MDC | P4 | P10 | GS | $1\times10^9$ | 18 | - | 317 | 1605 | 323 | 1928 | 107 | All (40 µm) |
| 1349 | MDC | P4 | P10 | GS | $1\times10^9$ | 18 | - | 305 | 1416 | 459 | 1875 | 104 | All (40 µm) |
| 700 | MDC | P4 | P10 | TA | $1\times10^9$ | 47 | - | 318 | 1723 | 122 | 1845 | 39 | All (40 µm) |
| 721 | MDC | P4 | P10 | TA | $1\times10^9$ | 22 | - | 310 | 1934 | 465 | 2399 | 109 | All (40 µm) |
| 1324 | MDC | P4 | P10 | TA | $1\times10^9$ | 17 | - | 292 | 2041 | 301 | 2342 | 138 | All (40 µm) |
|  |  |  |  | GS | $1\times10^{11}$ | N/A | N/A | 328 | 9185 | 2735 | 11920 | N/A | All (60 µm) |
| 1 | Salk | P2 | P10 | TA | $1\times10^{11}$ | N/A | N/A | 349 | 3330 | 731 | 4061 | N/A | All (60 µm) |
|  |  |  |  | GS | $1\times10^{11}$ | N/A | N/A | 303 | 8827 | 3867 | 12694 | N/A |  |
| 2 | Salk | P2 | P10 | TA | $1\times10^{11}$ | N/A | N/A | 294 | 3198 | 1132 | 4330 | N/A | All (60 µm) |
| a | Salk | P1 | P8 | GS | $1\times10^{10}$ | N/A | N/A | 248 | 334 | 42 | 376 | N/A | 1/9 (30 µm) |
| b | Salk | P1 | P8 | GS | $1\times10^{10}$ | N/A | N/A | 237 | 275 | 30 | 305 | N/A | 1/9 (30 µm) |

*Table 1 continued*

| Code | Lab | Injection | Perfusion | Muscle | Titre I.U. | MNs | Double labelled MNs | Ipsi dorsal median (µm) | Ipsi premotor INs | Contra premotor INs | Total premotor INs | premotor INs/MNs ratio | Section sampling |
|---|---|---|---|---|---|---|---|---|---|---|---|---|---|
| 22 a_4 | Salk | P2 | P9 | GS | $3\times10^{11}$ | N/A | N/A | 403 | 464 | 58 | 522 | N/A | All (60 µm) |
| 26 a_1 | Salk | P2 | P9 | GS | $3\times10^{11}$ | N/A | N/A | 383 | 941 | 91 | 1032 | N/A | All (60 µm) |
| 26 a_2 | Salk | P2 | P9 | GS | $3\times10^{11}$ | N/A | N/A | 351 | 1910 | 401 | 2311 | N/A | All (60 µm) |
| 26 a_4 | Salk | P2 | P9 | GS | $3\times10^{11}$ | N/A | N/A | 382 | 1923 | 392 | 2315 | N/A | All (60 µm) |
| 26_1 | Salk | P2 | P9 | TA | $3\times10^{11}$ | N/A | N/A | 348 | 3236 | 263 | 3499 | N/A | All (60 µm) |
| 26_3 | Salk | P2 | P9 | TA | $3\times10^{11}$ | N/A | N/A | 367 | 2078 | 465 | 2543 | N/A | All (60 µm) |
| 26_4 | Salk | P2 | P9 | TA | $3\times10^{11}$ | N/A | N/A | 350 | 2494 | 597 | 3091 | N/A | All (60 µm) |
| 1_1 PRV | Salk | P11 | P13 | GS | $1\times10^{9}$ | N/A | N/A | 318 | 430 | 54 | 484 | N/A | 1/4 (60 µm) |
|  |  |  |  | TA | $1\times10^{9}$ | N/A | N/A |  |  |  |  | N/A |  |
| 1_4 PRV | Salk | P11 | P13 | GS | $1\times10^{9}$ | N/A | N/A | 349 | 238 | 23 | 261 | N/A | 1/4 (60 µm) |
|  |  |  |  | TA | $1\times10^{9}$ | N/A | N/A |  |  |  |  | N/A |  |
| 2_2 PRV | Salk | P11 | P13 | GS | $1\times10^{9}$ | N/A | N/A | 357 | 515 | 82 | 597 | N/A | 1/4 (60 µm) |
|  |  |  |  | TA | $1\times10^{9}$ | N/A | N/A |  |  |  |  | N/A |  |
| 3_3 PRV | Salk | P11 | P13 | GS | $1\times10^{9}$ | N/A | N/A | 377 | 1005 | 53 | 1058 | N/A | 1/4 (60 µm) |
|  |  |  |  | TA | $1\times10^{9}$ | N/A | N/A |  |  |  |  | N/A |  |

these other motor neurons or by unintended primary infection of motor neurons due to non-specific muscle injections. We cannot say whether cholinergic cells in the medial motor column are indeed motor neurons. They could be either large cholinergic interneurons that are presynaptic to motor neurons, or (perhaps more likely) medial motor neurons that send recurrent axon collaterals to lateral motor neurons. In fact, while collaterals from lateral or medial motor neurons have not been traced outside their respective columns, the dendritic arborization from medial and lateral motor neuron columns extends to each other and labelling of lateral motor neurons following rabies injection of medial column innervating muscles has been observed (*Goetz et al., 2015*; *Balaskas et al., 2019*; *Feng et al., 2022*), indicating that some connectivity, at least from lateral to medial columns, is possible. With respect to 'ectopic' labelling in other LMC pools, it is important to consider leak of the virus at the injection sites. With our small volume (1 µl) injections, leak between muscles located on opposite side of the fibula (GS and TA) is extremely unlikely. However, the lower part of the biceps femoris muscle has some overlap with the gastrocnemius, potentially leading to leak of virus into the biceps compartment (*Sürmeli et al., 2011*). Careful post-hoc analysis of hindlimb muscles after ∆G-RabV injection did not reveal any evidence of non-specific muscle infection in the upper and lower aspects of the leg, but we cannot exclude contamination of non-targeted muscles that gave rise to expression of reporter protein that was below the threshold for detection. If this were the case, presumably a very small number of motor neurons belonging to a non-targeted muscle could have been infected. We therefore suggest that the majority of ectopic motor neurons were labelled transsynaptically and therefore represent second-order presynaptic neurons. This is not surprising, as motor neurons have been shown to form synapses with other motor neurons and their connections can extend to neighbouring spinal segments (*Bhumbra and Beato, 2018*), suggesting that the ectopic motor neurons found in our experiments were most likely due to rabies transsynaptic transfer. Regardless of the underlying reasons for the observed ectopic motor neuron labelling, its presence raises the possibility that what we defined as flexor and extensor premotor networks, might, in fact, originate from a mixed population of starter cells containing not only motor neurons of a single pool identity but also a fraction of, 'non-specific', motor neurons belonging to other pools, thereby potentially diluting any observable spatial difference between the premotor networks of flexor and extensor muscles. However, it is important to notice that in our experiments the number of presumed 'non-specific' starter cells is low and therefore unlikely to confound the results (*Table 1*; see below and Discussion).

## The number of infected motor neurons does not affect the distribution of premotor interneurons

Spinal or muscle injection of AAV to complement G expression is likely to result in infection of a subset of motor neurons within the targeted pool, whereas the genetic experiments will result in complementation in all motor neurons. Thus, it is possible that the difference in the results obtained with these two methods could be explained by the absolute number of motor neurons from which rabies synaptic transfer occurs. In order to test the effect of the number of starter cells in our experimental conditions, we reasoned that by reducing the viral titre of the rabies solution used for muscle injection, we would scale down the number of infected motor neurons. Therefore, we performed a series of muscle injections (7 LG and 6 TA, of which 3 double LG-TA) in *Chat$^{Cre/+}$*; *Rosa26$^{RΦGT}$* mice (*Figure 1D*) as in the experiments described above, but with diluted rabies virus (titre <$10^9$ I.U./ml) to reduce its infection efficiency (see *Table 1*). In low titre experiments, we detected an average 4.7 infected motor neurons compared to an average of 35.2 in the high titre experiments (titre >$5 × 10^9$ I.U./ml). Once again, we did not observe segregation in the medio-lateral distribution of LG and TA premotor interneurons (*Figure 4A–C*), nor along the rostro-caudal axis (*Figure 4—figure supplement 1*). The pairwise Hedges' G coefficients for all the experiments had median value of –0.05 (IQR –0.19, 0.15, *Figure 4D*). While there was a higher degree of variability between experiments compared to high titre injections, as shown in the correlation matrix of individual experiments (*Figure 4D*, *r*>0.45 for all comparisons), the median value of the medio-lateral positions in each experiment were very similar (LG = 321 µm and TA = 322 µm, hierarchical bootstrapped Hedges' G=−0.02, IQR –0.09, 0.03, *Figure 4E and F*, individual experiments shown in *Figure 12—figure supplements 8–10*). Next, we compared high and low titre experiments for each muscle injected (*Figure 4—figure supplement 2*). The distribution of premotor interneurons

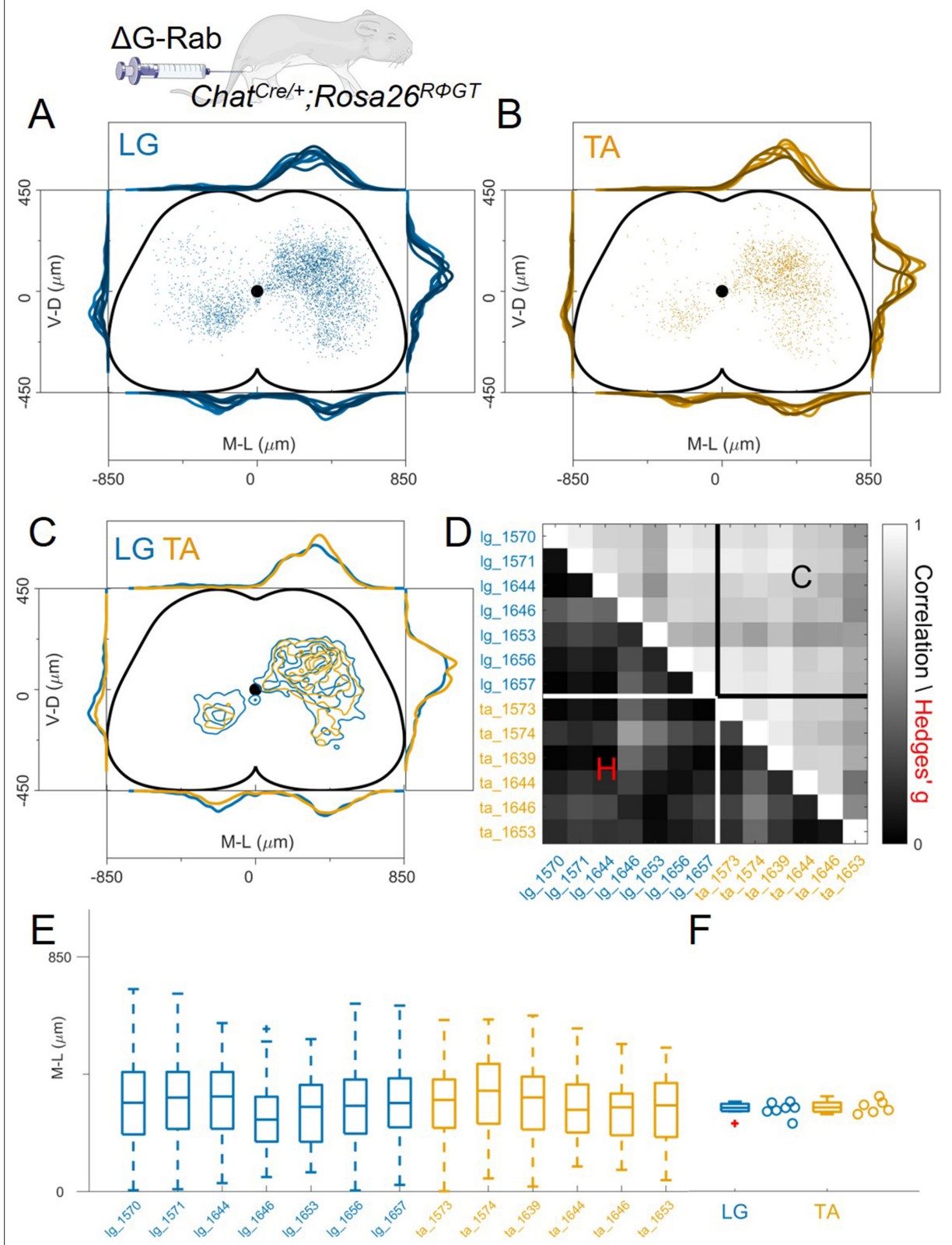

**Figure 4.** Injection with diluted RabV do not reveal any segregation between flexor and extensor premotor interneurons. (**A** and **B**) show the distribution of LG and TA premotor interneurons on the transverse plane for individual experiments, represented with different colour shades. For each section the data are scaled to the reference points indicated in the methods in order to account for size differences along the segments. (**C**) LG and TA distributions overlap and the premotor interneuron distributions are highly correlated across experiments and absolute value of Hedges' G coefficients are small (**D**).

*Figure 4 continued on next page*

*Figure 4 continued*

(**E**) Box and whisker plot of the mediolateral position of dorsal ipsilateral premotor interneurons in each experiment. (**F**) Values of the dorsal ipsilateral median for each LG and TA experiment.

The online version of this article includes the following source data and figure supplement(s) for figure 4:

**Source data 1.** Hedges'G and correlation across experiments in the lower and upper triangular matrix respectively shown in panel D.

**Source data 2.** Median of mediolateral coordinates in the ipsilateral dorsal quadrant for each experiment shown in panel F, LG (n=7 experiments) and TA (n=6 experiments).

**Figure supplement 1.** distribution of flexor and extensor premotor interneurons pooled across all LG and TA for low titre injections shown in the transverse plane (left) and as front (middle) and lateral (right) view along the rostrocaudal axis.

**Figure supplement 2.** High and low efficiency infections give rise to the same premotor interneurons distributions.

**Figure supplement 2—source data 1.** Hedges'G and correlation across experiments in the lower and upper triangular matrix respectively for LG high and low titre injections shown in panel B.

**Figure supplement 2—source data 2.** Median of mediolateral coordinates in the ipsilateral dorsal quadrant for each experiment shown in panel D, high titre LG (n=11 experiments) and low titre LG (n=7 experiments).

**Figure supplement 2—source data 3.** Hedges'G and correlation across experiments in the lower and upper triangular matrix respectively for TA high and low titre injections shown in panel F.

**Figure supplement 2—source data 4.** Median of mediolateral coordinates in the ipsilateral dorsal quadrant for each experiment shown in panel H, high titre TA (n=7 experiments) and low titre TA (n=6 experiments).

**Figure supplement 3.** the relation between the number of primary infected motor neurons and premotor interneurons follow a power law $y=ax^b$ with a=295 (155, 561 confidence intervals) and b=0.53 (0.37, 0.69 confidence intervals), $R^2$=0.48.

**Figure supplement 3—source data 1.** Number of labelled motor neurons and interneurons taken from *Table 1* and scaled according to the sampling intervals of the sections.

shows remarkable overlap for LG and TA injections (*Figure 4—figure supplement 2A, E*). The medio-lateral distributions were not different for LG (median Hedges' G=–0.05, IQR –0.21, 0.11, with high correlation values across experiments, between 0.77 and 0.98, *Figure 4—figure supplement 2B*, and medians of 321 µm and 329 µm for high and low titre experiments respectively, *Figure 4—figure supplement 2C–D*). When comparing the high and low titre injections of TA, we found high correlation values across experiments (across experiment median 0.83, IQR 0.74, 0.90) and low Hedges' G (median –0.10, IQR –0.26, 0.13, *Figure 4—figure supplement 2F*) and no differences in the lateral direction for low titre injections into TA (medians: high = 315 µm and low = 322 µm, *Figure 4—figure supplement 2G–H*). Of note, in low titre experiments we never observed ectopic motor neurons outside the expected nucleus (individual experiments are shown in *Figure 12—figure supplements 8–10*).

Despite the almost 10-fold difference in the estimated number of primary infected motor neurons between high and low titre experiments, the spatial distribution of premotor interneurons was not altered, but their absolute number was. This high variability in the number of primary infected neurons has been observed in the combined analysis of large datasets of rabies tracing experiments in the brain where the relation between primary and secondary infected cells is described by a power law (*Tran-Van-Minh et al., 2022*). To verify that our dataset followed the same statistical rules, the number of cells was scaled across different experiments to reflect the different sampling of the sections (see *Table 1*) and the number of interneurons vs. number of putative starter motor neurons was plotted (*Figure 4—figure supplement 3*). The data points are well fitted by a power law ($R^2$=0.48), in agreement with published brain datasets (*Tran-Van-Minh et al., 2022*). The ratio between interneurons and motor neurons numbers had medians of 51 (IQR 33–137) and 37 (IQR 25–67) for high and low titre injections respectively (Hedges'G=0.84). While high variability in the initial number of starter cells seems inherent to rabies tracing, our dataset reflects a similar dependency between the number of primary and secondary infected neurons as that observed in other published datasets obtained in different parts of the central nervous system from different laboratories. Together, these data indicate that neither the absolute number of starter motor neurons nor the infection of ectopic motor neurons observed in high titre experiments significantly affects the positional organization of premotor interneurons.

## The distribution of premotor interneurons is similar across different pairs of ankle flexors or extensors

Since it has been proposed that medio-lateral segregation of premotor interneurons is a general feature of flexor and extensor muscles, we analysed premotor interneurons of two more muscles controlling the movement of the ankle joint, PL and MG using the same viral strategy employed for the LG-TA injections (*Figure 1D*). The distributions of premotor interneurons of LG (6 LG-MG injections, 8 LG-TA injections and 4 single LG injections) and MG (6 LG-MG injections and 4 MG injections) did not reveal any difference in spatial organization (*Figure 5A*, individual experiments are shown in *Figure 12—figure supplements 3 and 4* and *Figure 12—figure supplement 6*). The Hedges' G for the mediolateral positions in the ipsilateral dorsal quadrants were computed for each pair of experiment and had a median value of 0.01 (IQR –0.12, 0.16). Throughout the cord, the positions of rabies-labelled neurons were highly correlated (*Figure 5B*, r≥0.74) and reproducible along the medio-lateral axis (*Figure 5C*), with median values of the medio-lateral position across experiments of 322 µm for LG and 327 µm for MG, (bootstrapped median Hedges' G=–0.06, IQR −0.03,–0.10, *Figure 5D*). The same result was observed for TA and PL premotor interneurons (*Figure 5E*; 2 TA-PL injections, 8 TA-LG injections, 3 TA single injections and 3 PL single injections, individual experiments are shown in *Figure 12—figure supplements 5 and 7*). The median of the Hedges' G coefficients for all the pairs of experiments was-0.01 (IQR –0.20, 0.18), with high correlation values between experiments (*Figure 5F*; r≥0.66). The medians of the mediolateral pooled distributions were 315 µm for TA and 330 µm for PL, similar medio-lateral distributions (*Figure 5G*) and median values were observed (*Figure 5H*). Hierarchical bootstrap of the data from TA and PL injections resulted in a median Hedges' G of –0.09 (IQR −0.15,–0.03). Together, these data show that premotor interneuron maps obtained using ΔG-RabV muscle injection in $Chat^{Cre/+}$; $Rosa26^{RΦGT}$ mice (*Figure 1D*) do not reveal any difference in the positional organization of interneurons controlling the activity of any of the main flexor and extensor muscles of the ankle.

## Spatial distribution of GlyT2$^{off}$ and GlyT2$^{on}$ premotor interneurons

Next, we examined whether there are differences in the spatial organization of GlyT2$^{off}$ vs GlyT2$^{on}$ premotor interneurons, where the GlyT2$^{off}$ population will largely comprise excitatory neurons, as well as some purely GABAergic interneurons. We performed single LG or TA injections of ΔG-RabV/mCherry in $Chat^{Cre/+}$; $Rosa26^{RΦGT}$ mice carrying an allele expressing GFP under the control of the neuronal glycine transporter (Slc6a5 or GlyT2; *Zeilhofer et al., 2005*).

We monitored the GlyT2 status of cells while performing transsynaptic labelling from LG and TA motor neurons. First, we compared the distribution of premotor GlyT2$^{off}$ interneurons (*Figure 6A–C*) and then examined the distribution of GlyT2$^{on}$ (*Figure 6D–F*). The distribution of GlyT2$^{off}$ premotor interneurons was the same for LG and TA motor neurons (4 LG and 3 TA single injections, *Figure 6A*). The medians of the medio-lateral position in the dorsal ipsilateral cord were 306 µm and 326 µm for LG and TA, respectively (hierarchical bootstrapped Hedges'G=–0.03, IQR –0.08, 0.01) and –112 µm and 74 µm (hierarchical bootstrapped Hedges'G=–0.14, IQR −0.24,–0.04) in the ventral spinal cord (*Figure 6B–C*). Similarly, we did not observe segregation in the distribution of GlyT2$^{on}$ LG and TA premotor interneurons (*Figure 6D*). The medians of the medio-lateral coordinates of the dorsal GlyT2$^{on}$ interneurons were 303 µm for LG and 346 µm for TA (hierarchical bootstrapped Hedges'G=–0.30, IQR −0.35,–0.24), while for ventral interneurons were 395 µm for LG and 437 µm for TA (hierarchical bootstrapped Hedges'G=–0.33, IQR –0.40, 0.27, *Figure 6E–F*). High correlation values (r>0.79). between all individual experiments underscored the conserved positional organization of LG and TA premotor interneurons. These data indicate that there is no significant difference in the distribution of GlyT2$^{off}$ and GlyT2$^{on}$ premotor interneurons controlling the activity of flexor and extensor muscles.

Finally, we compared the distributions of GlyT2$^{off}$ and GlyT2$^{on}$ premotor interneurons separately for each muscle, LG (*Figure 6G–I*) and TA (*Figure 6J–L*). No differences were observed for the medio-lateral distribution of inhibitory and excitatory dorsal premotor interneurons (medians for LG: GlyT2$^{on}$ = 303 µm and GlyT2$^{off}$ = 306 µm; hierarchical bootstrapped Hedges'G=–0.06, IQR 0.03, 0.09. Medians for TA: GlyT2$^{on}$ = 346 and GlyT2$^{off}$ = 326 µm; hierarchical bootstrapped Hedges'G=0.15, IQR 0.08, 0.23). In contrast, ventral ipsilateral GlyT2$^{on}$ were more abundant than GlyT2$^{off}$ for both LG (*Figure 6H–I*) and TA (*Figure 6K–L*). Conversely, GlyT2$^{off}$ premotor interneurons dominated the ventral contralateral side (*Figure 6H and K*; medians for LG: GlyT2$^{off}$ = –112 µm and GlyT2$^{on}$ = 395

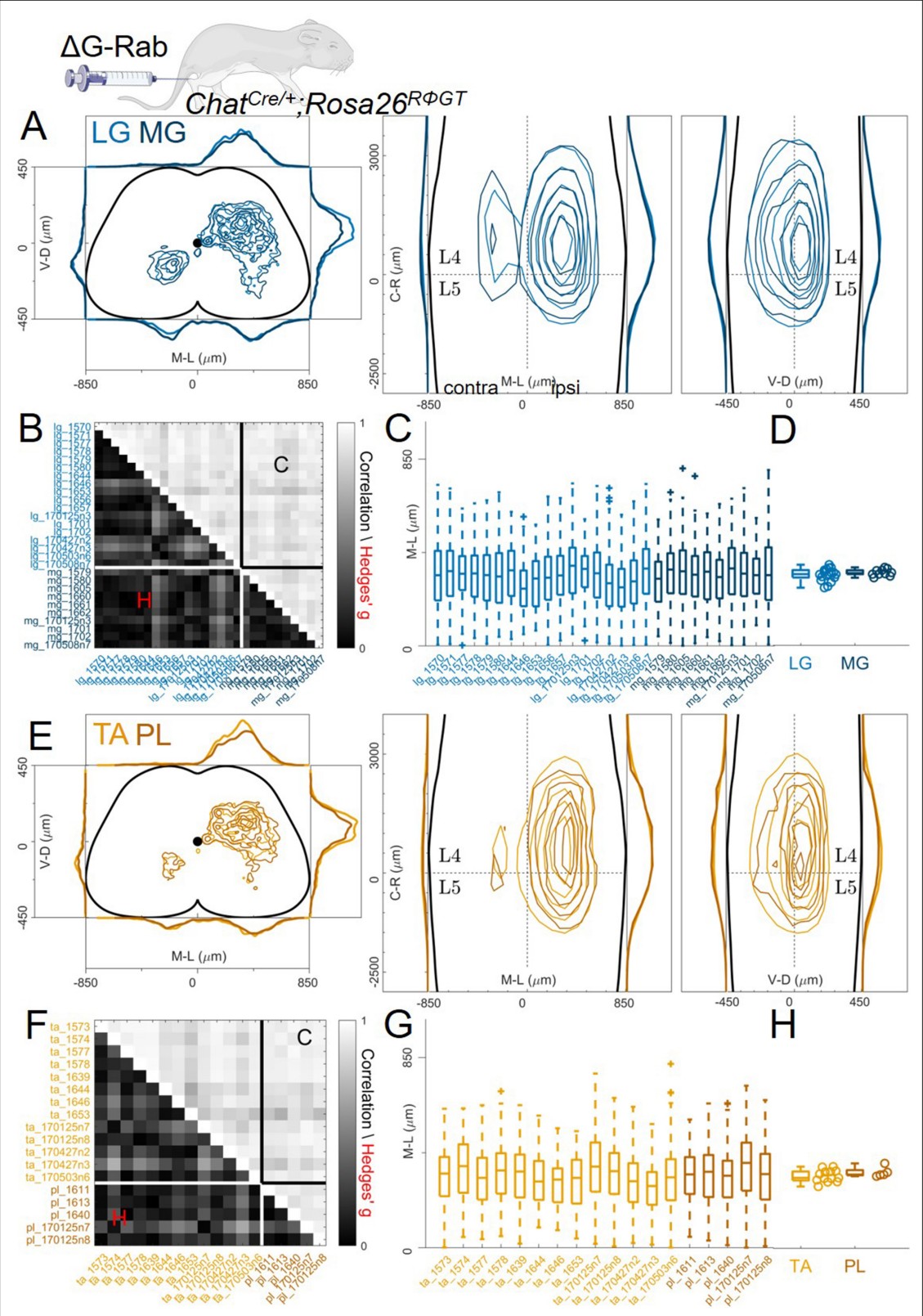

**Figure 5.** Pairs of flexor or extensor muscles show similar distributions of premotor interneurons. (**A**) Comparison of pooled data from extensor muscles LG and MG injections. (**B**) Correlation and absolute value of Hedges' G coefficients across all experiments. (**C**) Box and whisker plots of the mediolateral position of dorsal ipsilateral premotor interneurons for each experiment and distribution of median values (**D**). (**E**) Similar plot as A, showing the distribution of premotor interneurons following injections of the flexor muscles TA and PL. Correlations and absolute value of Hedges' G

*Figure 5 continued on next page*

*Figure 5 continued*

coefficients across each experiment are shown in (**F**). (**G and H**) shows the mediolateral distribution and the position of the median for each experiment, respectively. For each section, the data are scaled to the reference points indicated in the methods in order to account for size differences along the segments.

The online version of this article includes the following source data for figure 5:

**Source data 1.** Hedges'G and correlation across experiments with LG and MG injections in the lower and upper triangular matrix respectively shown in panel B.

**Source data 2.** Median of mediolateral coordinates in the ipsilateral dorsal quadrant for each experiment shown in panel D, LG (n=18 experiments) and MG (n=10 experiments).

**Source data 3.** Hedges'G and correlation across experiments with TA and PL injections in the lower and upper triangular matrix respectively shown in panel F.

**Source data 4.** Median of mediolateral coordinates in the ipsilateral dorsal quadrant for each experiment shown in panel H, TA (n=13 experiments) and PL (n=5 experiments).

μm; hierarchical bootstrapped Hedges'G=0.72, IQR 0.61, 0.84. Medians for TA: $GlyT2^{off}$ = 74 μm and $GlyT2^{on}$ = 437 μm; hierarchical bootstrapped Hedges'G=0.91, IQR 0.82, 0.99). Overall, the data show a clear segregation in the distributions of $GlyT2^{off}$ and $GlyT2^{on}$ premotor interneurons in the ventral half of the spinal cord: $GlyT2^{on}$ interneurons are almost exclusively found in the ipsilateral side while $GlyT2^{off}$ interneurons also present a prominent peak in the contralateral side (*Figure 6G–L*). These observations are reflected in the overall low correlation value in the position of $GlyT2^{off}$ and $GlyT2^{on}$ premotor interneurons for both LG and TA premotor interneurons ($r \leq 0.4$).

Taken together, these findings indicate that while we are able to detect significant differences in the positional organization of premotor interneurons with different transmitter phenotype, we found that the organization of flexor and extensor premotor circuits were always intermingled regardless of their neurotransmitter status.

## Flexor and extensor premotor interneurons tracing in $Olig2^{Cre/+}$; $Rosa26^{RΦGT}$ mice

An important consideration concerning the use of our genetic approach for G complementation is the expression specificity of the Cre driver: recombination in multiple neuronal subtypes can potentially result in loss of monosynaptic restriction and rabies transfer across multiple synapses (*Figure 1D and E*). In the spinal cord, $Chat^{Cre/+}$ is not only expressed in motor neurons but also cholinergic interneurons, including medial partition cells (V0c neurons) that have prominent projections to motor neurons (*Zagoraiou et al., 2009*). Therefore, given that under our experimental conditions, V0c neurons express G and are presynaptic to motor neurons, they could permit disynaptic rabies transfer: first from motor neurons to V0c neurons and second from V0c neurons to their presynaptic partners. However, it is important to note that V0c presynaptic partners have been previously characterised using rabies monosynaptic tracing and comprise many interneurons (and/or axonal arborisations) located in the dorsal laminae of the spinal cord (*Zampieri et al., 2014*), an area that is largely devoid of rabies labelling in our experiments.

In order to test whether disynaptic transfer from premotor interneurons is affecting our analysis, we performed a set of experiments (4 gastrocnemius, GS and 3 TA injections) using the $Olig2^{Cre/+}$ (*Dessaud et al., 2007*) instead of the $Chat^{Cre/+}$ line (*Figure 1E*). This line would ensure recombination in motor neurons but not in V0c or other cholinergic interneurons. However, Olig2 is transiently expressed during embryonic development in subsets of p2 and p3 progenitors (*Chen et al., 2011*). We reasoned that if additional transsynaptic transfer from premotor interneurons is significantly affecting our results, using a different Cre line to drive G expression in a non-overlapping subset of premotor interneurons should result in different labelling patterns. We performed monosynaptic tracing experiments after single injections of ΔG-RabV/mCherry in either the TA or GS muscles of P4 $Olig2^{Cre/+}$; $Rosa26^{RΦGT}$ mice. Six days following injection, we observed interneuron labelling with a pattern similar to that of those performed in $Chat^{Cre/+}$ mice (*Figure 7A–B*). There was no difference in the positional organization of flexor and extensor premotor interneurons in the transverse plane (*Figure 7C–E* individual experiments are shown in *Figure 12—figure supplements 11–12*) as well as along the rostro-caudal axis (*Figure 7—figure supplement 1*), with median values along the mediolateral axis of 295 μm for GS

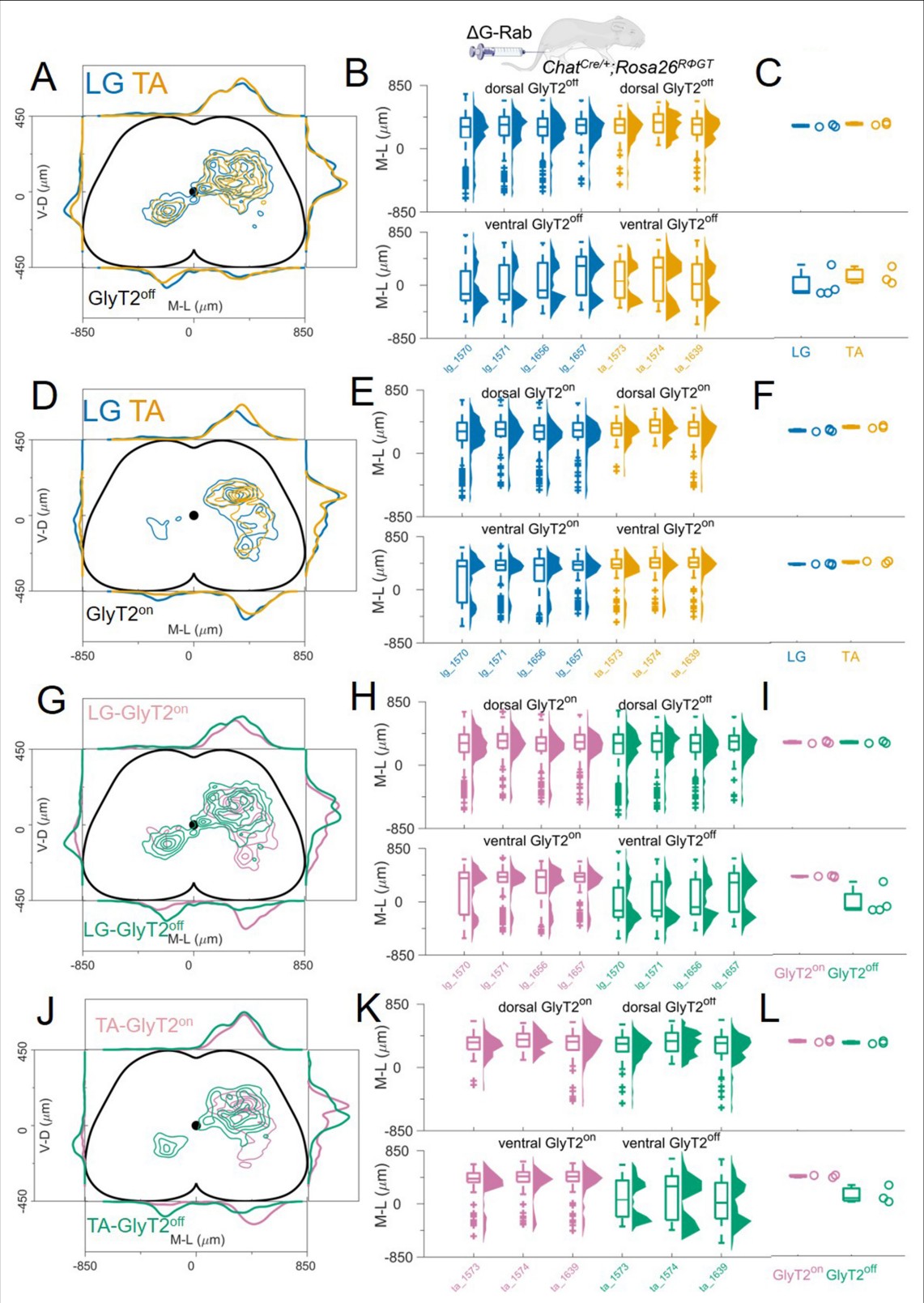

**Figure 6.** GlyT2$^{on}$ and GlyT2$^{off}$ premotor interneurons are distributed similarly for TA and LG. (**A**, **D**) Distribution of GlyT2$^{off}$ (**A**) and GlyT2$^{on}$ (**D**) premotor interneurons following LG and TA injections pooled from 4 LG and 3 TA experiments in *GlyT2-eGFP; Chat$^{Cre/+}$* mice crossed with *Rosa26$^{R\Phi GT}$* mice, indicating that neither class of premotor interneurons is segregated across muscles. Boxplots and violin plots (B for GlyT2$^{off}$ and (**E**) for GlyT2$^{on}$) show uniformity of distribution across experiments in both the dorsal (top) and ventral (bottom) halves of the cord. (**C**) (GlyT2$^{off}$) and (**F**) (GlyT2$^{on}$) show

*Figure 6 continued on next page*

*Figure 6 continued*

boxplots and individual values for the medians of the mediolateral distributions restricted to dorsal (top) or ventral (bottom) part of the cord. Ventral premotor GlyT2off and GlyT2on interneurons are differentially distributed. Comparison of excitatory and inhibitory premotor interneurons in LG (**G**) and TA (**J**) muscles are similar in the dorsal cord, but differ in the ventral cord, where most ipsilateral premotor interneurons are GlyT2on, and the majority of contralateral premotor interneurons are GlyT2off. Boxplots and violin plots of individual experiments are shown in H for LG and K for TA, highlighting the mediolateral differences in the ventral cord. The medians of the ventral and dorsal distributions are shown in I for LG and L for TA. For each section, the data are scaled to the reference points indicated in the methods in order to account for size differences along the segments.

The online version of this article includes the following source data for figure 6:

**Source data 1.** Median of mediolateral coordinates for GlyT2off ventral premotor interneurons from LG (n=4) and TA (n=3) experiments shown in panel C.

**Source data 2.** Median of mediolateral coordinates for GlyT2off dorsal premotor interneurons from LG (n=4) and TA (n=3) experiments shown in panel C.

**Source data 3.** Median of mediolateral coordinates for GlyT2on ventral premotor interneurons from LG (n=4) and TA (n=3) experiments shown in panel F.

**Source data 4.** Median of mediolateral coordinates for GlyT2on dorsal premotor interneurons from LG (n=4) and TA (n=3) experiments shown in panel F.

and 310 µm for TA (bootstrapped Hedges' G=–0.02, IQR –0.07, 0.03).Comparison of the premotor maps obtained from *Chat^{Cre/+}* and *Olig2^{Cre/+}* experiments showed that interneuron distributions were indistinguishable, as shown by the high correlation values across mouse lines and muscles ($r>0.9$) and the low values of hierarchical bootstraps of the Hedges' G coefficients across muscles and *Olig2^{Cre/+}* and *Chat^{Cre/+}* injections (*Figure 7F*). In addition, the median interneuron positions along the medio-lateral axis for each experiment were similar between *Chat^{Cre/+}* (322 µm for GS and 315 µm for TA, including high and low efficiency experiments) and *Olig2^{Cre/+}* (295 µm for GS and 310 µm for TA) animals injected in the same muscle (*Figure 7G*, Hedges' G=0.14 (IQR 0.09, 0.18) for GS pairs and 0.13 (IQR 0.08, 0.18) for TA pairs). Thus, these results indicate that under our experimental conditions the results of tracing experiments done in *Chat^{Cre/+}; Rosa26^{RΦGT}* and *Olig2^{Cre/+}; Rosa26^{RΦGT}* mice are unlikely to be influenced by disynaptic rabies transfer from spinal premotor interneurons.

## Flexor and extensor premotor interneurons tracing with AAV complementation methods

In contrast to previous findings using AAV-G (*Tripodi et al., 2011*), we found extensive intermingling of flexor and extensor premotor interneurons when using Cre-based methods to genetically express G-protein in order to complement rabies replication (*Figure 1D and E*). Therefore, we sought to replicate the previous findings using AAV-G complementation strategies (*Figure 1A and B*). We first performed injections of 1 µl of a 3:1 AAV-G: ΔG-RabV mixture in GS and TA muscles of P2 wild-type mice (*Figure 1A*; *Tripodi et al., 2011*). Two experiments were performed on GS only, with similar protocol, and GS single injections were pooled with the data resulting from the double injections. Ten days following injection, consistent labelling with both viruses was observed throughout the lumbar region (L1: *Figure 8A and L*: *Figure 8B*). Analysis of rabies-labelled interneurons did not reveal any apparent difference in the distributions of premotor circuits controlling the activity of antagonist muscles in the transverse plane as well as along the rostro-caudal axis (*Figure 8C*, individual experiments are shown in *Figure 12—figure supplement 13*). The Hedges' G coefficients of premotor interneuron distributions in the ipsilateral dorsal quadrant for all the experiments had a median of –0.02 (IQR –0.29, 0.28), indicating high uniformity across experiments, as also shown by the high correlation values of positional coordinates of flexor and extensor interneurons (*Figure 8D*). When the analysis was restricted to the medio-lateral positions in the ipsilateral dorsal quadrant, the median distances from the midline were 276 µm for GS and 321 µm for TA (*Figure 8E and F*), a negligible shift compared to the typical interneuron soma diameter. These findings obtained using AAV-G complementation are therefore consistent with the pattern of premotor interneuron mixing identified using genetic-delivery of G-protein.

In these AAV-G complementation experiments, we observed extensive labelling of superficial dorsal interneurons, a feature that was absent when using genetic complementation methods (compare *Figure 8C* with 3 C, 4 C, and 7E). We therefore reasoned that contribution of transsynaptic transfer from sensory afferents, precluded by our genetic complementation approaches, may result in tracing of dorsal and medial interneurons and thus affect analysis of premotor circuits (*Zampieri et al., 2014*). To test this idea, we restricted expression of G to motor neurons by injecting AAV-flex-optimizedG

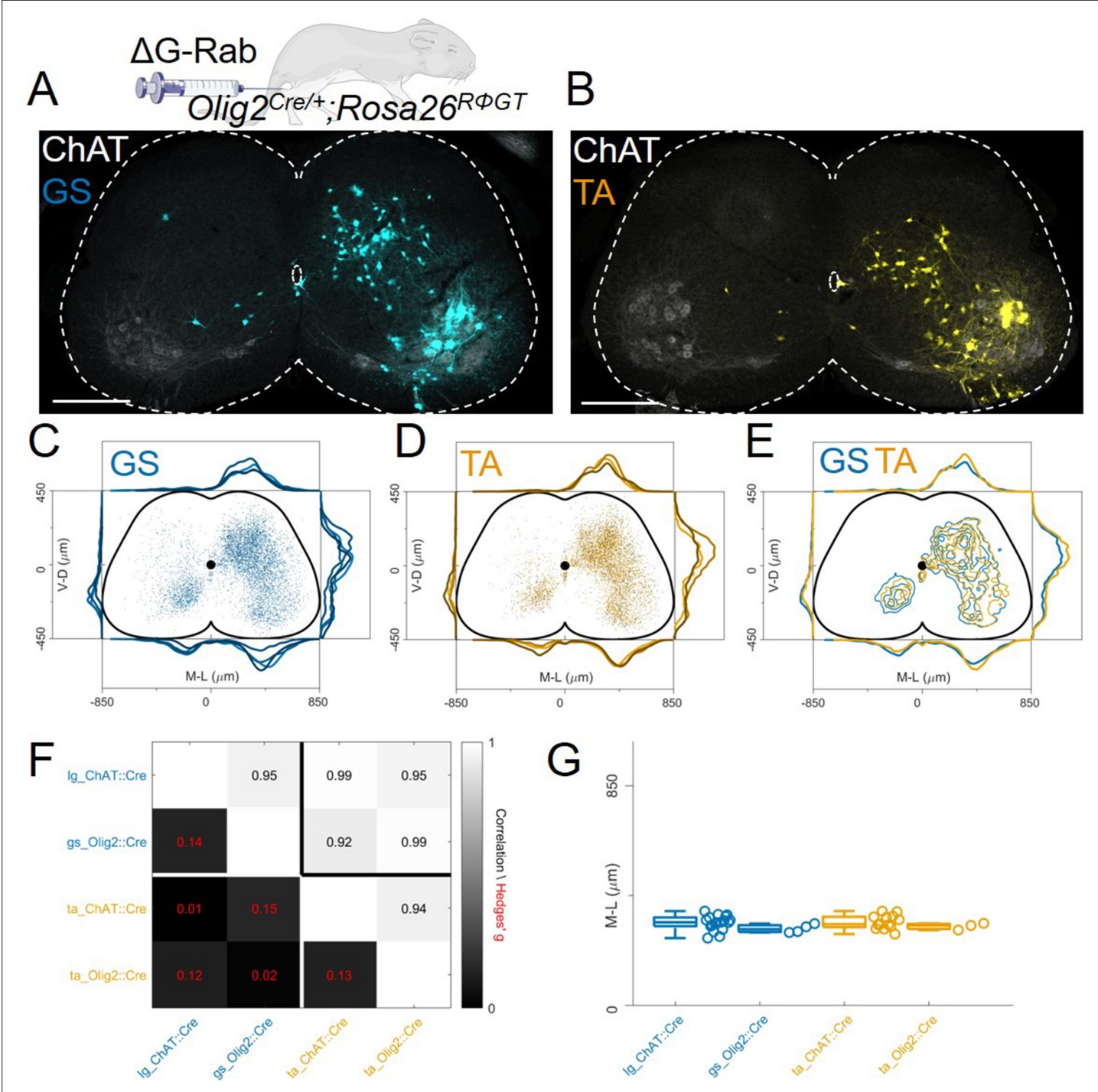

**Figure 7.** The distribution of flexor and extensor premotor INs is similar in *Olig2^{Cre/+}; Rosa26^{RΦGT}*. (A, B) Single lumbar sections form animals injected in the GS (**A**) or TA (**B**) muscles (scale bars 300 μm). (**C-E**) Overlay of individual GS (**C**) and TA (**D**) experiments and pooled experiments (**E**). For each section, the data are scaled to the reference points indicated in the methods in order to account for size differences along the segments. (**F**) Correlation coefficients and absolute value of hierarchical bootstrapped Hedges' G effect sizes between injections of different muscles and using a different driver for Cre expression. (**G**) Box and whisker plots of median values of all the medio-lateral distributions in the dorsal ipsilateral quadrant.

The online version of this article includes the following source data and figure supplement(s) for figure 7:

**Source data 1.** Median of mediolateral coordinates in the ipsilateral dorsal quadrant shown in panel G comparing flexor and extensor injections in Chat^{Cre/+} and Olig2^{Cre/+} mice.

**Figure supplement 1.** distribution of flexor and extensor premotor interneurons pooled across GS and TA injections performed in *Olig2^{Cre/+}; Rosa26^{RΦGT}* mice.

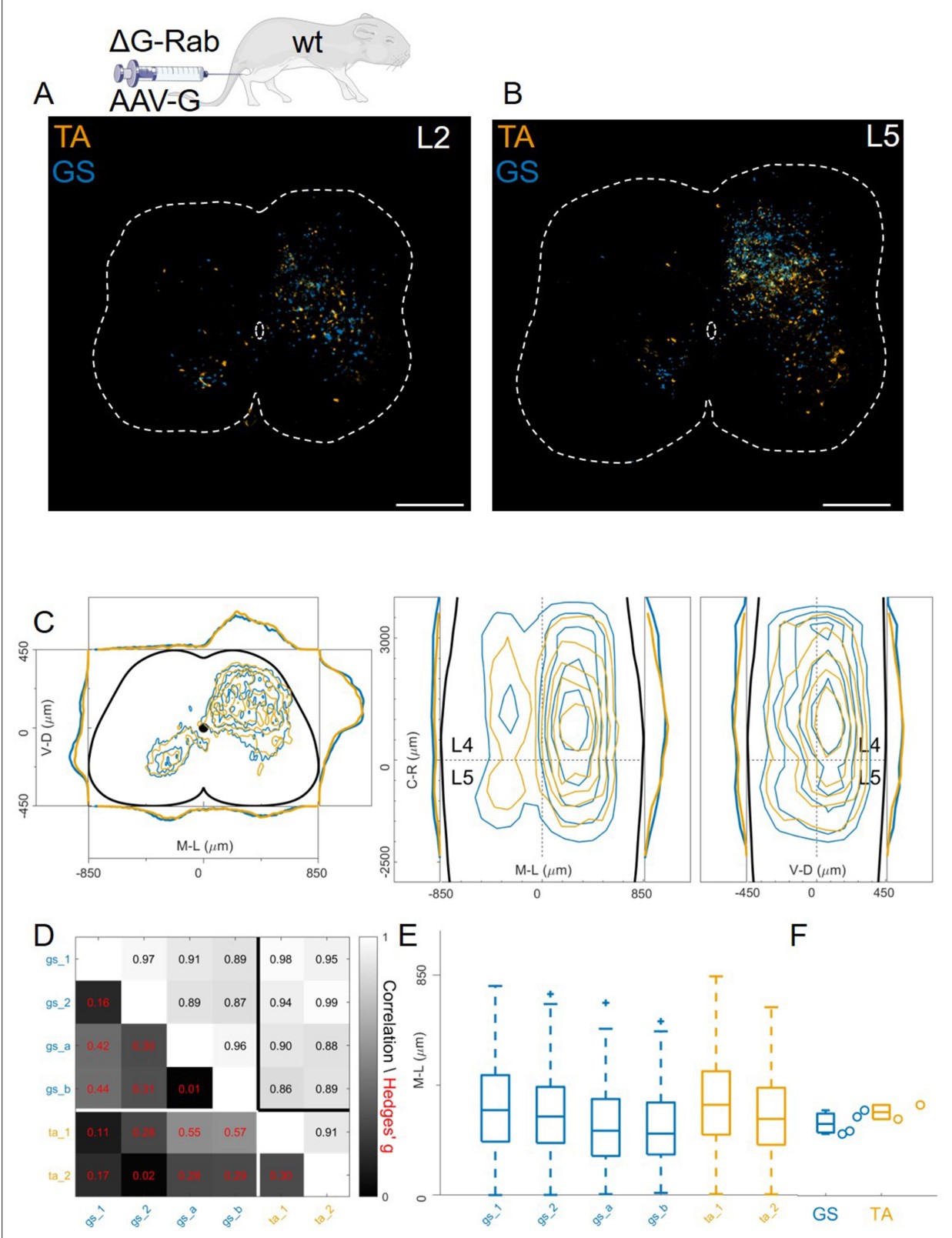

**Figure 8.** Injection of AAV-G and RabV into GS and TA muscles does not reveal segregation of premotor interneurons. Example of double infected upper (**A**) and lower (**B**) lumbar sections. Transverse and longitudinal pooled distributions of premotor interneurons from two experiments are overlapping in all quadrants (**C**). For each section, the data are scaled to the reference points indicated in the methods in order to account for size

*Figure 8 continued on next page*

*Figure 8 continued*

differences along the segments. Individual experiments are highly correlated and have low absolute value of Hedges' G effect sizes (**D**) and with similar distributions and medians in the ipsilateral dorsal quadrant (**E, F**). Scale bars 300 µm.

The online version of this article includes the following source data for figure 8:

**Source data 1.** Median of mediolateral coordinates in the ipsilateral dorsal quadrant for each experiment shown in panel F, GS (n=4) and TA (n=2).

(oG, **Kim et al., 2016**) in GS and TA muscles of *Chat^{Cre/+}* mice (1 µl of a 3:1 AAV-flex-oG: ΔG-RabV mix), a strategy that was previously used to show segregation of flexor and extensor premotor circuits (**Figure 1B**; **Wang et al., 2017**). In these experiments (4 GS and 3 TA injections), the distributions of GS and TA premotor interneurons were similar (**Figure 9A–C**, individual experiments are shown in **Figure 12—figure supplements 14–15**), with high correlation values across experiments (**Figure 9D**). Analysis of the medio-lateral positioning in the ipsilateral dorsal quadrant showed that the median position of TA related interneurons was shifted by only 33 µm with respect to that of the GS related interneurons, and that this was in a *medial* direction (383 µm for GS and 350 µm for TA; Hedges' G from hierarchical bootstrap = –0.15, IQR –0.24, –0.05, **Figure 9E and F**).

## Anterograde transsynaptic jumps alter the distribution of premotor interneurons

Regardless of whether the expression of G was restricted to motor neurons (**Figure 9**) or not (**Figure 8**), we observed no differences in the mediolateral position of dorsal ipsilateral premotor interneurons related to flexor or extensor muscles. However, the use of AAV-oG-Flex that, contrarily to AAV-G, prevents anterograde jumps by restricting G expression to motor neurons (**Figure 1A–B**), led to important differences. Indeed, despite being performed and analysed by the same lab at similar timepoints for injection (P1-P2) and harvesting (7–8 days post-injection), experiments using AAV-G result in an extensive labelling of superficial dorsal horn neurons for both tested muscles (**Figure 10A**).

The median values along the dorsoventral axis were 249 and 375 µm for GS and 295 and 344 µm for TA following AAV-G or AAV-Flex-oG injections respectively. The Hedges' G bootstrapped coefficients were –0.55 (IQR –0.64,–0.46) for the GS pairs and –0.14 (IQR –0.23,–0.08) for the TA pairs. In addition, there was an excess of medially located infected neurons, in both GS and TA experiments, when the expression of G was not restricted to motor neurons. The dorsoventral and mediolateral differences in the distributions observed in AAV-G and AAV-oG-Flex experiments were substantial for both GS and TA, thus indicating that when G could be expressed by sensory neurons (AAV-G muscle injection), the average position of labelled cells extended more dorsally and medially compared to experiments in which G could only be expressed in motor neurons (AAV-oG-Flex). The observed mediodorsal shift is compatible with a contribution of anterograde tracing from sensory neurons (**Figure 10A–C**), which is absent when G is restricted to motor neurons, either by using AAV-oG-Flex, or by genetic restrictions. Nonetheless, regardless of the possible effects derived from anterograde tracing, the distributions of labelled cells observed in flexor and extensor experiments did not differ.

## Flexor and extensor premotor interneurons organization using pseudorabies virus timed tracing experiments

Finally, we studied organization of premotor circuits using a different viral tracing method. Pseudorabies virus (PRV) is a neurotropic virus that travels transsynaptically in the retrograde direction. It has previously been used to resolve the connectivity order in polysynaptic circuits (**Gu et al., 2017**; **Ugolini, 2020**) using timed infections protocols. In particular, the Bartha strain has been shown to trace spinal premotor circuits around 40 hours (hr) after injection (**Jovanovic et al., 2010**). We simultaneously injected P11 GS and TA muscles with 0.5 µl of PRV-Bartha (PRV-152 and PRV-614). Analysis of the spinal cords two days after injection showed extensive labelling of interneurons throughout the lumbar segments (**Figure 11A–B**). There were no differences in the distributions or median mediolateral positioning of flexor and extensor premotor interneurons (**Figure 11C**, individual experiments are shown in **Figure 12—figure supplement 16**), and correlation analysis showed high correlation coefficients between experiments and across muscles (**Figure 11D**) and low values of the Hedges' G for pairs of experiments (median 0.09, IQR 0.04, 0.22). Accordingly, the medians of mediolateral

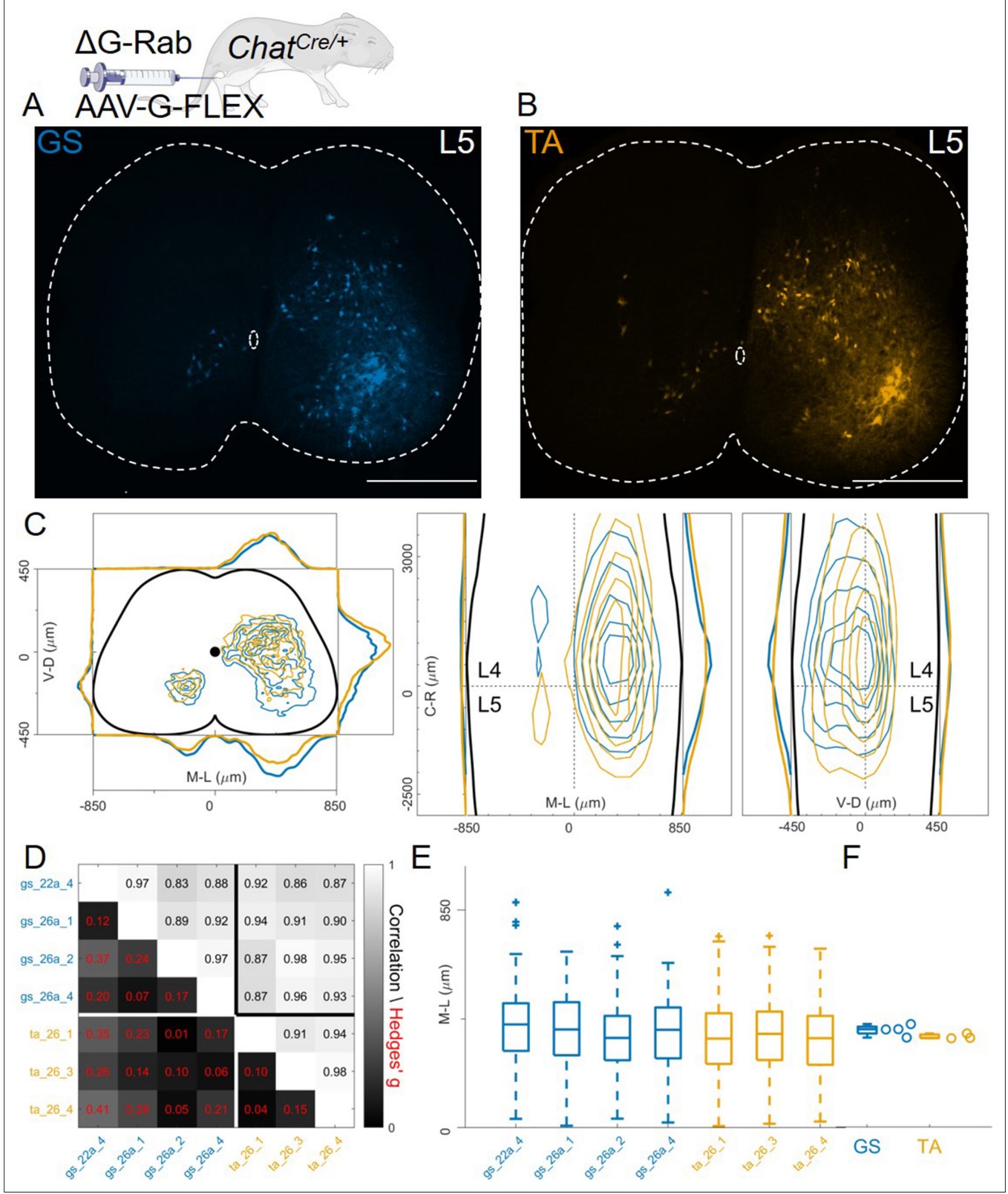

**Figure 9.** Injection of AAV-FLEX-G in Chat^Cre/+ mice gives rise to overlapping distributions of flexor and extensor related premotor interneurons. Extensive labelling is observed in two lower lumbar sections of two GS (**A**) and TA (**B**) injected mice. The pooled distributions of flexor and extensor related infected neurons are similar on the transverse and longitudinal planes (**C**). For each section, the data are scaled to the reference points indicated

*Figure 9 continued on next page*

*Figure 9 continued*

in the methods in order to account for size differences along the segments. There is strong correlation and low effect sizes across individual experiments (**D**) and distributions and medians in the ipsilateral dorsal quadrant are not different (**E, F**). Scale bars 400 µm.

The online version of this article includes the following source data for figure 9:

**Source data 1.** Median of mediolateral coordinates in the ipsilateral dorsal quadrant for each experiment shown in panel F, GS (n=4) and TA (n=3).

positions of flexor and extensor premotor interneurons were similar (353 µm for GS and 366 µm for TA; *Figure 11E and F*).

We finally pooled together all the results obtained with different methods. In all cases, the correlations between flexor and extensor muscles labelled with the same methods were very high (*Figure 12A*). The lowest correlation coefficients (which were still high) were typically observed between experiments with AAV-G injection in wild type mice and all the other methods, regardless of the muscle injected, consistent with a substantial contamination from interneurons anterogradely labelled through sensory afferents. Overall, the medians of the mediolateral positions (*Figure 12B*) did not differ between muscles for any of the methods used and the range of bootstrapped Hedges' G coefficient was between –0.23 and 0.17 across pairs of antagonist muscles throughout five different experimental paradigms tested. The whole set of experiments is shown in *Figure 12—figure supplements 1–16*. The labelling of experiments keeps the original lab conventions and is also reported in *Table 1*. Taken together, our findings indicate that premotor interneurons innervating flexor and extensor motor neurons are not spatially segregated.

## Discussion

Spinal circuits are responsible for integrating descending commands and sensory information to ensure precise control and coordination of movement. In order to understand how these circuits organise movement, it is necessary to first identify and then study the roles and contributions of spinal interneurons that control the activity of different muscles. Previous work (*Tripodi et al., 2011*) exploited rabies monosynaptic tracing to examine the organization of premotor circuits controlling the activity of selected muscles. These studies, using intramuscular injection of an AAV expressing the rabies glycoprotein G, identified clear segregation in the spatial organization of premotor interneurons directing the activity of flexor and extensor muscles (*Tripodi et al., 2011*; *Wang et al., 2017*; *Takeoka and Arber, 2019*). In contrast, our study using either genetic and AAV complementation of G expression, as well as PRV timed infections, demonstrates complete spatial overlap amongst flexor and extensor premotor interneurons.

### Mouse genetic-based strategies for rabies monosynaptic tracing of premotor circuits

We opted for a mouse genetic strategy that was previously used to trace premotor circuits of vibrissal, orofacial, and forelimb muscles (*Takatoh et al., 2013*; *Stanek et al., 2014*; *Skarlatou et al., 2020*). Combining a conditional allele expressing G from the rosa locus (*Rosa26^{RΦGT}* mice; *Takatoh et al., 2013*) with either the *Chat^{Cre/+}* or *Olig2^{Cre/+}* lines (*Figure 1D–E*) is predicted to result in high levels of G expression in all motor neurons at the time of rabies muscle injection and therefore in robust transsynaptic transfer. Indeed, under these conditions, several hundred premotor neurons can be reproducibly traced in each experiment (*Table 1*; *Skarlatou et al., 2020*; *Ronzano et al., 2021*). On the other hand, this strategy suffers from the undesirable consequences of lineage tracing, namely G complementation in all Cre expressing cells in the spinal cord, including those that transiently activate the targeted promoter during development. This problem is in part shared with the AAV-based experiments using intraspinal injection of AAV-flex-G in *Chat^{Cre/+}* mice (*Figure 1C*; *Takeoka and Arber, 2019*). Thus, it is unlikely that the differences in the results obtained using these two strategies were caused by disynaptic transfer through cholinergic interneurons. Indeed, work using rabies monosynaptic tracing to identify spinal neurons presynaptic to the most prominent population of premotor cholinergic interneurons, V0c neurons, found a high density of pre-V0c neurons located in superficial laminae of the dorsal horn (*Zampieri et al., 2014*), an area where no labelling was observed in our mouse genetic-based experiments.

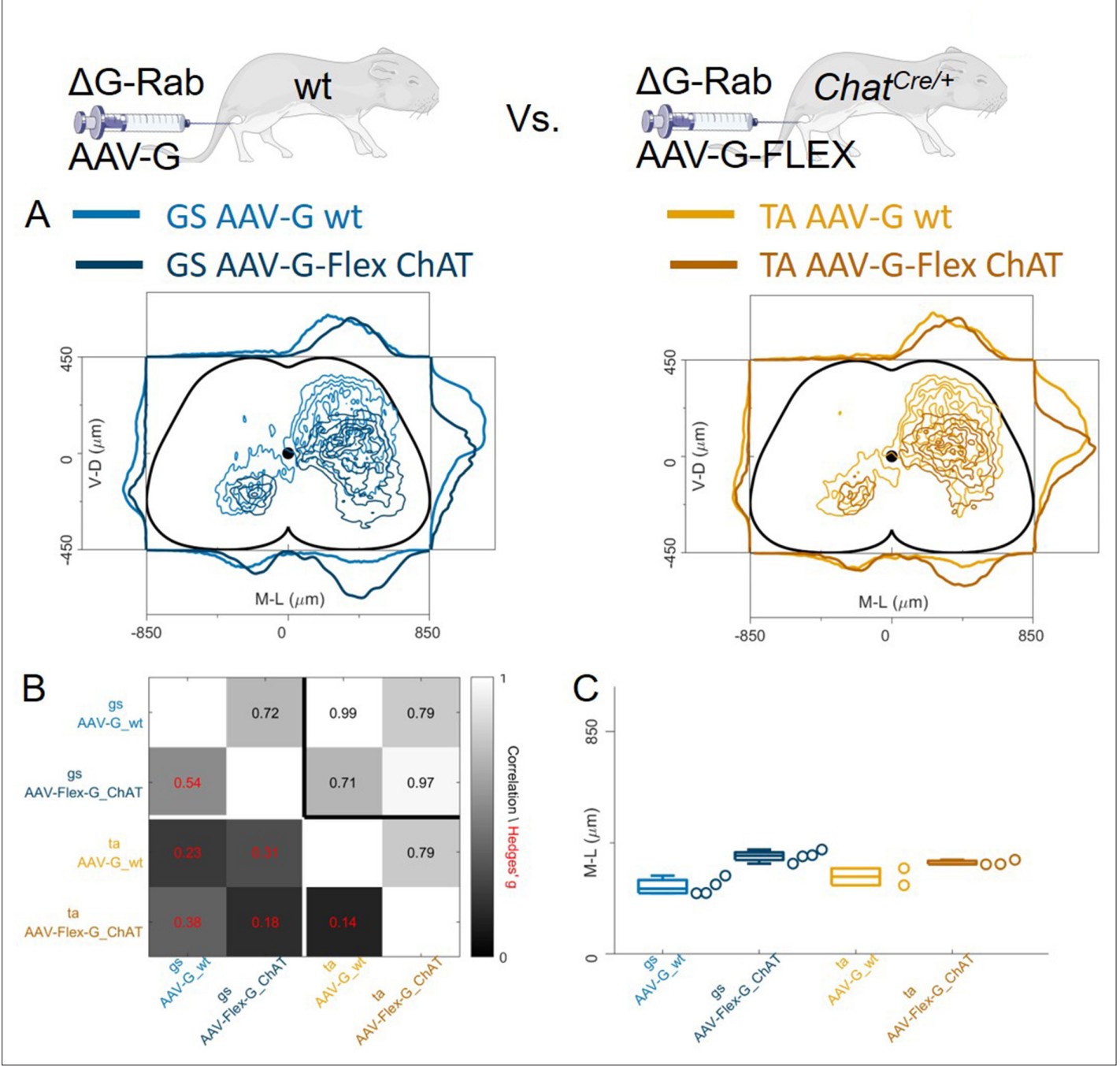

**Figure 10.** Injection of AAV-FLEX-G prevents anterograde spreading of RabV from afferent terminals. (**A**) Side by side comparison of pooled transverse distribution of infected interneurons from GS (left) and TA (right) with AAV-G or AAV-FLEX-G and RabV. The distributions obtained with AAV-G show a marked mediolateral shift and extensive labelling of superficial dorsal horn neurons, a feature that is absent when anterograde transfer is prevented by restricting the expression of the G protein to motor neurons only. The distributions obtained are very similar across the different muscles, but the mediolateral shift is reflected in the lower correlation values and high hierarchical bootstrapped effect size (**B**) and difference in medians (**C**) between AAV-G and AAV-FLEX-G injections.

The online version of this article includes the following source data for figure 10:

**Source data 1.** Panel C showing the comparison of median of mediolateral coordinates in the ipsilateral dorsal quadrant for injections of AAV-G in wild type and AAV-Flex-G in Chat^{Cre/+} mice for all GS and TA injections.

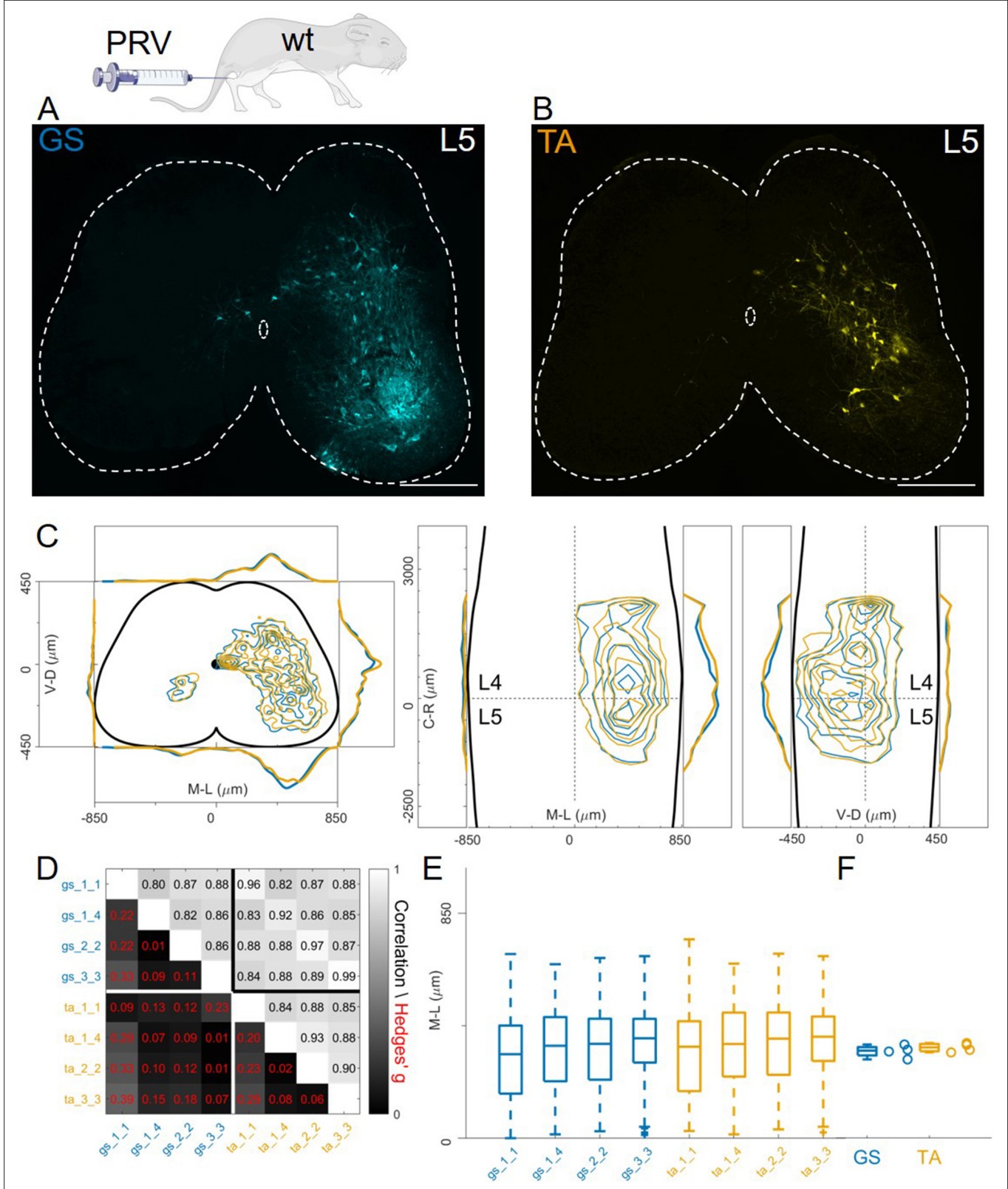

**Figure 11.** Injections of PRV-Bartha in GS and TA muscles give rise to similar distribution of premotor interneurons. Example of a lower lumbar section of an animal injected in GS(A) and TA (**B**). The distribution of premotor interneurons are similar along all axis (**C**). For each section, the data are scaled to the reference points indicated in the methods in order to account for size differences along the segments. Individual experiments are highly correlated,

*Figure 11 continued on next page*

Figure 11 continued

with small effect size (**D**) and their median values along the mediolateral axis of the dorsal ipsilateral quadrant are identical for flexor and extensor injections (**E, F**). Scale bars 300 μm.

The online version of this article includes the following source data for figure 11:

**Source data 1.** Median of mediolateral coordinates in the ipsilateral dorsal quadrant for each experiment shown in panel F, GS (n=4) and TA (n=4).

Nonetheless, to further explore the possibility of disynaptic pathways via cholinergic spinal interneurons, we examined the organization of flexor and extensor premotor circuits in experiments conducted with the *Olig2^{Cre/+}; Rosa26^{RΦGT}* line. Olig2 is expressed in motor neurons and in subsets of p2 and p3 progenitors, thus potentially generating opportunities for disynaptic transfer through V2 and V3 premotor interneurons (*Chen et al., 2011*). We did not find any difference in the distribution of premotor interneurons obtained in these mice compared to those in *Chat^{Cre/+}; Rosa26^{RΦGT}* mice. Therefore, it appears that disynaptic transfer is not a frequent event in our experimental conditions; otherwise, neuronal labelling in *Chat^{Cre/+}* experiments would reflect the contribution of cholinergic interneurons, and *Olig2^{Cre/+}* experiments would reflect jumps through V2 and V3 interneurons. It is also important to consider the timing of rabies transsynaptic transfer (*Ugolini, 2011*). The earliest expression of rabies in primary infected motor neurons is first observed 3–4 days after injection and monosynaptic transfer not earlier than 5 days after injection, with strong labelling observed around 7–8 days. Since in our experiments, mice were sacrificed between 6–9 days following RabV injections (8–9 days in *Chat^{Cre/+}* and 6 days in *Olig2^{Cre/+}* mice), it is unlikely that many, if any, double jumps would have occurred in this time window. We cannot exclude that at least some of the labelled interneurons were generated by second-order transfers, but arguably these are rare events and unlikely to be the source of the different results obtained in AAV vs mouse genetic experiments.

## AAV- based strategies for rabies monosynaptic tracing of premotor circuits

In previous studies in which segregation of flexor and extensor premotor interneurons have been observed, AAV was used to express G in motor neurons. In the first report, AAV-G and ΔG-RabV were co-injected intramuscularly in wild type mice (*Tripodi et al., 2011*). This approach has the advantage of complementing G only in motor neurons projecting to the targeted muscle, thus avoiding the problem of G expression in spinal interneurons that could lead to loss of monosynaptic restriction. However, since sensory neurons in the dorsal root ganglia also innervate muscles, such strategy could lead to anterograde transsynaptic spread to the spinal cord through the sensory route (*Figure 1A*, *Zampieri et al., 2014*). In order to avoid this problem, intramuscular co-injection of a conditional AAV vector (AAV-flex-G) with ΔG-RabV in *Chat^{Cre/+}* mice was used (*Figure 1B*, *Wang et al., 2017*). In this more stringent condition, G would only be expressed in motor neurons. A more recent study used intraspinal injection of AAV-flex-G in *Chat^{Cre/+}* mice (*Figure 1C*, *Takeoka and Arber, 2019*), which would also avoid transfer from sensory neurons. However, despite the fact that these AAV-based strategies have distinct advantages and disadvantages, they all resulted in labelling of flexor and extensor premotor interneurons with distributions that were medio-laterally segregated in the dorsal ipsilateral quadrant of the spinal cord in experiments performed on neonatal (*Tripodi et al., 2011*) and adult (*Takeoka and Arber, 2019*) mouse hindlimbs, as well as neonatal forelimbs (*Wang et al., 2017*).

In order to resolve the discrepancy with the results obtained with genetic complementation, we tried to replicate the previous findings by directly testing two of the AAV complementation strategies, namely AAV-G and ΔG-RabV co-injection in wild type mice (*Figure 1A*; *Tripodi et al., 2011*), and AAV-flex-G and ΔG-RabV co-injection in *Chat^{Cre/+}* mice (*Figure 1B*; *Wang et al., 2017*). Given the small size of the injected muscles, we limited our injection volumes to 1 μl, as opposed to the 5 μl used in the original study (*Tripodi et al., 2011*) routinely checking for injection specificity by careful examination of the muscles. Surprisingly, despite our attempts at replication, we did not observe segregation of flexor and extensor premotor circuits. However, the contribution of anterograde transsynaptic spread to the spinal cord through the sensory route was clearly detected in AAV-G experiments (but, as expected, not in AAV-flex-oG experiments), reflecting the contribution

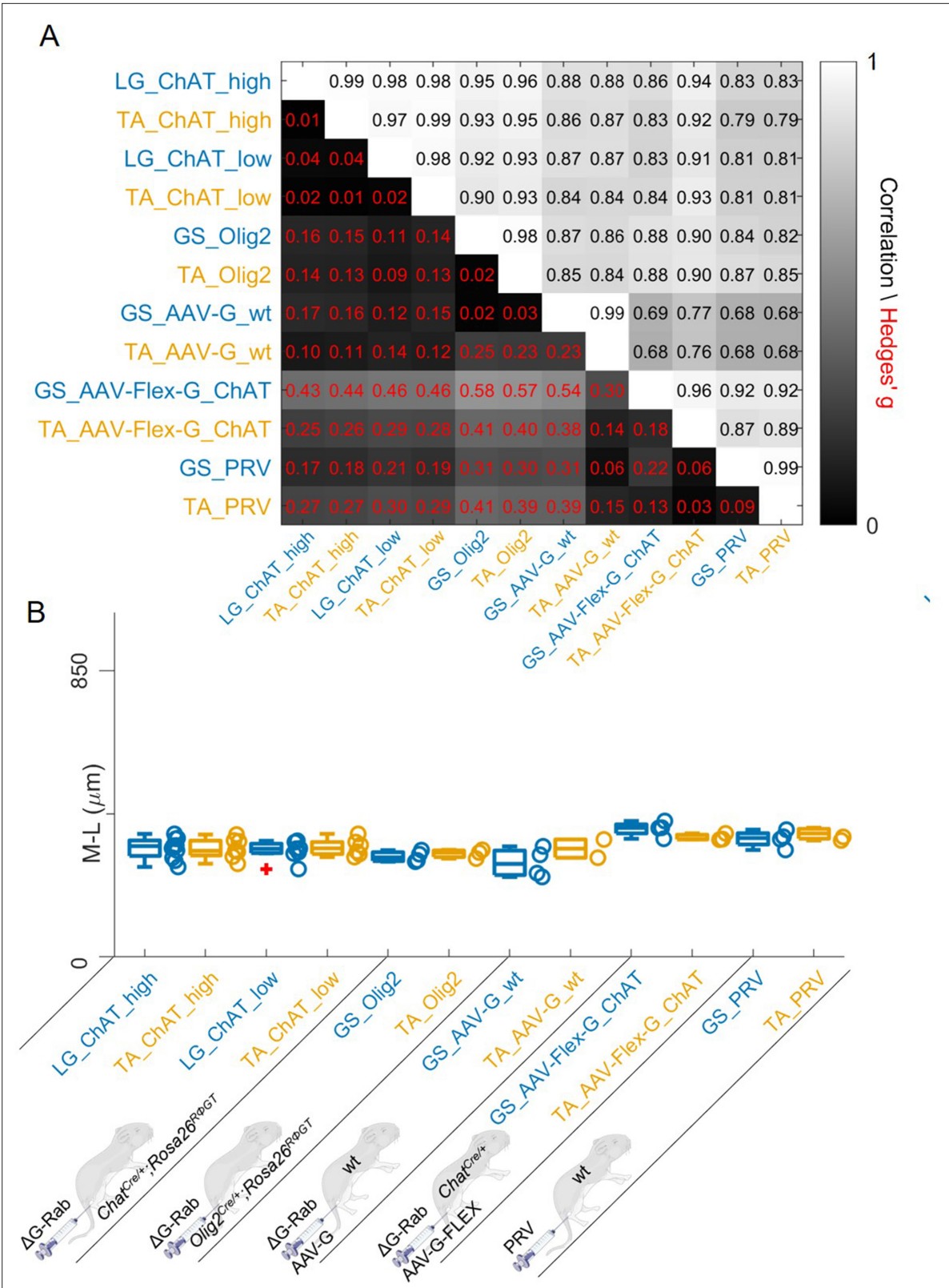

**Figure 12.** Regardless of the tracing methods, flexor and extensor related premotor interneurons distributions are always overlapping. The correlation and absolute value of hierarchical bootstrapped Hedges' G coefficient matrix across pooled experiments obtained with different injection strategies exhibits high values of correlation and low effect sizes across different techniques and across the two different muscles (**A**). The median values of the mediolateral position of premotor interneurons in the dorsal ipsilateral quadrant are similar for all conditions (**B**).

*Figure 12 continued on next page*

*Figure 12 continued*

The online version of this article includes the following source data and figure supplement(s) for figure 12:

**Source data 1.** Panel B showing the comparison of median of mediolateral coordinates of extensor (LG or GS) and flexor (TA) in the ipsilateral dorsal quadrant for all the viral tracing techniques used in this paper.

**Figure supplement 1.** Distribution of motor neurons (triangles) and premotor interneurons (dots) of 3 LG (blue) and TA (yellow) double injections (UCL) of rabies virus in *Chat$^{Cre/+}$; Rosa26$^{RΦGT}$* mice.

**Figure supplement 1—source data 1.** Cartesian x-y-z coordinates of interneurons (sheet 1) and motor neurons (sheet 2) from LG injection 170427n2(UCL).

**Figure supplement 1—source data 2.** Cartesian x-y-z coordinates of interneurons (sheet 1) and motor neurons (sheet 2) from LG injection 170427n3 (UCL).

**Figure supplement 1—source data 3.** Cartesian x-y-z coordinates of interneurons (sheet 1) and motor neurons (sheet 2) from LG injection 170503n6 (UCL).

**Figure supplement 1—source data 4.** Cartesian x-y-z coordinates of interneurons (sheet 1) and motor neurons (sheet 2) from TA injection 170427n2 (UCL).

**Figure supplement 1—source data 5.** Cartesian x-y-z coordinates of interneurons (sheet 1) and motor neurons (sheet 2) from TA injection 170427n3 (UCL).

**Figure supplement 1—source data 6.** Cartesian x-y-z coordinates of interneurons (sheet 1) and motor neurons (sheet 2) from TA injection 170503n6 (UCL).

**Figure supplement 2.** Distribution of motor neurons (triangles) and premotor interneurons (dots) of 2 LG (blue) and TA (yellow) double injections (UoG) of rabies virus in *Chat$^{Cre/+}$; Rosa26$^{RΦGT}$* mice.

**Figure supplement 2—source data 1.** Cartesian x-y-z coordinates of interneurons (sheet 1) and motor neurons (sheet 2) from LG injection 1577 (UoG).

**Figure supplement 2—source data 2.** Cartesian x-y-z coordinates of interneurons (sheet 1) and motor neurons (sheet 2) from LG injection 1578 (UoG).

**Figure supplement 2—source data 3.** Cartesian x-y-z coordinates of interneurons (sheet 1) and motor neurons (sheet 2) from TA injection 1577 (UoG).

**Figure supplement 2—source data 4.** Cartesian x-y-z coordinates of interneurons (sheet 1) and motor neurons (sheet 2) from TA injection 1578 (UoG).

**Figure supplement 3.** Distribution of motor neurons (triangles) and premotor interneurons (dots) of 2 LG (blue) and MG (dark blue) double injections (UCL) of rabies virus in *Chat$^{Cre/+}$; Rosa26$^{RΦGT}$* mice.

**Figure supplement 3—source data 1.** Cartesian x-y-z coordinates of interneurons (sheet 1) and motor neurons (sheet 2) from LG injection 170125n3 (UCL).

**Figure supplement 3—source data 2.** Cartesian x-y-z coordinates of interneurons (sheet 1) and motor neurons (sheet 2) from LG injection 170508n7 (UCL).

**Figure supplement 3—source data 3.** Cartesian x-y-z coordinates of interneurons (sheet 1) and motor neurons (sheet 2) from MG injection 170125n3 (UCL).

**Figure supplement 3—source data 4.** Cartesian x-y-z coordinates of interneurons (sheet 1) and motor neurons (sheet 2) from MG injection 170508n7 (UCL).

**Figure supplement 4.** Distribution of motor neurons (triangles) and premotor interneurons (dots) of 4 LG (blue) and MG (yellow) double injections (UoG) of rabies virus in *Chat$^{Cre/+}$; Rosa26$^{RΦGT}$* mice.

**Figure supplement 4—source data 1.** Cartesian x-y-z coordinates of interneurons (sheet 1) and motor neurons (sheet 2) from LG injection 1579 (UoG).

**Figure supplement 4—source data 2.** Cartesian x-y-z coordinates of interneurons (sheet 1) and motor neurons (sheet 2) from LG injection 1580 (UoG).

**Figure supplement 4—source data 3.** Cartesian x-y-z coordinates of interneurons (sheet 1) and motor neurons (sheet 2) from LG injection 1701(UoG).

**Figure supplement 4—source data 4.** Cartesian x-y-z coordinates of interneurons (sheet 1) and motor neurons (sheet 2) from LG injection 1702 (UoG).

**Figure supplement 4—source data 5.** Cartesian x-y-z coordinates of interneurons (sheet 1) and motor neurons (sheet 2) from MG injection 1579 (UoG).

**Figure supplement 4—source data 6.** Cartesian x-y-z coordinates of interneurons (sheet 1) and motor neurons (sheet 2) from MG injection 1580 (UoG).

**Figure supplement 4—source data 7.** Cartesian x-y-z coordinates of interneurons (sheet 1) and motor neurons (sheet 2) from MG injection 1701 (UoG).

**Figure supplement 4—source data 8.** Cartesian x-y-z coordinates of interneurons (sheet 1) and motor neurons (sheet 2) from MG injection 1702 (UoG).

**Figure supplement 5.** Distribution of motor neurons (triangles) and premotor interneurons (dots) of 2 TA (yellow) and PL (dark orange) double injections (UCL) of rabies virus in *Chat$^{Cre/+}$; Rosa26$^{RΦGT}$* mice.

**Figure supplement 5—source data 1.** Cartesian x-y-z coordinates of interneurons (sheet 1) and motor neurons (sheet 2) from PL injection 170125n7 (UCL).

**Figure supplement 5—source data 2.** Cartesian x-y-z coordinates of interneurons (sheet 1) and motor neurons (sheet 2) from PL injection 170125n8 (UCL).

**Figure supplement 5—source data 3.** Cartesian x-y-z coordinates of interneurons (sheet 1) and motor neurons (sheet 2) from TA injection 170125n7 (UCL).

**Figure supplement 5—source data 4.** Cartesian x-y-z coordinates of interneurons (sheet 1) and motor neurons (sheet 2) from TA injection 170125n8

*Figure 12 continued*

(UCL).

**Figure supplement 6.** Distribution of motor neurons (triangles) and premotor interneurons (dots) of 4 MG (dark blue) single injections (UoG) of rabies virus in *Chat^{Cre/+}; Rosa26^{RΦGT}* mice.

**Figure supplement 6—source data 1.** Cartesian x-y-z coordinates of interneurons (sheet 1) and motor neurons (sheet 2) from MG injection 1605 (UoG).

**Figure supplement 6—source data 2.** Cartesian x-y-z coordinates of interneurons (sheet 1) and motor neurons (sheet 2) from MG injection 1660 (UoG).

**Figure supplement 6—source data 3.** Cartesian x-y-z coordinates of interneurons (sheet 1) and motor neurons (sheet 2) from MG injection 1661 (UoG).

**Figure supplement 6—source data 4.** Cartesian x-y-z coordinates of interneurons (sheet 1) and motor neurons (sheet 2) from MG injection 1662 (UoG).

**Figure supplement 7.** Distribution of motor neurons (triangles) and premotor interneurons (dots) of 3 PL (dark orange) single injections (UoG) of rabies virus in *Chat^{Cre/+}; Rosa26^{RΦGT}* mice.

**Figure supplement 7—source data 1.** Cartesian x-y-z coordinates of interneurons (sheet 1) and motor neurons (sheet 2) from PL injection 1611 (UoG).

**Figure supplement 7—source data 2.** Cartesian x-y-z coordinates of interneurons (sheet 1) and motor neurons (sheet 2) from PL injection 1613 (UoG).

**Figure supplement 7—source data 3.** Cartesian x-y-z coordinates of interneurons (sheet 1) and motor neurons (sheet 2) from PL injection 1640 (UoG).

**Figure supplement 8.** Distribution of motor neurons (triangles) and premotor interneurons (dots) of 3 LG (blue) and TA (yellow) double injections with low titre rabies virus in *Chat^{Cre/+}; Rosa26^{RΦGT}* mice (UoG).

**Figure supplement 8—source data 1.** Cartesian x-y-z coordinates of interneurons (sheet 1) and motor neurons (sheet 2) from LG injection 1644 (UoG).

**Figure supplement 8—source data 2.** Cartesian x-y-z coordinates of interneurons (sheet 1) and motor neurons (sheet 2) from LG injection 1646 (UoG).

**Figure supplement 8—source data 3.** Cartesian x-y-z coordinates of interneurons (sheet 1) and motor neurons (sheet 2) from LG injection 1653 (UoG).

**Figure supplement 8—source data 4.** Cartesian x-y-z coordinates of interneurons (sheet 1) and motor neurons (sheet 2) from TA injection 1644 (UoG).

**Figure supplement 8—source data 5.** Cartesian x-y-z coordinates of interneurons (sheet 1) and motor neurons (sheet 2) from TA injection 1646 (UoG).

**Figure supplement 8—source data 6.** Cartesian x-y-z coordinates of interneurons (sheet 1) and motor neurons (sheet 2) from TA injection 1653 (UoG).

**Figure supplement 9.** Distribution of motor neurons (triangles) and premotor interneurons (dots) of 4 LG (blue) single injections with low titre rabies virus in *Chat^{Cre/+}; Rosa26^{RΦGT}* mice (UoG).

**Figure supplement 9—source data 1.** Cartesian x-y-z coordinates of interneurons (sheet 1) and motor neurons (sheet 2) from LG injection 1656 (UoG).

**Figure supplement 9—source data 2.** Cartesian x-y-z coordinates of interneurons (sheet 1) and motor neurons (sheet 2) from LG injection 1657 (UoG).

**Figure supplement 9—source data 3.** Cartesian x-y-z coordinates of interneurons (sheet 1) and motor neurons (sheet 2) from LG injection 1570 (UoG).

**Figure supplement 9—source data 4.** Cartesian x-y-z coordinates of interneurons (sheet 1) and motor neurons (sheet 2) from LG injection 1571 (UoG).

**Figure supplement 10.** Distribution of motor neurons (triangles) and premotor interneurons (dots) of 3 TA (yellow) single injections with low titre rabies virus in *Chat^{Cre/+}; Rosa26^{RΦGT}* mice (UoG).

**Figure supplement 10—source data 1.** Cartesian x-y-z coordinates of interneurons (sheet 1) and motor neurons (sheet 2) from TA injection 1573 (UoG).

**Figure supplement 10—source data 2.** Cartesian x-y-z coordinates of interneurons (sheet 1) and motor neurons (sheet 2) from TA injection 1574 (UoG).

**Figure supplement 10—source data 3.** Cartesian x-y-z coordinates of interneurons (sheet 1) and motor neurons (sheet 2) from TA injection 1639 (UoG).

**Figure supplement 11.** Distribution of motor neurons (triangles) and premotor interneurons (dots) of 4 GS (blue) single injections with rabies virus in *Olig2^{Cre/+}; Rosa26^{RΦGT}* mice (MDC).

**Figure supplement 11—source data 1.** Cartesian x-y-z coordinates of interneurons (sheet 1) and motor neurons (sheet 2) from GS injection 353 (MDC).

**Figure supplement 11—source data 2.** Cartesian x-y-z coordinates of interneurons (sheet 1) and motor neurons (sheet 2) from GS injection 399 (MDC).

**Figure supplement 11—source data 3.** Cartesian x-y-z coordinates of interneurons (sheet 1) and motor neurons (sheet 2) from GS injection 1332 (MDC).

**Figure supplement 11—source data 4.** Cartesian x-y-z coordinates of interneurons (sheet 1) and motor neurons (sheet 2) from GS injection 1349 (MDC).

**Figure supplement 12.** Distribution of motor neurons (triangles) and premotor interneurons (dots) of 3 TA (yellow) single injections with rabies virus in *Olig2^{Cre/+}; Rosa26^{RΦGT}* mice (MDC).

**Figure supplement 12—source data 1.** Cartesian x-y-z coordinates of interneurons (sheet 1) and motor neurons (sheet 2) from TA injection 700 (MDC).

**Figure supplement 12—source data 2.** Cartesian x-y-z coordinates of interneurons (sheet 1) and motor neurons (sheet 2) from TA injection 721 (MDC).

**Figure supplement 12—source data 3.** Cartesian x-y-z coordinates of interneurons (sheet 1) and motor neurons (sheet 2) from TA injection 1324 (MDC).

**Figure supplement 13.** Distribution of infected neurons (primary infected motor neurons or secondary infected interneurons are not distinguished) of 2 GS (blue) and TA (yellow) double injections and 2 GS single injections of rabies and AAV-Ef1a-B19G in wild-type mice (Salk).

**Figure supplement 13—source data 1.** Cartesian x-y-z coordinates of infected neurons from GS injection 1 (Salk).

**Figure supplement 13—source data 2.** Cartesian x-y-z coordinates of infected neurons from GS injection 2 (Salk).

**Figure supplement 13—source data 3.** Cartesian x-y-z coordinates of infected neurons from GS injection a (Salk).

*Figure 12 continued*

**Figure supplement 13—source data 4.** Cartesian x-y-z coordinates of infected neurons from GS injection b (Salk).

**Figure supplement 13—source data 5.** Cartesian x-y-z coordinates of infected neurons from TA injection 1 (Salk).

**Figure supplement 13—source data 6.** Cartesian x-y-z coordinates of infected neurons from TA injection 2 (Salk).

**Figure supplement 14.** Distribution of infected neurons (primary infected motor neurons or secondary infected interneurons are not distinguished) of 4 GS (blue) single injections of rabies and AAV-CAG-Flex-oG in *Chat^{Cre/+}* mice (Salk).

**Figure supplement 14—source data 1.** Cartesian x-y-z coordinates of infected neurons from GS injection 22a_4 (Salk).

**Figure supplement 14—source data 2.** Cartesian x-y-z coordinates of infected neurons from GS injection 26a_1 (Salk).

**Figure supplement 14—source data 3.** Cartesian x-y-z coordinates of infected neurons from GS injection 26a_2 (Salk).

**Figure supplement 14—source data 4.** Cartesian x-y-z coordinates of infected neurons from GS injection 26a_4 (Salk).

**Figure supplement 15.** Distribution of infected neurons (primary infected motor neurons or secondary infected interneurons are not distinguished) of 3 TA (yellow) single injections of rabies and AAV-CAG-Flex-oG in *Chat^{Cre/+}* mice (Salk).

**Figure supplement 15—source data 1.** Cartesian x-y-z coordinates of infected neurons from TA injection 26_1 (Salk).

**Figure supplement 15—source data 2.** Cartesian x-y-z coordinates of infected neurons from TA injection 26_3 (Salk).

**Figure supplement 15—source data 3.** Cartesian x-y-z coordinates of infected neurons from TA injection 26_4 (Salk).

**Figure supplement 16.** Distribution of infected neurons (primary infected motor neurons or secondary infected interneurons are not distinguished) of 4 GS (blue) and TA (yellow) double injections of PRV-152 and PRV-614 in wild type mice (Salk).

**Figure supplement 16—source data 1.** Cartesian x-y-z coordinates of infected neurons from GS injection 1_1 (Salk).

**Figure supplement 16—source data 2.** Cartesian x-y-z coordinates of infected neurons from GS injection 1_4 (Salk).

**Figure supplement 16—source data 3.** Cartesian x-y-z coordinates of infected neurons from GS injection 2_2 (Salk).

**Figure supplement 16—source data 4.** Cartesian x-y-z coordinates of infected neurons from GS injection 3_3 (Salk).

**Figure supplement 16—source data 5.** Cartesian x-y-z coordinates of infected neurons from TA injection 1_1 (Salk).

**Figure supplement 16—source data 6.** Cartesian x-y-z coordinates of infected neurons from TA injection 1_4 (Salk).

**Figure supplement 16—source data 7.** Cartesian x-y-z coordinates of infected neurons from TA injection 2_2 (Salk).

**Figure supplement 16—source data 8.** Cartesian x-y-z coordinates of infected neurons from TA injection 3_3 (Salk).

of muscle-innervating sensory afferents that have post-synaptic targets predominantly located in the dorsal and medial aspects of the spinal cord (*Zampieri et al., 2014*; *Pimpinella and Zampieri, 2021*). Notably, this observation does not explain why we failed to replicate segregation of flexor and extensor premotor circuits, as we detected overlapping distributions both in the presence (AAV-G) or absence (AAV-flex-oG) of sensory contributions. It is nonetheless interesting to notice that in the original report using AAV-G and ΔG-RabV co-injection in wild type mice, the only condition where flexor-extensor segregation was not reported by the authors is upon elimination of the sensory route by ablation of proprioceptors with diphtheria toxin (*Tripodi et al., 2011*).

It is worth noting that while the original paper used a cytomegalovirus (CMV) promoter to drive G expression (*Tripodi et al., 2011*), we used either a Human elongation factor-1 alpha promoter (EF-1a) or a CMV early enhancer/chicken β actin promoter (CAG). In addition, we used an AAV to RabV ratio of 3:1 as opposed to 1:1. Although these differences may account for higher expression of G and increased efficacy in jumps, they cannot explain the observed lack of segregation. Finally, the major difference in the experiments is the total volume of viral suspension injected intramuscularly. We limited our injections to 1 µl as we measured the volumes of the GS and TA muscles to be around 2 µl in the early post-natal period. Thus, it is possible that the larger volume (5 µl) injected in the other studies might have affected the results, for example by differential infection of proprioceptive afferents innervating the flexor vs extensor muscles, or by infection of cutaneous afferents supplying the overlying skin (*Li et al., 2011*).

## The issue of starter motor neurons

The identity and number of starter cells are the main determinants of reproducibility in rabies tracing experiments and thus represent key parameters for comparing different approaches. For experiments using ΔG-RabV, starter cells are those that are both primarily infected with RabV and express G. In general, for both the AAV and mouse genetics methods discussed here, it is difficult to precisely determine these factors, as neither approach employs expression of a reporter gene

to mark G-expressing cells. Moreover, RabV is known to be toxic to neurons and some primary infected motor neurons are likely to die before analysis (*Reardon et al., 2016*). Because of the well-known topographic organization of neuromuscular maps, muscle identity of infected motor neurons can be inferred by their stereotyped position in the spinal cord (*Romanes, 1964*; *McHanwell and Biscoe, 1981*). Thus, for all the methodologies discussed here, it is only possible to approximate the identity and number of starter motor neurons by surveying the position of RabV-infected motor neurons present at the end of the experiment.

Restriction of starter 'status' to motor neurons connected to a single muscle is determined by two aspects: the specificity of rabies virus injection and the availability of sufficient levels of G protein in the same cells (*Callaway and Luo, 2015*). All the approaches discussed here used intramuscular injections of G-deleted rabies virus (SAD-B19) to selectively infect a motor pool. In this step, sources of variability are represented by (1) specificity of muscle injection and (2) the titre of the rabies virus injected. Muscle injection specificity was routinely checked following injections of adjacent synergist muscles and for all the co-injections of AAV-G and RabV. Rabies leak from antagonist muscles (LG and TA) located in different anatomical compartments on opposite sides of the tibia and fibula would be very unlikely. The titre of the injected rabies virus can affect the efficiency of primary infection: the data presented here show that the RabV titre, while affecting the number of motor neurons and secondary neurons labelled, does not affect the overall distribution of premotor interneurons. The same data indicate that the presence of a small number of 'ectopic' motor neurons (seen only following high titre injections) does not significantly contribute to the tracing results, as the premotor distributions in high and low titre experiments are not different. Furthermore, these 'ectopic' motor neurons likely represent recurrently connected presynaptic motor neurons (*Bhumbra and Beato, 2018*); therefore any labelling originating from them would represent a much less frequent disynaptic transfer event.

In the short term, the introduction of a reporter system to label G-expressing neurons, as routinely done in many rabies experiments, combined with the use of non-toxic rabies variants that would prevent motor neuron death (*Reardon et al., 2016*; *Ciabatti et al., 2017*; *Chatterjee et al., 2018*) will help resolve potential confusion about the identity and number of starter cells. Such tools could be used in both the AAV and the mouse genetic approaches. In addition, the ability to precisely restrict the selection of starter motor neurons either by the introduction of more specific Cre lines (e.g. *Koronfel et al., 2021*) or through the use of novel intersectional strategies could improve premotor tracing experiments. Finally, tracing from single motor neurons using delivery of DNA for G and TVA expression via patch clamp is a precise way to generate specific starter cells (*Marshel et al., 2010*; *Rancz et al., 2011*). This approach, followed by intraspinal injection of EnvA-pseudotyped ΔG-RabV, would ensure infection and pre-synaptic tracing from only selected neurons. This method would have the added value of directly showing whether functionally distinct motor neurons within a pool receive differentially distributed presynaptic input, but it would rely on being able to perform patch clamp recordings from motor neurons in vivo, followed by a sufficient survival time to allow for sufficient viral expression, a feat that to our knowledge has not been attempted and might not even be possible.

## Functional implications

While it has been proposed that spatial segregation of premotor interneurons provides an anatomical substrate for labelled line inputs from proprioceptive afferents to motor neurons (*Tripodi et al., 2011*), our data do not support such a model. Proprioceptive afferents projecting to the intermediate and dorsal spinal cord relay many types of information (e.g. changes in muscle length, muscle length itself, force, joint position), and form synapses with both excitatory and inhibitory interneurons that process and convey these data to flexor and extensor motor neurons to precisely regulate patterns of contraction. It is clear that, at least in the case of motor neurons, position plays an important role in the specificity of afferent inputs received (*Sürmeli et al., 2011*), and it is reasonable to think that it might also be the case for interneurons. Indeed, it has been shown for V1 inhibitory interneurons that cell body positioning constrains wiring from proprioceptive afferents (*Bikoff et al., 2016*). However, the location of V1 subtypes does not seem to influence their output connectivity to motor neurons, as indicated by the case of Sp8 +V1 interneurons that are located in a medial position in the intermediate spinal cord, in about the same location

**Table 2.** Summary of pros and cons of each described method.

| Method | Pros | Cons | Outcome | Reference |
|---|---|---|---|---|
| Muscle injection of AAV-G (serotype 2.6)+RabV (*Figure 1A*) | Avoids the possibility of retrograde disynaptic transfer from second order motor neurons due to restriction of G expression to targeted motor neurons | The labelled premotor population could be contaminated by anterogradely labelled neurons from primary sensory neurons. | Flexor- extensor segregation No flexor-extensor segregation | (*Tripodi et al., 2011*) Present study |
| | Avoids the possibility of retrograde disynaptic transfer from premotor spinal interneurons. | | | |
| Muscle injection of AAV-flex-G (serotype 2.6)+RabV in *Chat^Cre/+* mice (*Figure 1B*) | Avoids the possibility of retrograde disynaptic transfer from second order motor neurons due to restriction of G expression to targeted motor neurons | | Flexor- extensor segregation No flexor-extensor segregation | (*Wang et al., 2017*) Present study |
| | Avoids the possibility of retrograde disynaptic transfer from premotor spinal interneurons. | | | |
| | Avoids potential anterograde sensory contamination. | Conditional expression of G may be inefficient | | |
| Central injection of AAV-flex-G (serotype 2.9) in *Chat^Cre/+* mice followed by muscle injection of RabV, in adults (*Figure 1C*) | Limits the issue of potential disynaptic transfer from cholinergic interneurons | Potential for disynaptic transfer from cholinergic premotor interneurons, transsynaptically labelled motor neurons and mis-targeted primary motor neurons | Flexor- extensor segregation | *Takeoka and Arber, 2019* |
| | Avoids potential anterograde tracing from sensory neurons | | | |
| Genetically driven expression of G in *Chat^Cre/+* or *Olig2^Cre/+* mice + muscle RabV injection in neonates (*Figure 1D and E*) | Avoids potential anterograde tracing from sensory neurons | Potential for disynaptic transfer from premotor spinal interneurons, transsynaptically labelled motor neurons and mis-targeted primary motor neurons. | No flexor-extensor segregation | Present study |
| | Ensures homogenous expression of G in all motor neurons | | | |
| Muscle injection of PRV-Bartha with strictly timed fixation of tissue (*Figure 1F*) | High efficiency in transsynaptic transmission. Not reliant on viral recombination. | Timed fixation does not guarantee that transsynaptic jumps occur only up to the second order | No flexor-extensor segregation | Present study |

described for premotor interneuron serving extensor motor pools (*Tripodi et al., 2011*) but send uniform synaptic output to both flexor and extensor motor pools (*Bikoff et al., 2016*). Our data are consistent with the idea that interneuron position plays an important role in controlling sensory input connectivity, but not their output connectivity to motor neurons. That is, positional organization has been shown to represent a developmental strategy that facilitates wiring onto spatially segregated populations of neurons, rather than determining output connectivity, where neuronal position does not seem to have a major role. It is important to stress that both the original study showing segregation of premotor interneurons (*Tripodi et al., 2011*) and ours, showing lack of it, are performed on neonatal animals, because the variant of rabies used in both studies cannot cross the mature neuromuscular junction and infect adult motor neurons (*Stepien et al., 2010*, but see *Takeoka and Arber, 2019*). We have no evidence that the output connectivity of premotor interneurons that we describe here is conserved throughout development, because the initial wiring of circuits is later refined as the mice develop, adapt to their environment and learn new motor tasks. Nonetheless, we cannot attribute the observed lack of segregation to developmental factors, since we have used animals in the same age range as in the original study.

## Conclusion

In conclusion, despite using five different methods, we have not been able to find evidence that spinal premotor interneurons innervating flexor vs extensor motor neurons are segregated. It is important to stress that none of the methods discussed here is completely exempt from potential problems (*Table 2*). However, full appreciation of the strengths and weaknesses of each approach can guide both the choice of method for mapping premotor circuits and the interpretation of the results obtained.

# Methods

**Key resources table**

| Reagent type (species) or resource | Designation | Source or reference | Identifiers | Additional information |
|---|---|---|---|---|
| Strain, strain background (Rabies virus) | ΔG-Rab-eGFP | Gift from M. Tripodi lab, LMCB Cambridge | | |
| Strain, strain background (Rabies virus) | ΔG-Rab-mCherry | Gift from M. Tripodi lab, LMCB Cambridge | | |
| Strain, strain background (Adeno associated virus) | AAV6-Ef1a-B19G | Produced by Applied Viromics (USA) | | |
| Strain, strain background (Adeno associated virus) | AAV6-CAG-Flex-oG | Produced at the Salk GT3 virus core facility | | |
| Strain, strain background (*M. musculus, Chat$^{Cre/+}$*) | ChAT-IRES-Cre | Jackson laboratory | IMSR Cat# JAX:006410; RRID:IMSR_JAX:006410 | allele symbol: Chat$^{tm2(cre)Lowl}$; maintained on a C57BL6/J background |
| Strain, strain background (*M. musculus, Olig2$^{Cre/+}$*) | Olig2-Cre | Jackson laboratory | IMSR Cat# JAX:025567; RRID:IMSR_JAX:025567 | allele symbol: B6.129-Olig2$^{tm1.1(cre)Wdr}$/J maintained on a C57BL6/J background |
| Strain, strain background (*M. musculus, Rosa26$^{RΦGT}$*) | RΦGT | Jackson laboratory | IMSR Cat# JAX:024708; RRID:IMSR_JAX:024708 | allele symbol: Gt(ROSA)26Sortm1(CAG-RABVgp4,-TVA)Arenk; maintained on a C57BL6/J background |
| Strain, strain background (*M. musculus, Slc6A5$^{eGFP}$*) | *Slc6A5$^{eGFP}$* | Gift from H. Zeilhofer lab, University of Zurich | IMSR Cat# RBRC04708; RRID:IMSR_RBRC04708 | allele symbol: Tg(Slc6a5-EGFP)1Uze; maintained on a C57BL6/J background |
| Strain, strain background (*M. musculus, Rosa26$^{RCL-tdTom}$*) | Ai9(RCL-tdT) | Jackson laboratory | IMSR Cat# JAX:007909; RRID:IMSR_JAX:007909 | allele symbol: Gt(ROSA)26Sortm9(CAG-tdTomato)Hze/J; maintained on a C57BL6/J background |
| Cell line (*Homo-sapiens*, female) | HEK293t/17 | Gift from M. Tripodi lab, LMCB Cambridge | RRID:CVCL_1926 | ATCC, cat. no. CRL-1126 |
| Cell line (*Mesocricetus auratus*, male) | BHK-21 | Gift from M. Tripodi lab, LMCB Cambridge | RRID: CVCL_1915 | ATCC # CCL-10 |
| Cell line (*Mesocricetus auratus*, male) | BHK-G | Gift from M. Tripodi lab, LMCB Cambridge | RRID:CVCL_1915 | Derived from ATCC # CCL-10 |
| Antibody (UCL) | Anti-ChAT (Goat polyclonal) | Millipore | Cat# AB144P; RRID:AB_2079751 | IF (1:100) |
| Antibody (UCL) | Anti-mCherry (Chicken polyclonal) | Abcam | Cat# ab205402; RRID:AB_2722769 | IF (1:2500) |
| Antibody (UCL) | Anti-GFP (Rabbit polyclonal) | Abcam | Cat# ab290; RRID:AB_303395 | IF (1:2500) |
| Antibody (UCL) | Anti-vGluT2 (Guinea pig polyclonal) | Millipore | Cat# AB2251-I; RRID:AB_2665454 | IF (1:2500) |
| Antibody (UCL) | Anti-Isl1 (Guinea pig polyclonal) | Gift from T. Jessell lab, Columbia University, New York | | IF (1:7500) |
| Antibody (UCL) | Anti-guinea pig IgG H&L Alexa Fluor 647 (Donkey polyclonal) | Millipore | Cat# AP193SA6; RRID:AB_2340477 | IF (1:700) |
| Antibody (UCL) | Anti-Goat IgG H&L Alexa Fluor 405 (Donkey polyclonal preadsorbed) | Abcam | Abcam Cat# AB175665; RRID:AB_2636888 | IF (1:200) |

*Continued on next page*

*Continued*

| Reagent type (species) or resource | Designation | Source or reference | Identifiers | Additional information |
|---|---|---|---|---|
| Antibody (UCL) | Anti-Rabbit IgG H&L Alexa Fluor488 (Donkey polyclonal Highly Cross-Adsorbed) | Thermo Fisher Scientific | Cat# A-21206; RRID:AB_2535792 | IF (1:1000) |
| Antibody (UCL) | Anti-Chicken IgY (IgG) H&L Cy3-AffiniPure (Donkey polyclonal) | Jackson ImmunoResearch Labs | Cat# 703-165-155; RRID:AB_2340363 | IF (1:1000) |
| Antibody (Glasgow University) | Anti-GFP (chicken polyclonal) | Abcam | Cat# Ab13970 RRID:AB_300798 | IF (1:1000) |
| Antibody (Glasgow University) | Anti-mCherry (rabbit polyclonal) | Abcam | Cat# Ab167453 RRID:AB_2571870 | IF (1:2000) |
| Antibody (Glasgow University) | Anti-chicken IgY H&L Alexa Fluor488 (Donkey polyclonal) | Jackson ImmunoResearch Labs | Cat# 703-545-155; RRID:AB_2340363 | IF (1:500) |
| Antibody (MDC) | Anti-ChAT (rabbit polyclonal) | Abcam | Cat# Ab2750952 RRID:AB_2750952 | IF (1:16,000) |
| Antibody (MDC) | Anti-Rabbit IgG H&L Alexa Fluor488 (Donkey polyclonal) | Thermo Fisher Scientific | Cat# A-21206; RRID:AB_2535792 | IF (1:1000) |
| Antibody (Salk) | Anti-GFP (goat polyclonal) | Rockland | Cat#600-101-215; RRID:AB_218182 | IF (1:1000) |
| Antibody (Salk) | Anti-RFP (rabbit polyclonal) | Rockland | Cat#600-401-379; RRID:AB_2209751 | IF (1:1000) |
| Antibody (Salk) | Anti-goat IgY H&L Alexa Fluor488 (Donkey polyclonal) | Invitrogen | Cat#A11055; RRID:AB_2534102 | IF (1:1000) |
| Antibody (Salk) | Anti-rabbit IgY H&L Alexa Fluor555 (Donkey polyclonal) | Invitrogen | Cat#A32794; RRID:AB_2762834 | IF (1:1000) |
| Chemical compound, drug | Mowiol 4–88 | Sigma Aldrich | Cat# 81381–250 G | |
| Software, algorithm | ZEN Digital Imaging for Light Microscopy: Zen Blue 2.3 | Carl Zeiss light microscopy imaging systems | RRID:SCR_013672 | |
| Software, algorithm | Imaris 9.1 | Bitplane | RRID:SCR_007370 | |
| Software, algorithm | Adobe illustrator version CC 2019 | Adobe | RRID:SCR_010279 | |
| Software, algorithm | Matlab version 2021b | Mathworks | RRID:SCR_001622 | |

## Experimental settings

The experiments were performed and analysed across 4 different laboratories. The injections labelled as UoG (University of Glasgow) were performed in the Beato lab at UCL (three different operators) and processed in Glasgow (Maxwell and Todd labs) using RabV produced in the Beato lab. The injections labelled as UCL were performed in the Brownstone lab (two different operators) using RabV produced in the Brownstone lab and the tissue was processed at UCL (Brownstone lab). The injections in *Olig2*^Cre/+ mice were performed and analysed at the Max Delbrück Center (MDC), with locally produced RabV. The injections of AAV-G and AAV-G-Flex in wild type and *Chat*^Cre/+ mice respectively were performed and analysed at the Salk Institute. The AAV-G construct was produced from Applied Viromics, while the AAV-G-Flex was produced at the Salk GT3 Core and the RabV was obtained from

Janelia Farm. Experiments with PRV-Bartha rabies strain were performed and analysed at the Salk Institute. PRV-Bartha was obtained from NIH CNNV.

## Animal experimentation ethical approval

All experiments at UCL were carried out according to the Animals (Scientific Procedures) Act UK (1986) and were approved by the UCL AWERB committee under project licence number 70/9098. All experiments performed at the MDC were carried out in compliance with the German Animal Welfare Act and approved by the Regional Office for Health and Social Affairs Berlin (LAGeSo). All experiments performed at the Salk Institute were conducted in accordance with IACUC and AAALAC guidelines of the Salk Institute for Biological Studies. Experimental design followed the ARRIVE guidelines.

## Mouse strains

Homozygous *Chat*$^{Cre/Cre}$ mice (*Rossi et al., 2011*, Jackson lab, stock #006410) or heterozygous *Olig2*$^{Cre/+}$ mice (*Dessaud et al., 2007*) were crossed with homozygous *Rosa26*$^{RΦGT}$ mice (Jackson Lab, stock #024708), to generate *Chat*$^{Cre/+}$; *Rosa26*$^{RΦGT}$ or *Olig2*$^{Cre/+}$; *Rosa26*$^{RΦGT}$ mice (*Takatoh et al., 2013*; *Skarlatou et al., 2020*) that were used for rabies tracing experiments. For experiments aimed at distinguishing excitatory and inhibitory populations of premotor interneurons, we crossed homozygous *Chat*$^{Cre/Cre}$ mice with heterozygous *slc6A5*$^{eGFP/+}$ mice (termed *GlyT2-eGFP* here, a gift from Prof. Zeilhofer, University of Zurich, *Zeilhofer et al., 2005*) and their double-positive offspring were mated with the homozygous *Rosa26*$^{RΦGT}$ mice. AAV-G-Flex constructs were injected in heterozygous *Chat*$^{Cre/+}$ mice.

To quantify possible 'leak' of Cre expression in the spinal cord, we crossed *Chat*$^{Cre/+}$ and *Rosa26*$^{RCL-tdTom}$ (Ai9, Jax stock #007909) mice (13 sections from 3 mice), and found tdTom expression in cholinergic motor neurons and interneurons as expected, as well as in some non-cholinergic neurons distributed in intermediate (10%, 70/690) and dorsal (9%, 62/690) laminae, with the remaining 35 located ventrally (*Figure 13*) indicating that ectopic expression of Cre in ChAT-negative neurons is minimal but not nil, and mostly confined to superficial dorsal laminae, an area that is devoid of premotor interneurons, indicating that while double jumps from 'leaky' Cre expressing interneurons remains a possibility, its extent would not be sufficient to alter the distribution of labelled interneurons. The possible 'leak' of G and the avian receptor protein (TVA) expression in *Rosa26*$^{RΦGT}$ mice was then tested by injecting EnvA-ΔG-Rab-eGFP, produced according to standard protocols (*Osakada and Callaway, 2013*) to a titre of $1×10^9$ IU/ml. Lack of contamination from non-pseudotyped virus was confirmed by infecting HEK cells at high (up to 20) multiplicity of infection. Three *Rosa26*$^{RΦGT}$ heterozygous mice were injected in the lateral gastrocnemius muscle at P1 and fixed 9 days post injections. The tissue was cut as described below, but along the horizontal plane in 60 μm sections, in order to isolate the dorsal motor column. Following immunoreaction for eGFP, in each of the three cords, we found a maximum of three labelled motor neurons (1, 1 and 3 motor neurons in n=3 animals) but no interneurons labelled, indicating some leakage in the expression of the TVA-IRES-glycoprotein cassette from the *Rosa26*$^{RΦGT}$ mice, but insufficient G expression to support transsynaptic jumps (*Figure 13—figure supplement 1*).

## Virus production

Rabies virus used in experiments performed at UCL, was obtained from in house stocks of a variant of the SAD-B19 rabies strain where the sequence coding for the glycoprotein was replaced by the sequence for either eGFP or mCherry (*Wickersham et al., 2007*). Virus was produced at high concentration using the protocol described in *Osakada and Callaway, 2013*: baby hamster kidney cells stably expressing the rabies glycoprotein (BHK-G, kindly provided by Dr. Tripodi LMCB, Cambridge) were thawed and plated in standard Dulbecco modified medium, supplemented with 10% fetal bovine serum (FBS), incubated at 37 °C with 5% $CO_2$ and split until ~70% confluence was obtained in 5 dishes (10 ml medium each). The cell lines were not used as an experimental system, but only for virus production. The cells were then inoculated at 0.2–0.3 multiplicity of infection with either the ΔG-RabV-eGFP or the ΔG-RabV-mCherry (initial samples kindly provided by Prof. Arber and Dr. Tripodi). Cells were incubated for 6 hr at 35 °C and 3% $CO_2$ and then split 1–4 into 20 dishes (10 ml) with 10% FBS medium and kept at 37 °C and 5% $CO_2$ for 12–24 hr, until ~70% confluent and medium was then replaced with 2% FBS medium and cells incubated at 35 °C and 3% $CO_2$ for virus production. The supernatant was collected after ~3 days, new medium was added for another round of production and supernatant

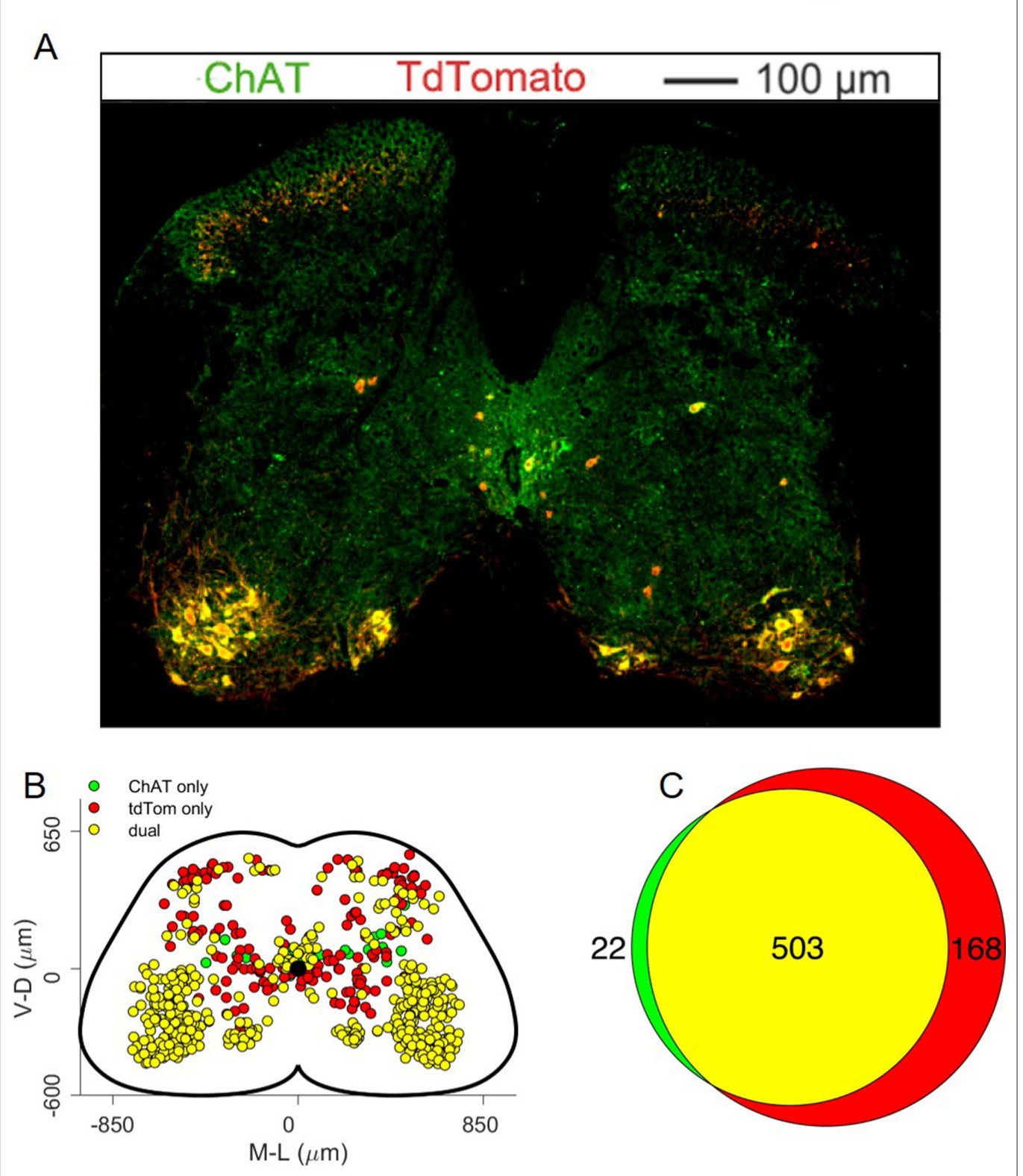

**Figure 13.** Analysis of ectopic Cre expression in *Chat^Cre/+^; Rosa26^RCL-tdTom^* mice. (**A**) Representative lumbar section stained with antibodies against ChAT (green) and tdTom (red) and (**B**) map of neurons labelled with both or one of the two antibodies in all the 13 analysed sections from 3 mice, showing that some of the tdTom positive neurons do not express ChAT, indicating either a developmental downregulation of ChAT expression or a modest leak in the Cre expression. (**C**) Venn diagram showing the overall number of mapped neurons.

*Figure 13 continued on next page*

*Figure 13 continued*

The online version of this article includes the following source data and figure supplement(s) for figure 13:

**Source data 1.** Cartesian x-y coordinates of neurons labelled with ChAT and/or TdTomato in the lower lumber spinal cord of *ChAT$^{Cre}$; Rosa26$^{RCL-tdTom}$* mice.

**Figure supplement 1.** Two examples of a longitudinal section of two different spinal cords from a heterozygous *Rosa26$^{RΦGT}$* mouse injected in the LG with EnvA-ΔG-Rab-EGFP, showing a small number of infected motor neurons, but no evidence of transsynaptic jumps, indicating ectopic expression of the TVA receptor, but not of the rabies glycoprotein.

**Figure supplement 2.** Schematic of a section of the spinal cord, indicating the reference points used for normalization.

was filtered (0.45 µm filter) and centrifuged for 2 hr at 19,400 rpm (SW28 Beckman rotor). The pellets were suspended in phosphate buffer saline (PBS), dispersed and collected in a single tube and further centrifuged for 4 hr at 21,000 rpm in a 20% sucrose gradient (SW55 Beckman rotor). The resulting pellet was suspended in 100 µl PBS and the virus was aliquoted (5–10 µl) and stored in a –80° freezer. The viral titre of each round of production was measured by serial 10-fold dilution of three different aliquots using standard protocols (*Osakada and Callaway, 2013*). As a measure of the quality of the virus production, the virus titre was routinely measured in multiple aliquotes (n=3) for each round of production. For each injection, the virus titre is reported in *Table 1*. In a subset of experiments, we diluted the virus 10-fold in order to limit the number of starter cells. Rabies virus used for experiments performed at MDC was produced as previously described (*Skarlatou et al., 2020*). For the premotor labelling experiments mediated through co-injection of AAV-Flex-G and delta-G-RabV at the Salk Institute, pSADΔG-mCherry and pSADΔG-eGFP were obtained from Janelia Viral Tools facility. AAV6-Ef1a-B19G (1x10$^{13}$ GC/mL) was generated at Applied Viriomics, while AAV6-CAG-Flex-oG was generated at the GT3 core at the Salk Institute. For retrograde PRV-Bartha experiments, PRV-152 (GFP) and PRV-614 (mRFP) were obtained from the Center for Neuroanatomy with Neurotropic Viruses (CNNV).

## Intramuscular injection

Neonatal pups (P1-P2) were anaesthetized using isoflurane inhalation and an incision was made on the skin to expose the belly of the targeted muscle, either lateral or medial gastrocnemius (LG, MG, whose primary role is that of ankle extensors, but also contribute to knee flexion), tibialis anterior (TA) or peroneus longus (PL, ankle flexors). The virus was injected intramuscularly using a 5 µl Hamilton syringe (model 7652–01) fixed to a manual Narishige micromanipulator (M-3333) and loaded with a bevelled glass pipette of inner diameter 50–70 µm. The volume injected was 1 µl, compatible with the estimated volume of the muscles at this age (~2 µl), in order to minimize the risk of leaks to adjacent muscles. Viral batches of similar titres were slowly (>1 min) injected, the skin incision was sutured with Vicryl 8–0 (Ethicon, USA) and the pups received a subcutaneous injection of carprofen (10%) for pain management. Mice were closely monitored for the next 24 hr for signs of movement impairment and were perfused under terminal general anaesthesia 8–9 days after injection.

In order to compare directly the distributions of flexor and extensor associated premotor interneurons and avoid confounding factors in the coordinate representations of these interneurons across different spinal cords, we performed double injections of ΔG-RabV in the same animal, using ΔG-Rab-eGFP and ΔG-Rab-mCherry injected in pairs of antagonist muscles, lateral gastrocnemius and tibialis anterior. For comparison, we also performed double injections in pairs of synergist muscles: LG and MG or TA and PL. Due to the close proximity of these pairs of muscles, before cutting the spinal tissue for immunohistochemistry, we dissected the injected leg and confirmed that there was no contamination of virus across the injected muscles or in adjacent muscles below or above the knee. To exclude confounding factors in our observed premotor interneuron distributions due to systematic viral interference (*Ohara et al., 2009a*; *Ohara et al., 2009b*), in a subset of experiments single injections of ΔG-Rab-mCherry (4 LG, 4 MG, 2 TA and 4 PL) were performed in the progeny of either *Chat$^{Cre/+}$; Rosa26$^{RΦGT}$* or *GlyT2-eGFP;Chat$^{Cre/+}$; Rosa26$^{RΦGT}$* mice.

For experiments performed at MDC, intramuscular injections were done as previously described (*Skarlatou et al., 2020*). Briefly, P4 animals were anesthetized with isoflurane and a small incision in the skin was made to reveal either the gastrocnemius (GS, 4 experiments, no attempts were made at selective targeting of the two heads of the GS muscle) or the tibialis anterior (3 experiments) muscles. A volume of 1.5 µl of ΔG-RabV-mCherry was injected *in Olig2$^{Cre/+}$; Rosa26$^{RΦGT}$* mice using a

glass capillary. Animals were perfused at P10, 6 days after injection in order to minimize the chance of disynaptic transfer.

For wild type ΔG-RabV premotor labelling experiments performed at the Salk Institute, P2 pups were anesthetised on ice and 1 µl of a 3:1 mixture of AAV-Ef1a-B19G and ΔG-Rab-GFP or ΔG-Rab-mCherry was injected through the skin into the GS or TA muscles of the same animal, using a 10 µl Hamilton syringe with 30-gauge metal needle. Animals were fixed by perfusion under terminal general anaesthesia, as described below, 10 days post injection. For *Chat*<sup>Cre/+</sup> experiments, a 3:1 mixture of AAV-CAG-Flex-oG (Addgene #74292) and ΔG-Rab-GFP was injected into either the GS or TA muscles of P1-P2 pups. Anaesthesia, injection volume, and syringe used were the same as above. Animals were fixed by perfusion under terminal general anaesthesia, as described below, 7 days post-injection. Hindlimbs were visualized under a stereoscope to confirm correct targeting of muscles.

For PRV experiments, intramuscular injections of PRV were performed in P11 pups under 2–3% isoflurane anaesthesia. Briefly, a small incision was made in the skin to expose the TA and GS and 0.5 µl of PRV-152 or PRV-614 was delivered to either muscle, in the same animal, using a glass needle and picospritzer. Animals were perfused 48 hr post injection. Hindlimbs were visualized under a stereoscope and sectioned to confirmed correct targeting of muscles.

## Tissue collection and immunohistochemistry

Under ketamine/xylazine terminal anaesthesia (i.p. 80 mg/kg and 10 mg/kg respectively), mice were intracardially perfused with phosphate buffer solution (0.1 M PBS), followed by 4% paraformaldehyde in PBS. The spinal cords were dissected and post-fixed for 2 hr at 4 °C, cryoprotected overnight at 4 °C in 30% PBS sucrose solution and embedded in OCT (optimal cutting temperature, Tissue-Tek, #4583) compound.

For UCL experiments, injections were all conducted at UCL, whereas the immunohistochemistry and imaging were conducted on different animals independently in two different laboratories (Maxwell at Glasgow University and Beato/Brownstone at UCL). For tissue processing performed at UCL, lumbar spinal cords were cut (30 µm thickness) in series in the transverse plane with a cryostat (Bright Instruments, UK) mounted onto charged glass slides (VWR, #631–0108), and stored at –20 °C. Sections were incubated for 36 hours at 4 °C with primary antibodies and overnight at 4 °C with secondary antibodies in PBS containing 0.3 M NaCl, 0.2% Triton 100 X (Sigma, T9284-500ml), 7% donkey normal serum (Sigma, D9663-10ml). The primary antibodies used were: guinea pig anti-Isl1 (1:7500, from Dr. T Jessell, Columbia University, New York), goat anti choline acetyl transferase (ChAT, 1:100, Millipore, AB144P), rabbit anti-GFP (1:2500, Abcam. Ab290), chicken anti-mCherry (1:2500, Abcam, Ab205402). The secondary antibodies were: donkey anti-guinea pig Alexa 647 (1:700, Millipore, AP193SA6), donkey anti-goat preabsorbed Alexa 405 (1:200, Abcam, ab175665), donkey anti-rabbit Alexa 488 (1:1000, Thermofisher, A21206), and donkey anti-chicken Cy3 (1:1000, Jackson Immunoresearch, #703-165-155). The slides were mounted in Mowiol (Sigma, 81381–250 G) and cover-slipped (VWR, #631–0147) for imaging.

At Glasgow University, the spinal cords were sectioned using a Leica VT1000 vibratome (thickness 60 µm) and incubated in 50% ethanol for 30 min. Primary antibodies used were: chicken anti-GFP (1:1000, Abcam, Ab13970), rabbit anti-mCherry (1:2000, Abcam, Ab167453) and goat anti-ChAT (1:100, Abcam, AB254118). The secondary antibodies were: donkey anti-chicken A488 (1:500, Jackson Immunoresearch, 703-545-155), and donkey anti-rabbit Rhodamine red (1:100, Jackson Immunoresearch, 711-295-152). Sections were mounted in Vectashield (Vector Laboratories, Peterborough, UK) and coverslipped.

For the experiments performed at MDC, spinal cords were processed as previously described (*Skarlatou et al., 2020*). Briefly, animals were anesthetized by intraperitoneal injection of ketamine /xylazine mix and transcardially perfused with ice-cold PBS (until the liver was cleared of blood), followed by freshly made ice-cold 4% PFA. Spinal cords were dissected and post-fixed for 90 min with 4% PFA on ice. Consecutive 40 µm spinal cord cryosections including the caudal thoracic and lumbar spinal regions were obtained using a Leica cryostat and incubated overnight at 4 °C with rabbit anti-ChAT 1:16000 (*Sürmeli et al., 2011* RRID:AB_2750952) followed by 1 hr incubation at room temperature with secondary antibody (Alexa-Fluor 488, 1:1000). Slides were mounted in Vectashield.

For PRV and ΔG-RabV experiments performed at the Salk Institute, animals were transcardially perfused with ice cold 1 x PBS followed by ice cold 4% PFA (Electron Microscopy Sciences,

100504–858). Spinal cords were dissected and post-fixed in 4% PFA for 2 hr at 4 °C. After the post-fixation, samples were washed with 1 x PBS for 10 min followed by a 1–3 day incubation in 30% sucrose for cryoprotection (4 °C). The lumbar spinal cord was then embedded in Tissue-Tek OCT (Sakura 4583) for cryosectioning onto glass slides (Fisherbrand Superfrost slides 12-550-15) in the transverse plane. The section thickness was 60 µm for delta-G-RabV experiments and 30 µm for PRV experiments. Immunohistochemistry was performed by incubating tissue with primary antibodies overnight at 4 °C and secondary antibodies for 2 hr at room temperature, in a 0.3% Triton-X and 20% donkey serum blocking buffer. After staining slides were mounted with a glass coverslip and Fluoromount-G (00-49-58-02). The antibodies used for this portion of the study were: goat anti-GFP (1:1000, Rockland 600-101-215), rabbit anti RFP (1:1000, Rockland 600-401-379), donkey anti-goat Alexa 488 (1:1000, Invitrogen A11055), donkey anti-rabbit-Alexa 555 (1:1000, Invitrogen A32794).

## Confocal imaging and analysis

For UCL experiments, confocal images were acquired using a Zeiss LSM800 confocal microscope with a 20 x (0.8 NA) air objective and tile advanced set up function of ZEN Blue 2.3 software for imaging of the entire slice. The tiles were stitched using ZEN Blue software and cell detection was performed using Imaris (version up to 9.1, Bitplane) software. Cell counts were manually performed on every other section, in order to minimize the risk of counting the same cell twice in two consecutive sections.

For experiments performed at Glasgow University, the images were acquired using a Zeiss LSM710, with a 20 x (0.8 NA) air objective and cells were counted manually using Neurolucida. Only a subset of sections was analysed (1 in every 8 consecutive sections), thus accounting for approximately 2 sections for every spinal segment.

For experiments performed at the MDC, confocal images were acquired using a Zeiss LSM800 confocal microscope. Regions of interest corresponding to each section and consisting of 8 tiles were imaged with a 10 x (0.3 NA) air objective. The tiles were subsequently stitched using ZEN 2.3 Software. Acquisition and processing were performed immediately after immunohistochemistry where possible, to obtain the best possible signal. The resulting images were used for three-dimensional positional analysis as previously described (*Skarlatou et al., 2020*).

For experiments performed at the Salk Institute, fluorescent images were acquire using an Olympus VS-120 virtual slide scanner microscope with a 10 x objective (0.3 NA). For ΔG-RabV experiments cells counts were performed on every section. For PRV experiments cell counts were performed on every 8[th] section. Both samples were analyzed in MATLAB using a custom script for cell detection.

A consistent system of coordinates was established using the central canal as origin of the x-y plane. The y-axis was defined as parallel to the dorso-ventral axis, with positive values towards the dorsal side and the x-axis was determined by the mediolateral direction, with positive values on the side of injection. For UCL and University of Glasgow experiments the L4-L5 border in the z direction was determined during the slicing procedure and its location was confirmed post-processing by identifying the slices with the widest mediolateral width. For MDC and Salk Institute experiments, the border between T13 and L1 was chosen as the starting point for slicing. The z coordinates obtained in different labs were subsequently aligned using the widest section as a point of reference for the border between L4 and L5 segments. For both Neurolucida and Imaris data files, in order to account for the different shapes of sections throughout the lumbar cord and deformation of individual sections, normalization of coordinates was performed independently for each quadrant using as reference points those indicated in *Figure 13—figure supplement 2*: the x dimension was normalized to the outer edge of the white matter at the level of the central canal, while the y dimension was normalized for each quadrant using the outermost points of the white matter for both dorsal and ventral horns. The resulting cylindrical reconstruction of the spinal cord was then scaled to the idealized spinal cord size (1700 µm in the mediolateral direction and 900 µm in the dorsoventral direction) for illustrational purposes. All coordinate transformations were performed using a custom script in MATLAB, adapted to read both Neurolucida and Imaris file formats. For experiments performed at UCL and MDC, infected motor neurons were identified by co-localization of either Isl1 or ChAT, and the presence of the reporter fluorescent protein (eGFP or mCherry) expressed after rabies infection. For experiments performed at Glasgow University, sections were not reacted for ChAT, but infected motor neurons were identified by size and location. For experiments analysed at the Salk Institute, no attempt was made to identify infected motor neurons, therefore, the resulting spatial distributions

do not distinguish between primary and secondary infected cells. Since infected motor neurons are located in the ventral horn, the inclusion of motor neurons in the overall distributions does not affect our analysis of the mediolateral segregation in the dorsal horn. Distributions of infected interneurons were calculated using a Gaussian convolution with a kernel size evaluated from the standard deviation of the original data (*Bhumbra and Dyball, 2010*).

Gaussian convolutions were calculated splitting the transverse, normalized, spinal cord profile into ipsi- and contra-lateral, and dorsal and ventral halves, with the corresponding distributions shown surrounding the transverse spinal cord maps. Areas under the top-bottom or left-right distributions of each label sum to 1. Correlations across individual experiments were calculated from the x-y coordinates projected along the rostrocaudal axis by computing a density matrix $\rho_n(x_i, y_i)$ for each experiment $n$ and evaluating the correlation coefficient $r_{nm}$ between experiments $n$ and $m$ using the formula

$$r_{nm} = \frac{\sum_i \sum_j \left( \rho_n(x_i, x_j) - median(\rho_n) \right) \left( \rho_m(x_i, x_j) - median(\rho_m) \right)}{\sqrt{\left( \sum_i \sum_j \left( \rho_n(x_i, x_j) - median(\rho_n) \right)^2 \right) \left( \sum_i \sum_j \left( \rho_m(x_i, x_j) - median(\rho_m) \right)^2 \right)}}$$

Correlations across groups of experiments were calculated using the same method, after pooling all the experiments corresponding to one muscle and one injection technique. We decided not to base our analysis on p-values standard hypothesis testing for two reasons: first, the p-value obtained for any statistical test, be it parametric or not, depends strongly on the sample size (*Gómez-de-Mariscal et al., 2021*). Given our large sample size (coordinates for several thousands of neurons), this would lead to an unacceptable rate of false positive. Indeed, within our dataset, we obtain very low (<10⁻⁴) p-values for distributions whose medians where only a few μm apart. Since the spatial resolution of our cell location procedure is no higher than the diameter of the infected cells, such results cannot be considered meaningful. Second, our data have an intrinsically nested structure, therefore, hypothesis testing on pooled experiments would lead to an error caused by ignoring the hierarchical nature of our data. Therefore, when comparing different methods of injections where distributions from multiple animals were analysed, we first performed a hierarchical bootstrap procedure (described in detail in *Saravanan et al., 2020*): briefly if n animals were injected with a given technique, we resampled (with replacement) the animals (first level bootstrap) and then within each animal, we resampled (with replacement) the coordinates distribution (second level bootstrap). This procedure was repeated 5000 times, giving rise to 5000 resampled experiments. In order to compare such experiments, we calculated the Hedges' G coefficient (*Hedges, 1981*) across pairs randomly selected within the 5000 replicas and obtained a distribution of Hedges' G The medians of such distribution and inter quartile range (IQR) are reported for each pair of techniques or muscles we compared. When individual experiments are compared, the Hedges' G is reported for each pair of individual experiments. Rather than providing a dichotomous decision, absolute values of effect sizes can be classified as no effect (0–0.19), small (0.2–0.49), medium (0.5–0.79) and large (0.8 and above, *Hedges, 1981*). Sample sizes and median values are reported individually for each experiment in *Table 1*. All data processing was performed in MATLAB, using custom written software. The paper can be downloaded in executable format as a MATLAB live script from https://github.com/marcobeato/Spinal_premotor_interneurons_controlling_antagonistic_muscles_are_spatially_intermingled; *Beato, 2022*, where all the data are available An R version of the executable paper is available at https://mybinder.org/v2/gh/rronzano/Spinal_premotor_interneurons_controlling_antagonistic_muscles_are_spatially_intermingled.git/HEAD?urlpath=rstudio and https://github.com/rronzano/Spinal_premotor_interneurons_controlling_antagonistic_muscles_are_spatially_intermingled; *Ronzano, 2022* .

## Acknowledgements

This work was supported by BBSRC grants BB/L001454 to MB, AJT and DJM and BB/S005943/1 to MB, a Leverhulme Trust grant to MB (grant number RPG-2013–176) and a Wellcome Trust Investigator Award to RMB (110193). JDM was supported by a fellowship from the Jane Coffins Child Memorial Fund for Medical Research and 5K99HD096512 from NICHD. BKB was supported by UCSD Cell and Molecular Genetics Training Program (T32 GM007240), Timken-Sturgis Foundation, Salk Women in Science and Marguerite Vogt Award. RMB is supported by Brain Research UK. SS and NZ were supported by the DFG (ZA 885/1–1 and EXC 257 NeuroCure). SLP is supported as the Benjamin

H Lewis chair in neuroscience and Sol Goldman Charitable Trust, and National Institutes of Health (1 U19 NS112959-01 and 1 R01 NS123160-01). RR is funded by a Wellcome Trust Early Career Award (225674/Z/22/Z). We thank Dr. Ariel Levine for participation in some early experiments and helpful discussions on the data. We are grateful to Drs. Silvia Arber and Marco Tripodi for constructive discussions and helpful comments during the preparation of this manuscript. We thank Roser Montañana-Rosell for critical reading of the manuscript.

## Additional information

### Competing interests

Calvin Chad Smith: is currently an employee of Sania therapeutics. His work is not related to that reported in this manuscript. Andrew J Murray: is a co-founder of Sania Therapeutics. His work is not related to that reported in this manuscript. AM has founders stock in Sania Therapeutics, Ltd. Robert M Brownstone: has a grant from Medical Research Council, is a a co-founder of the parent company, Sania Therapeutics, Ltd. Work not related to that reported in this manuscript. RB is as an Associate Editor for Journal of Neurophysiology, has a patent (18 July, 2020, Priority patent GB 2010981.5; published 20 Jan, 2022, WO/2022/013396 "Gene therapy for neuromuscular and neuromotor disorders") unrelated to this manuscript and is an unpaid Trustee for Stoke Mandeville Spinal Research, International Spinal Research Trust and Scientific Advisory Board, Brain Research UK. RB has founders stock in Sania Therapeutics, Ltd. The other authors declare that no competing interests exist.

### Funding

| Funder | Grant reference number | Author |
|---|---|---|
| Biotechnology and Biological Sciences Research Council | BB/L001454 | Andrew J Todd David J Maxwell Marco Beato |
| Biotechnology and Biological Sciences Research Council | BB/S005943/1 | Marco Beato |
| Leverhulme Trust | RPG-2013-176 | Marco Beato |
| Wellcome Trust | 110193 | Robert M Brownstone |
| Jane Coffin Childs Memorial Fund for Medical Research | | Jeffrey D Moore |
| Eunice Kennedy Shriver National Institute of Child Health and Human Development | 5K99HD096512 | Jeffrey D Moore |
| University of California, San Diego | T32 GM007240 | Bianca K Barriga |
| Timken-Sturgis foundation | | Bianca K Barriga |
| Salk Institute for Biological Studies | | Bianca K Barriga |
| Marguerite Vogt Award | | Bianca K Barriga |
| Brain Research UK | | Robert M Brownstone |
| Deutsche Forschungsgemeinschaft | ZA 885/1-1 | Sophie Skarlatou Niccolò Zampieri |
| Deutsche Forschungsgemeinschaft | EXC 257 NeuroCure | Sophie Skarlatou Niccolò Zampieri |
| Benjamin Lewis Chair in Neuroscience | | Samuel L Pfaff |

| Funder | Grant reference number | Author |
|--------|------------------------|--------|
| Sol Goldman Charitable Trust | | Samuel L Pfaff |
| National Institute of health | 1 U19 NS112959-01 | Samuel L Pfaff |
| National institute of health | 1 R01 NS123160-01 | Samuel L Pfaff |
| Wellcome Trust | 225674/Z/22/Z | Remi Ronzano |

The funders had no role in study design, data collection and interpretation, or the decision to submit the work for publication. For the purpose of Open Access, the authors have applied a CC BY public copyright license to any Author Accepted Manuscript version arising from this submission.

## Author contributions

Remi Ronzano, Conceptualization, Data curation, Formal analysis, Investigation, Visualization, Writing – review and editing; Sophie Skarlatou, Data curation, Formal analysis, Validation, Investigation, Visualization, Writing – review and editing; Bianca K Barriga, Resources, Data curation, Software, Formal analysis, Validation, Investigation, Writing – review and editing; B Anne Bannatyne, Formal analysis, Validation, Investigation, Visualization; Gardave Singh Bhumbra, Software, Validation, Investigation, Visualization, Methodology, Writing – review and editing; Joshua D Foster, Formal analysis, Investigation; Jeffrey D Moore, Data curation, Investigation, Writing – review and editing; Camille Lancelin, Formal analysis, Supervision, Validation, Investigation, Methodology, Writing – review and editing; Amanda M Pocratsky, Data curation, Investigation, Methodology; Mustafa Görkem Özyurt, Investigation; Calvin Chad Smith, Data curation, Investigation, Visualization; Andrew J Todd, Conceptualization, Data curation, Funding acquisition, Validation, Investigation, Project administration; David J Maxwell, Conceptualization, Funding acquisition, Validation, Investigation, Methodology; Andrew J Murray, Conceptualization, Resources, Writing – review and editing; Samuel L Pfaff, Conceptualization, Supervision, Funding acquisition, Validation, Project administration, Writing – review and editing; Robert M Brownstone, Conceptualization, Data curation, Supervision, Funding acquisition, Validation, Writing – original draft, Project administration, Writing – review and editing; Niccolò Zampieri, Conceptualization, Resources, Data curation, Formal analysis, Supervision, Funding acquisition, Validation, Investigation, Visualization, Methodology, Writing – original draft, Project administration, Writing – review and editing; Marco Beato, Conceptualization, Resources, Data curation, Software, Formal analysis, Supervision, Funding acquisition, Validation, Investigation, Visualization, Methodology, Writing – original draft, Project administration, Writing – review and editing

## Author ORCIDs

Remi Ronzano (iD) http://orcid.org/0000-0002-4927-9474
Andrew J Todd (iD) http://orcid.org/0000-0002-3007-6749
Samuel L Pfaff (iD) http://orcid.org/0000-0002-2142-166X
Robert M Brownstone (iD) http://orcid.org/0000-0001-5135-2725
Niccolò Zampieri (iD) http://orcid.org/0000-0002-2228-9453
Marco Beato (iD) http://orcid.org/0000-0002-7283-8318

## Ethics

All experiments were performed in strict adherence to the Animals (Scientific Procedures) Act UK (1986) and certified by the UCL AWERB committee, under project licence number 70/9098. All experiments performed at the MDC were carried out in compliance with the German Animal Welfare Act and approved by the Regional Office for Health and Social Affairs Berlin (LAGeSo). All experiments performed at the Salk Institute were conducted in accordance with IACUC and AAALAC guidelines of the Salk Institute for Biological Studies. All surgeries were performed under general isofluorane anaesthesia. The mice were closely monitored for a 24-hr period following surgery to detect any sign of distress or motor impairment. Every effort was made to minimize suffering.

## Decision letter and Author response

Decision letter https://doi.org/10.7554/eLife.81976.sa1
Author response https://doi.org/10.7554/eLife.81976.sa2

# Additional files

## Supplementary files
- Transparent reporting form
- MDAR checklist

## Data availability

All data generated during this study are included in the manuscript and supporting files. We also provide a link to two GitHub repositories: one includes the whole manuscript in a MATLAB executable format (requires a licence) that allows the reader to interact with the original plots and change the settings of the gaussian kernel used to represent the data. The second is a GitHub repository containing the R version of the manuscript.

The following datasets were generated:

| Author(s) | Year | Dataset title | Dataset URL | Database and Identifier |
|---|---|---|---|---|
| Ronzano R, Skarlatou S, Barriga BK, Bannatyne BA, Bhumbra GS, Foster JD, Moore JD, Lancelin C, Foster JD, Pocratsky A, Pocratsky A, Smith CC, Todd AJ, Maxwell DJ, Murray AJ, Pfaff SL, Brownstone RM, Zampieri N, Beato M | 2022 | Spinal premotor interneurons controlling antagonistic muscles are spatially intermingled MATLAB | https://github.com/marcobeato/Spinal_premotor_interneurons_controlling_antagonistic_muscles_are_spatially_intermingled | GitHub, marcobeato/Spinal_premotor_interneurons_controlling_antagonistic_muscles_are_spatially_intermingled |
| Ronzano R, Skarlatou S, Barriga BK, Bannatyne BA, Bhumbra GS, Foster JD, Moore JD, Lancelin C, Foster JD, Pocratsky A, Smith CC, Todd AJ, Maxwell DJ, Murray A, Pfaff SL, Brownstone RM, Zampieri N, Beato M | 2022 | Spinal premotor interneurons controlling antagonistic muscles are spatially intermingled R | https://github.com/rronzano/Spinal_premotor_interneurons_controlling_antagonistic_muscles_are_spatially_intermingled | GitHub, rronzano/Spinal_premotor_interneurons_controlling_antagonistic_muscles_are_spatially_intermingled |

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
