## [Editor Report]

This is a tour-de-force fundamental study of the spatial organization of flexor and extensor premotor interneurons in the mouse spinal cord by comprehensively employing most of the available premotor circuit tracing strategies involving genetically modified mouse strains and rabies virus. The important results are consistent with rigorous positional reconstructions of the premotor neuron labeling from the multiple circuit mapping approaches employed, convincingly demonstrating overlapping spatial distributions of these premotor neurons, regardless of their putative excitatory and inhibitory neurotransmitter identity. These new results compellingly revise our understanding of the spatial organization of spinal premotor circuits.

---

## [Decision Letter]

**Decision letter after peer review:**

Thank you for submitting your article "Spinal premotor interneurons controlling antagonistic muscles are spatially intermingled" for consideration by *eLife*. Your article has been reviewed by 3 peer reviewers, including Jeffrey C Smith as Reviewing Editor and Reviewer #1, and the evaluation has been overseen by John Huguenard as the Senior Editor.

Essential revisions:

1) The connections and spatial distributions of lumbar spinal premotoneurons were studied here in neonatal mice. This is a current limitation of the technique. Most other labs have also used neonatal animals for their studies. This is potentially a significant limitation and should be carefully discussed since it limits interpretation because there is still significant ongoing synaptogenesis and refinement of connections at this age. We understand that other similar studies have similar limitations but, with the emphasis on technical issues in the paper, this limitation should be discussed.

2) The authors should comment on the large variability between experiments and possible sources. Examination of the number of motoneurons labeled in these experiments provided in Table 1 and Figure 12 supplements show high variability.

3) Figure 3—figure supplement 1 illustrates data from segments lumbar 1 to 6 in which all segments are identical in size and shape due to the normalization procedure used. To avoid confusion, the authors could show the actual sizes before normalization, and then the normalized outlines and distributions. At least make sure the main text explains these are normalized representations (outside the Materials and methods).

4) Currently it is unclear which experiments were analyzed throughout the whole lumbar enlargement and which did not. It would be beneficial to state in the Table the segments analyzed (and the total number of sections) in each experiment. Supplemental Figure 12 depicts only L4 and L5 sections. See Reviewer #2 for related comments.

5) There are questions about the authors' interpretation of ectopic motoneurons outside of expected motor pools. Please see Reviewer #2: Ectopic motoneurons.

6) When using a motoneuron-specific AAV-flex-G in ChAT-Cre mice, there are many more interneurons labeled in these experiments compared to experiments using transgenesis for introducing G in motoneurons. This requires some explanation because the main argument of using transgenic animals to express G is that it would increase and normalize the number of starter motoneurons reducing variability due to the probabilistic nature of co-infections with two viruses.

7) The peaks of the contours in Figure 8C are indiscernible (transverse section only, particularly the ipsilateral side). Since this is a key replication experiment, the results should be clear for comparison. Using darker shades of the colors for contrast may be enough.

8) The leak in Cre expression indicated by Cre/tdTomato labeling in Figure 13 is extensive with 25% of TdTomato labeled neurons being ChAT negative. This is a significant number of interneurons that could initiate second trans-synaptic jumps, which should be pointed out more clearly. If the leaky Cre of the ChAT mouse marks V2 and V3 neurons, this would explain the similar results between the mice. The experiment in the RΦGT mouse without Cre (Figure 13 supplement 1 is necessary because it shows that the RΦGT mouse likely contributes little, if at all, to the issue but this does not get at the potential problem of the leaky Cre. There should be a few lines devoted to this, given the results with the tomato reporter.

9) Extending the spatial analyses to consider the locations of excitatory and inhibitory interneurons is an important component of these studies, although there are questions about the authors' identification/nomenclature of excitatory and inhibitory interneurons. Based on previous anatomical studies suggesting co-expression of GABA and glycine in spinal interneurons, the authors interpret GlyT2-off neurons as excitatory. As argued by Reviewer #2, the statement that all GABAergic spinal neurons should be GlyT2-on may be inaccurate. It is recommended that the authors be more conservative in their interpretation, for instance, by just naming these cells as non-glycinergic, and not making assumptions on these GlyT2-off neurons are excitatory interneurons without direct evidence. Please see the questions by Reviewer #2.

10) The argument (lines 264-266) that interneurons presynaptic to V0c neurons are located in superficial laminae and that most labeling shown in here is more ventral is weak. Reviewer #2, for the reasons outlined in the review, recommends that the authors do not discard possible contributions of disynaptic jumps that might not be revealed by the current analyses. This should be discussed.

11) In the Discussion the authors propose that intracellular delivery of DNA for G and TVA followed by EnvA-coated RV could increase specificity and allow tracing from single functionally defined motoneurons. The transsynaptic jump and expression of these genes takes days and therefore in vivo experiments allowing for enough survival time are necessary. Is motoneuron patch-clamping with in vivo experiments followed by a survival period possible? Are the authors proposing a feasible approach?

12) Although the authors made several attempts to provide explanations for the discrepancies with the previously published mapping results, the question remains – how did two groups who published independent studies using different strategies demonstrate flexor and extensor separation in the dorsal horn, when this study, using several strategies in multiple labs, show that the premotor neurons are in complete overlap? Additional small differences in methodologies could be identified which are not discussed and may provide potential explanations. There are small differences that are not discussed and could provide potential explanations for discrepancies within single techniques, but none will resolve the conflict since they each apply only to a single strategy. For example, the AAV promoters are different (Efa1 here and CMV in the Arber lab papers) and the ratio of Rab to G complement is different (3:1 here and 1:1 in the prior). These could fit well into the section of the Discussion (lines 449-454) because, where they could differentially affect the transmission in the co-injection to muscle experiments, they do not provide answers for discrepancies with other techniques, including the conditional virus experiments.

*Reviewer #1 (Recommendations for the authors):*

All the experimental approaches and data analyses are technically elegant, well designed and executed, and clearly presented with due consideration in the manuscript for any technical limitations. Overall, the experiments are rigorously performed with a design that attempts to reduce biases associated with the various RabV-based circuit tracing methods. The data, although very extensive with numerous data source files and supplemental illustrations (which is justified given the scientific problem and controversy requiring data transparency), are clearly presented. I do not have major concerns.

*Reviewer #2 (Recommendations for the authors):*

I congratulate the authors for a comprehensive set of thorough analyses with a large number of experiments and experimental variables tested to shed some light into the important problem of the organization of premotor interneurons in the spinal cord. As I mentioned in the public review, I believe this is a very useful study. However, there are some important details that need to be clarified such that confusion is not introduced for potential future users of this complex methodology to map premotor interneurons in the spinal cord.

1) Figure 3-figures supplement 1 illustrates data from segments lumbar 1 to 6. All segments are identical in size and shape because the normalization procedure used by the authors and explained in methods. This is misleading to most readers which will be largely of unaware of spinal cord anatomy. I recommend showing the actual realistic sizes, before normalization, and then the normalized outlines and distributions. Also, please make sure the main text explains these are normalized representations (outside the Materials and methods). I believe this normalization was not done in Tripodi et al., who seem to have just added all cells in 2D projections with the original positional values (distance to the central canal) independent of lumbar segments. Tripodi argues that medio-lateral differences are more apparent rostrally, which makes sense because the narrower lateral extension of L1 compared to L4/L5. Figure 2C only shows TA and LG interneurons distributions overlapping in L4 and L5, but in the supplemental figure it looks there are many more TA interneurons in L1 compared to LG. Is it possible to illustrate the distribution of these interneurons in an horizonal projection of all lumbar segments as in 2C? Are TA interneurons indeed more abundant in L1 and because of this they bias more medially when all segments are added together without normalization? Is the normalization obscuring one further explanation for the differences among the studies? On a related noted I was not sure which experiments were analyzed throughout the whole lumbar enlargement and which did not. It will be good to state in the Table the segments analyzed (and total number of sections) in each experiment. Supplemental Figure 12 depicts only L4 and L5 sections.

2) Lines 122-124. I do not understand why the authors argue that lack of identification of G complementation prevents identification of starter neurons. With the transgenic approaches they are using, all motoneurons should express G and precise identification of starter cells thus only depends on how many motoneurons are infected. Of course, this is difficult to estimate because progressive cell death, as indicated in the manuscript. I recommend deleting lines 122 and 123 and just state that it is difficult to identify all starter motoneurons unambiguously because the possibility of cell death.

3) Variability in cell numbers: Close examination of the number of motoneurons labeled in these experiments provided in table 1 and figures 12 supplements show high variability. For example, LG muscle injections in Chat-cre with high titer virus at UCL or UoG vary from 8 motoneurons to 110 (even when taking away differences in counting methods and section sampling), with more motoneurons in experiments labeled as UCL compared to UoG. Similar variability occurs in recovered interneurons (356 to 2574). MDC experiments in Olig2-cre mice seem to be in between. It is remarkable that the dorso-ventral and medio-lateral distribution are the same despite such wide variations in interneurons labeled. This strengthens the conclusions, but I would suggest the authors comment on the large variability between experiments and possible sources. Moreover, from Materials and methods it was unclear whether in injections of UoG were also done that UCL and then processed at Glasgow. It seems this was the case but it could be made clearer, given that in the table they are labeled as a different institution.

4) Ectopic motoneurons. The authors argue that ectopic motoneurons outside the expected motor pools are caused mainly by motoneuron-to-motoneuron connections and transynaptic transfer, this might be for some, however in Figure 2C and Figure 12 supplements some are in the medial motor column. As far as I know there are no reports of recurrent motor axon collaterals between LMC and MMC motoneurons and therefore the authors need to consider alternative explanations. In addition, many of the motoneurons ventral to LG and TA motor pools could well belong to biceps motor pools. By difference to the statement made by the authors, the lower part of the biceps covers anatomically the LG in rodents and it is located in the same limb compartment. In such small animals it might be difficult to avoid virus spread, even with 1 microliter injections. The authors indicate that they confirm muscle injection specificity post-hoc after euthanizing the animals, but this is not shown. Was there any small spread to overlaying biceps? At least the authors should modify their statement to reflect the possibility of some leakage at the injection site to nearby muscles. Moreover, suggesting that all motor pools ventral to LG and TA innervate muscle compartments above the knee is not accurate. Entertaining the possibility of some, quite likely, injection leakage should not affect the main conclusions of the study if estimates suggest they occur in low numbers. They also have a nice control with the injections at different titers. In other words, I do not see the need to try to explain ectopic motoneurons based only on motoneuron-to-motoneuron connections. This is a relatively weak given the data.

5) Definition of excitatory and inhibitory interneurons. To compare the distribution of excitatory and inhibitory interneurons they introduced the GlyT2-GFP allele in Chat-cre animals. Based on anatomical studies from Andrew Todd's group suggesting extensive localization between GABAergic and glycinergic interneurons, they interpret GlyT2-off neurons as excitatory. However, the papers cited (Todd and Sullivan 1990 and Todd et al. 1996) focus on Lamina I to III inhibitory neurons and similar detailed analyses have not yet been performed in lower laminae or in the ventral horn. Moreover, one of these papers was based on the assumption that gephyrin is a marker of glycinergic synapses only, and we now know this is not the case. Therefore, the statement that all GABAergic spinal cord neurons should be GlyT2-on might be inexact. For example, ventrally projecting presynaptic GAD65 interneurons that reside in the medial dorsal horn are not GlyT2 positive and although they are labeled "presynaptic to Ia afferents" because this is one of the major functions, they also synapse on postsynaptic motoneurons (sometimes on the soma) in triadic arrangements that have been well known and reported in the electron microscopy literature for decades. Another possibility are the large GABAergic neurons from LVIII projecting contralaterally and that have been shown previously to reach lamina IX. However, the data presented here shows that contralaterally projecting neurons are almost exclusively GlyT2-off. Is there evidence of contralateral monosynaptic inputs onto motoneurons being predominantly excitatory? Can this population be explained solely by V3s? I suggest to the authors to be more conservative in their interpretation and just name these cells as non-glycinergic, not making assumptions on the excitatory nature of GlyT2-off neurons without direct evidence. For example, why not use in a similar approach the VGLUT2-Chr2-YFP transgene developed by Ole Kiehn?

6) Lines 264-266. The argument that interneurons presynaptic to V0c neurons are located in superficial laminae and that most labeling shown in here is more ventral is weak. Analysis of the Figures in Zampieri et al. suggest that interneurons presynaptic to V0c distribute throughout the spinal cord and that the strong labeling in superficial laminae contains a dense staining that appears mainly axonal and probably derived from the TrkA nociceptive DRG neurons that were also abundantly transynapticaly stained from the V0c population. In addition, Figure 13 shows a more extensive distribution of Chat-cre genetic interneuronal labeling in this animal beyond pitx2 cells. The comparison with Olig2-cre is a much better approach than this explanation which might be based on partially wrong premises: distribution of interneurons presynaptic to V0c are mainly superficial laminae and interneurons targeted in Chat-cre animals are mainly pitx2 interneurons. Therefore, I do not see the need for lines 264-266 that might introduce confusion. It is unfortunate that neither Olig2 or Chat are fully restricted to motoneurons. Nevertheless, together reinforce the conclusion that the distribution of flexor vs extensor coupled interneurons do not differ in location despite the likely presence of some contamination from disynaptic jumps. I see nothing wrong accepting that there is currently no animal model for uniquely targeting motoneurons with a single gene. I recommend the authors do not discard possible contributions of disynaptic jumps because if these occur in interneurons that are so intermingled in the spinal cord, they might not be revealed by the current analyses.

7) Tracing with AAV complementation methods. These experiments confirm the thesis of this manuscript, but I was surprised by the fact that even when using a motoneuron specific AAV-flex-G in Chat-cre mice there are thousands more interneurons labeled in these experiments compared to experiments using transgenesis for introducing G in motoneurons (even taking into account that in these experiments every section was counted while in the others only half or 1/8th of sections were counted). This requires some explanation because the main argument of using transgenic animals to express G is that it would increase and normalize the number of starter motoneurons reducing variability due to the probabilistic nature of co-infections with two viruses. On the contrary, according to the data presented, co-injection of AAV-G and RV was similar or more efficient. Unfortunately, in these experiments the starter motoneurons and specificity of muscle injections were not analyzed and therefore it is difficult to compare whether motoneuron and pool specificity was comparable with the experiments using RITVA mice to introduce G in motoneurons. I agree with the authors in that "The identity and number of starter cells are the main determinants of reproducibility in rabies tracing experiments".

8) In the discussion the authors propose that intracellular delivery of DNA for G and TVA followed by EnvA-coated RV could increase specificity and allow tracing from single functionally defined motoneurons. But I wonder how this experiment will work. The transsynaptic jump and expression of these genes takes days and therefore in vivo experiments allowing for enough survival time are necessary. Is motoneuron patch-clamping with in vivo experiments followed by a survival period possible? Are the authors proposing a feasible approach?

9) The study is based on two muscles. Using LG or G as a model extensor and TA as a flexor. However, in reality LG is bifunctional by inserting in the femur and therefore also contributing flexion at the knee. I understand that current literature using genetic approaches routinely make these useful simplifications, but that does not make them true. It could be helpful to introduce the concept that these studies analyze the distribution of interneurons controlling two specific muscles with different roles at the ankle, and move away from strict extensor/flexors dichotomies that might not be exact.

10) The important issue of how representative are the connections revealed at P1/P2 to the spinal cord in mature animals remains. This is a very significant limitation of the technique and affects all conclusions and should be carefully discussed.

*Reviewer #3 (Recommendations for the authors):*

– There are small differences which are not discussed that could provide potential explanations for discrepancies within single techniques but none will resolve the conflict since they each apply only to a single strategy. For example, the AAV promoters are different (Efa1 here and CMV in the Arber lab papers) and the ratio of Rab to G complement are different (3:1 here and 1:1 in the prior). These could fit well into the section of the Discussion (lines 449-454) because, where they could differentially affect the transmission in the co-injection to muscle experiments, they do not provide answers for discrepancies with other techniques, including the conditional virus experiments. (The only apparent difference that could apply to all is the distortion for normalization but I fail to see how that would differentially affect the flexor vs. extensor data.)

– Restriction of cre-dependent G to ChAT or Olig neurons in the genetic strategy is not directly shown, for good reasons, but the leak in expression of Cre/tdTomato in the indirect experiment (Figure 13) is quite extensive and concerning. 25% of TdTomato neurons are ChAT-. Whether it be due to developmental downregulation of ChAT or ectopic cre expression, this is a significant number of interneurons that could initiate second transsynaptic jumps. The Olig2cre experiments are nice but not really helpful with this. If the leaky cre of the ChAT mouse marks V2 and V3 neurons, this would explain the similar results between the mice. The experiment in the RΦGT mouse without cre (Figure 13 supp 1) is necessary because it shows that the RΦGT mouse likely contributes little, if at all, to the issue but this does not get at the potential problem of the leaky cre. There should be a few lines devoted to this, given the results with the tomato reporter. However, again, this provides a potential confound or reason for a discrepancy in results which only applies to some of the experiments (those using the cre-dependent strategies), and the results and conclusions drawn from these cre-dependent strategies are consistent with the other non-genetic strategies used.

– The peaks of the contours in Figure 8C are indiscernible (tranverse section only, particularly the ipsilateral side). Since this is a key replication experiment, the results should be clear for comparison. Using darker shades of the colors for contrast may be enough.

– Only ankle flexor/extensors tested. The wording throughout the text reflects this, except for the Abstract. The sentence in line 8-10 could include "motor pools controlling flexion and extension of the ankle".

---

## [Author Response]

Essential revisions:1) The connections and spatial distributions of lumbar spinal premotoneurons were studied here in neonatal mice. This is a current limitation of the technique. Most other labs have also used neonatal animals for their studies. This is potentially a significant limitation and should be carefully discussed since it limits interpretation because there is still significant ongoing synaptogenesis and refinement of connections at this age. We understand that other similar studies have similar limitations but, with the emphasis on technical issues in the paper, this limitation should be discussed.

See response to Reviewer #2, comment number 10

2) The authors should comment on the large variability between experiments and possible sources. Examination of the number of motoneurons labeled in these experiments provided in Table 1 and Figure 12 supplements show high variability.

See response to Reviewer #2, comment number 3

3) Figure 3—figure supplement 1 illustrates data from segments lumbar 1 to 6 in which all segments are identical in size and shape due to the normalization procedure used. To avoid confusion, the authors could show the actual sizes before normalization, and then the normalized outlines and distributions. At least make sure the main text explains these are normalized representations (outside the Materials and methods).

See response to Reviewer #2, comment number 1, first part

4) Currently it is unclear which experiments were analyzed throughout the whole lumbar enlargement and which did not. It would be beneficial to state in the Table the segments analyzed (and the total number of sections) in each experiment. Supplemental Figure 12 depicts only L4 and L5 sections. See Reviewer #2 for related comments.

See response to Reviewer #2, comment number 1, second and third part

5) There are questions about the authors' interpretation of ectopic motoneurons outside of expected motor pools. Please see Reviewer #2: Ectopic motoneurons.

See response to Reviewer #2, comment number 4

6) When using a motoneuron-specific AAV-flex-G in ChAT-Cre mice, there are many more interneurons labeled in these experiments compared to experiments using transgenesis for introducing G in motoneurons. This requires some explanation because the main argument of using transgenic animals to express G is that it would increase and normalize the number of starter motoneurons reducing variability due to the probabilistic nature of co-infections with two viruses.

See response to Reviewer #2, comment number 7

7) The peaks of the contours in Figure 8C are indiscernible (transverse section only, particularly the ipsilateral side). Since this is a key replication experiment, the results should be clear for comparison. Using darker shades of the colors for contrast may be enough.

See response to Reviewer #3, comment number 3

8) The leak in Cre expression indicated by Cre/tdTomato labeling in Figure 13 is extensive with 25% of TdTomato labeled neurons being ChAT negative. This is a significant number of interneurons that could initiate second trans-synaptic jumps, which should be pointed out more clearly. If the leaky Cre of the ChAT mouse marks V2 and V3 neurons, this would explain the similar results between the mice. The experiment in the RΦGT mouse without Cre (Figure 13 supplement 1 is necessary because it shows that the RΦGT mouse likely contributes little, if at all, to the issue but this does not get at the potential problem of the leaky Cre. There should be a few lines devoted to this, given the results with the tomato reporter.

See response to Reviewer #3, comment number 2

9) Extending the spatial analyses to consider the locations of excitatory and inhibitory interneurons is an important component of these studies, although there are questions about the authors' identification/nomenclature of excitatory and inhibitory interneurons. Based on previous anatomical studies suggesting co-expression of GABA and glycine in spinal interneurons, the authors interpret GlyT2-off neurons as excitatory. As argued by Reviewer #2, the statement that all GABAergic spinal neurons should be GlyT2-on may be inaccurate. It is recommended that the authors be more conservative in their interpretation, for instance, by just naming these cells as non-glycinergic, and not making assumptions on these GlyT2-off neurons are excitatory interneurons without direct evidence. Please see the questions by Reviewer #2.

See response to Reviewer #2, comment number 5

10) The argument (lines 264-266) that interneurons presynaptic to V0c neurons are located in superficial laminae and that most labeling shown in here is more ventral is weak. Reviewer #2, for the reasons outlined in the review, recommends that the authors do not discard possible contributions of disynaptic jumps that might not be revealed by the current analyses. This should be discussed.

See response to Reviewer #2, comment number 6

11) In the Discussion the authors propose that intracellular delivery of DNA for G and TVA followed by EnvA-coated RV could increase specificity and allow tracing from single functionally defined motoneurons. The transsynaptic jump and expression of these genes takes days and therefore in vivo experiments allowing for enough survival time are necessary. Is motoneuron patch-clamping with in vivo experiments followed by a survival period possible? Are the authors proposing a feasible approach?

See response to Reviewer #2, comment number 8

12) Although the authors made several attempts to provide explanations for the discrepancies with the previously published mapping results, the question remains – how did two groups who published independent studies using different strategies demonstrate flexor and extensor separation in the dorsal horn, when this study, using several strategies in multiple labs, show that the premotor neurons are in complete overlap? Additional small differences in methodologies could be identified which are not discussed and may provide potential explanations. There are small differences that are not discussed and could provide potential explanations for discrepancies within single techniques, but none will resolve the conflict since they each apply only to a single strategy. For example, the AAV promoters are different (Efa1 here and CMV in the Arber lab papers) and the ratio of Rab to G complement is different (3:1 here and 1:1 in the prior). These could fit well into the section of the Discussion (lines 449-454) because, where they could differentially affect the transmission in the co-injection to muscle experiments, they do not provide answers for discrepancies with other techniques, including the conditional virus experiments.

See response to Reviewer #3, comment number 1

Reviewer #1 (Recommendations for the authors):All the experimental approaches and data analyses are technically elegant, well designed and executed, and clearly presented with due consideration in the manuscript for any technical limitations. Overall, the experiments are rigorously performed with a design that attempts to reduce biases associated with the various RabV-based circuit tracing methods. The data, although very extensive with numerous data source files and supplemental illustrations (which is justified given the scientific problem and controversy requiring data transparency), are clearly presented. I do not have major concerns.

We thank Reviewer #1 for their detailed and supportive comments. We are delighted that the reviewer finds that our data are clearly presented, because a lot of effort went into combining them in a coherent and consistent way. We hope we are adding further clarity and transparency, by making all the data available and providing a fully executable version of the manuscript in Matlab and R format.

Reviewer #2 (Recommendations for the authors):I congratulate the authors for a comprehensive set of thorough analyses with a large number of experiments and experimental variables tested to shed some light into the important problem of the organization of premotor interneurons in the spinal cord. As I mentioned in the public review, I believe this is a very useful study. However, there are some important details that need to be clarified such that confusion is not introduced for potential future users of this complex methodology to map premotor interneurons in the spinal cord.1) Figure 3-figures supplement 1 illustrates data from segments lumbar 1 to 6. All segments are identical in size and shape because the normalization procedure used by the authors and explained in methods. This is misleading to most readers which will be largely of unaware of spinal cord anatomy. I recommend showing the actual realistic sizes, before normalization, and then the normalized outlines and distributions. Also, please make sure the main text explains these are normalized representations (outside the Materials and methods). I believe this normalization was not done in Tripodi et al., who seem to have just added all cells in 2D projections with the original positional values (distance to the central canal) independent of lumbar segments.

We have added another supplementary figure (Figure 3—figure supplement 2, described in the text at lines 120-123) showing the pooled data from all the high titre LG and TA injections split across segments before normalization. The size of the idealized spinal cord profiles has been scaled using the six normalization points described in the methods. While the area occupied by premotor interneurons is obviously narrower in the rostral segments, the distributions are still overlapping at all segmental levels, confirming that the normalization procedure does not skew the distributions. In order to keep the text lighter, we have chosen to specify that the data have been scaled to the dimension of individual sections in all the legends of the figures where interneurons distributions are shown (“For each section the data are scaled to the reference points indicated in the methods in order to account for size differences along the segments”), but if preferred by the reviewer, we can transfer this note to the text body. It is difficult to perform a direct comparison of the Tripodi data, because we have not been given access (despite numerous requests) to the raw data and their methods section fails to specify whether their data were normalized (and if yes, how), since they only mention that the position of labelled neurons were reconstructed using the “reference axes” plug-in run in the processing suite “Qu”, none of which have been made available.

Tripodi argues that medio-lateral differences are more apparent rostrally, which makes sense because the narrower lateral extension of L1 compared to L4/L5. Figure 2C only shows TA and LG interneurons distributions overlapping in L4 and L5, but in the supplemental figure it looks there are many more TA interneurons in L1 compared to LG. Is it possible to illustrate the distribution of these interneurons in an horizonal projection of all lumbar segments as in 2C? Are TA interneurons indeed more abundant in L1 and because of this they bias more medially when all segments are added together without normalization? Is the normalization obscuring one further explanation for the differences among the studies?

In the original version of the paper we have chosen to show the full rostrocaudal projections only for individual experiments (as in Figure 2C) in the main text, but we added supplementary figures and data for all the experiments in which rostrocaudal distributions are shown. In the figures describing pooled experiments for *ChAT::Cre;* RϕGT and Olig2::Cre; RϕGT injections we were only showing the transverse projection (this is where the mediolateral segregation, if any, would be more evident) in order to keep a compact figure layout. However, we agree with the referee that this does not give a direct visual representation of the pooled experiments and still leaves the possibility of different relative abundance of LG or TA interneurons in different segments. In order to clarify this, we have added the longitudinal pooled representation of the data as a supplement to the figures where only the transverse sections were shown (namely Figures 3, 4, 7, now in Figure 3—figure supplement 3, Figure 4—figure supplement 1, Figure 7—figure supplement 1). Scrolling through the new figures, it is clear that there is no rostrocaudal bias in the density of premotor interneurons belonging to flexors and extensor muscles. Therefore, the mediolateral segregation cannot have originated from differences in the density of premotor interneurons along the rostrocaudal axis. The new supplementary figures are now called in the text at the appropriate points (lines 123-125 for Figure 3—figure supplement 3, line 186-188 for Figure 4—figure supplement 1 and 308-309 for Figure 7—figure supplement 1).

On a related noted I was not sure which experiments were analyzed throughout the whole lumbar enlargement and which did not. It will be good to state in the Table the segments analyzed (and total number of sections) in each experiment. Supplemental Figure 12 depicts only L4 and L5 sections.

Depending on the exact position of the starting level for sectioning, the representations along the rostrocaudal axis do not encompass exactly the same range across labs (for instance, in MDC experiments the rostrocaudal extension does not reach L1). The whole set of supplementary figures to Figure 12 represent the data in their entirety and therefore already contain the information requested by the reviewer. We would leave it as an editorial choice whether to repeat this information also in the summary table, which is already quite extensivemas this would require some changes in format and probably font size.

2) Lines 122-124. I do not understand why the authors argue that lack of identification of G complementation prevents identification of starter neurons. With the transgenic approaches they are using, all motoneurons should express G and precise identification of starter cells thus only depends on how many motoneurons are infected. Of course, this is difficult to estimate because progressive cell death, as indicated in the manuscript. I recommend deleting lines 122 and 123 and just state that it is difficult to identify all starter motoneurons unambiguously because the possibility of cell death.

This observation is correct. Our statement is valid only when referring to methods requiring the use of a helper virus in order to express G (AAV or AAV-FLEX, Figures 8 and 9), but not when genetic tools are used (Figures 2-7). We have removed the sentence as suggested by the reviewer.

3) Variability in cell numbers: Close examination of the number of motoneurons labeled in these experiments provided in table 1 and figures 12 supplements show high variability. For example, LG muscle injections in Chat-cre with high titer virus at UCL or UoG vary from 8 motoneurons to 110 (even when taking away differences in counting methods and section sampling), with more motoneurons in experiments labeled as UCL compared to UoG. Similar variability occurs in recovered interneurons (356 to 2574). MDC experiments in Olig2-cre mice seem to be in between. It is remarkable that the dorso-ventral and medio-lateral distribution are the same despite such wide variations in interneurons labeled. This strengthens the conclusions, but I would suggest the authors comment on the large variability between experiments and possible sources.

We agree that the efficiency of rabies infection exhibits large variability, both at the level of primary and secondary infected neurons. This is an important point that deserves to be highlighted. We have added an in-depth analysis of this issue and an extra supplementary figure, following the methods described in Tran-Van-Minh et al., BioRxiv 2022 (doi: https://doi.org/10.1101/2022.06.08.494952). The new text is now at lines 204-219, describing Figure 4—figure supplement 3. Briefly, we observe that the number of interneurons vs number of putative starter cells follows a power law, similar to what has been shown in the brain by combining data from many different groups, summarized in the paper cited above. We have no explanation for the large difference in the number of starter neurons, other than subtle differences in the site of injection. However, we observed that reduced titre invariably leads to a reduced number of starter cells (~5 against ~ 35 on average between low and high titre experiments), while the relation between interneurons and primary infected neurons follows a predictable pattern.

Moreover, from Materials and methods it was unclear whether in injections of UoG were also done that UCL and then processed at Glasgow. It seems this was the case but it could be made clearer, given that in the table they are labeled as a different institution.

We have changed the initial sentence in the methods section (now on lines 575-579) to make it clearer where the injections were performed and the tissue analysed. The injections labelled as UoG were performed in the Beato Lab at UCL and the tissue was processed at Glasgow University (Todd and Maxwell labs). The injections labelled as UCL were performed independently and 1-2 years later in the Brownstone lab (again at UCL) where the tissue was processed.

4) Ectopic motoneurons. The authors argue that ectopic motoneurons outside the expected motor pools are caused mainly by motoneuron-to-motoneuron connections and transynaptic transfer, this might be for some, however in Figure 2C and Figure 12 supplements some are in the medial motor column. As far as I know there are no reports of recurrent motor axon collaterals between LMC and MMC motoneurons and therefore the authors need to consider alternative explanations.

We were equally surprised by the observation of some motoneurons labelled in the medial motor column. This has happened in a subset of experiments: 170427n2 (n=3), 1578 (n=2), 170125n3 (n=6), 170508n7 (n=4), 1579 (n=2), 170125n7 (n=5), 170125n8 (n=1), that is 7 injections out of the 53 in which motoneurons were identified, and 23 putative medial motoneurons out of a total of 1174 labelled motoneurons. While these numbers are negligible, we agree with the reviewer that the finding is nonetheless interesting. These findings could best be explained in two different ways:

1) There are genuine synaptic contacts between the medial and lateral motor column. The statement that axons from MMCs invading LMCs have not been described does not prove that they do not exist. A current work in progress in the Beato lab is focussed on motoneuron-motoneuron connections (studied electrophysiologically) and it is revealing somewhat surprising rules of connectivity. While we haven’t (yet) directly looked at MMC-LMC connections, the appearance of a few no longer surprises us. Interestingly, published images from the Arber lab clearly demonstrate the converse: that LMC motoneurons are presynaptic to MMC neurons (Goetz et al., 2015, Figure 1G, 10.1016/j.neuron.2014.11.024). It would be wonderful if we could see their data in their entirety for a more careful analysis.

2) The large cells labelled in the medial motor column are perhaps not motoneurons. In our study, we have defined motoneurons as ChAT positive cells infected by rabies, and have excluded the ones that can be clearly seen as cholinergic interneurons, such as the population of cholinergic partition neurons (V0c) located around the central canal. These neurons are known to be premotor and are labelled by rabies tracing (Stepien et al). Therefore, it is possible that at least some of the cells that were labelled in the medial motor column were cholinergic interneurons pre-synaptic to the infected nucleus. We have inserted this caveat at lines 144-153.

In addition, many of the motoneurons ventral to LG and TA motor pools could well belong to biceps motor pools. By difference to the statement made by the authors, the lower part of the biceps covers anatomically the LG in rodents and it is located in the same limb compartment. In such small animals it might be difficult to avoid virus spread, even with 1 microliter injections. The authors indicate that they confirm muscle injection specificity post-hoc after euthanizing the animals, but this is not shown. Was there any small spread to overlaying biceps? At least the authors should modify their statement to reflect the possibility of some leakage at the injection site to nearby muscles. Moreover, suggesting that all motor pools ventral to LG and TA innervate muscle compartments above the knee is not accurate. Entertaining the possibility of some, quite likely, injection leakage should not affect the main conclusions of the study if estimates suggest they occur in low numbers. They also have a nice control with the injections at different titers. In other words, I do not see the need to try to explain ectopic motoneurons based only on motoneuron-to-motoneuron connections. This is a relatively weak given the data.

We have added a discussion of potential leak in the biceps muscle (as described by Surmeli et al) at lines 156-162 and included this as a possibility. We have however kept the alternative explanation of trans-synaptic jumps between motoneurons because we have abundant functional evidence (paired electrophysiological recordings in the Beato and Brownstone labs) of synaptic connectivity between motoneurons belonging to different nuclei, including those innervating the antagonist muscles studied in the present manuscript (paper in preparation). Overall, we agree with the reviewer that the exact reason behind the presence of ectopic motoneurons is less important than the fact that they are a minority and their presence is unlikely to affect the overall distribution of premotor interneurons.

5) Definition of excitatory and inhibitory interneurons. To compare the distribution of excitatory and inhibitory interneurons they introduced the GlyT2-GFP allele in Chat-cre animals. Based on anatomical studies from Andrew Todd's group suggesting extensive localization between GABAergic and glycinergic interneurons, they interpret GlyT2-off neurons as excitatory. However, the papers cited (Todd and Sullivan 1990 and Todd et al. 1996) focus on Lamina I to III inhibitory neurons and similar detailed analyses have not yet been performed in lower laminae or in the ventral horn. Moreover, one of these papers was based on the assumption that gephyrin is a marker of glycinergic synapses only, and we now know this is not the case. Therefore, the statement that all GABAergic spinal cord neurons should be GlyT2-on might be inexact. For example, ventrally projecting presynaptic GAD65 interneurons that reside in the medial dorsal horn are not GlyT2 positive and although they are labeled "presynaptic to Ia afferents" because this is one of the major functions, they also synapse on postsynaptic motoneurons (sometimes on the soma) in triadic arrangements that have been well known and reported in the electron microscopy literature for decades. Another possibility are the large GABAergic neurons from LVIII projecting contralaterally and that have been shown previously to reach lamina IX. However, the data presented here shows that contralaterally projecting neurons are almost exclusively GlyT2-off. Is there evidence of contralateral monosynaptic inputs onto motoneurons being predominantly excitatory? Can this population be explained solely by V3s? I suggest to the authors to be more conservative in their interpretation and just name these cells as non-glycinergic, not making assumptions on the excitatory nature of GlyT2-off neurons without direct evidence. For example, why not use in a similar approach the VGLUT2-Chr2-YFP transgene developed by Ole Kiehn?

We agree with the caution suggested by the referee. The neurons were correctly labelled as GlyT2_on or _off in the figure, but the wrong reference to excitatory and inhibitory was maintained in the text. We have now removed all instances of excitatory and inhibitory and toned down any conclusion based on the wrong identification of Glyt2_off cells as an exclusive excitatory population (entire paragraph starting at line 245).

6) Lines 264-266. The argument that interneurons presynaptic to V0c neurons are located in superficial laminae and that most labeling shown in here is more ventral is weak. Analysis of the Figures in Zampieri et al. suggest that interneurons presynaptic to V0c distribute throughout the spinal cord and that the strong labeling in superficial laminae contains a dense staining that appears mainly axonal and probably derived from the TrkA nociceptive DRG neurons that were also abundantly transynapticaly stained from the V0c population. In addition, Figure 13 shows a more extensive distribution of Chat-cre genetic interneuronal labeling in this animal beyond pitx2 cells. The comparison with Olig2-cre is a much better approach than this explanation which might be based on partially wrong premises: distribution of interneurons presynaptic to V0c are mainly superficial laminae and interneurons targeted in Chat-cre animals are mainly pitx2 interneurons. Therefore, I do not see the need for lines 264-266 that might introduce confusion. It is unfortunate that neither Olig2 or Chat are fully restricted to motoneurons. Nevertheless, together reinforce the conclusion that the distribution of flexor vs extensor coupled interneurons do not differ in location despite the likely presence of some contamination from disynaptic jumps. I see nothing wrong accepting that there is currently no animal model for uniquely targeting motoneurons with a single gene. I recommend the authors do not discard possible contributions of disynaptic jumps because if these occur in interneurons that are so intermingled in the spinal cord, they might not be revealed by the current analyses.

The reviewer highlights the limitation of the genetic approach for the expression of the rabies glycoprotein, that is, a single specific molecular marker for motor neurons does not exist. We have corrected our initial, wrong (as correctly pointed out by the referee) statement that ‘V0c presynaptic partners … comprise mostly interneurons located in the dorsal laminae of the spinal cord’, replacing ‘mostly’ with ‘many’ (line 295). The argument that double jumps from V0c interneurons are unlikely still holds, because if that was the case, we should see at least some of those ‘many’ interneurons – or axons, as the reviewer points out – pre-synaptic to V0c.

We still think that double jumps are a rare event and, as stressed by the reviewer, this is proved by the near perfect overlap of distributions from ChAT and Olig2 mice, in which the contribution of double jumps, if any, would arise from entirely different populations of premotor interneurons, making our observations unlikely in the extreme. A substantial contribution of double jumps is further ruled out by the experiments with AAV trans-complementation of the glycoprotein, where the occurrence of double jumps is impossible. Following the reviewer’s recommendation, we have toned down the parts of the text where we discuss the possibility of double jumps, but still cite our arguments against them (see lines 445-447).

7) Tracing with AAV complementation methods. These experiments confirm the thesis of this manuscript, but I was surprised by the fact that even when using a motoneuron specific AAV-flex-G in Chat-cre mice there are thousands more interneurons labeled in these experiments compared to experiments using transgenesis for introducing G in motoneurons (even taking into account that in these experiments every section was counted while in the others only half or 1/8th of sections were counted). This requires some explanation because the main argument of using transgenic animals to express G is that it would increase and normalize the number of starter motoneurons reducing variability due to the probabilistic nature of co-infections with two viruses. On the contrary, according to the data presented, co-injection of AAV-G and RV was similar or more efficient. Unfortunately, in these experiments the starter motoneurons and specificity of muscle injections were not analyzed and therefore it is difficult to compare whether motoneuron and pool specificity was comparable with the experiments using RITVA mice to introduce G in motoneurons. I agree with the authors in that "The identity and number of starter cells are the main determinants of reproducibility in rabies tracing experiments".

We agree with the reviewer that the number of labelled interneurons in some of the AAV experiments is exceptionally high, but once the section sampling is taken into account, it is still in the range (5000-10000) of the most efficient among the experiments using the generic trans-complementation strategy (see new Figure 4—figure supplement 2). Knowledge of the number of motoneurons would have allowed to test whether the ratio of secondary to primary infections obeys the power law described in Figure 4—figure supplement 2 also in this set of experiments. It is known that one important determinant of efficiency is the quality of the AAV-G, and it is possible that in these sets of experiment the expression of G was particularly efficient. We do not have any direct means of comparison from muscle injections, because none of the published papers using this technique (Stepien et al., Tripodi et al. and Takeoka and Arber) report the raw number of infected cells, but judging by their figures it is possible that some of their experiments labelled neurons in the thousands, so not far from what we are typically seeing in our dataset. It is also worth noting that the AAV-FLEX injections were done with the enhanced glycoprotein (oG) that in the Callaway lab has been shown to increase the efficiency of trans-synaptic jumps (DOI: 10.1016/j.celrep.2016.03.067). Furthermore, in the set of experiments using AAV-G, we think that the distribution observed resulted from both retrograde and anterograde tracing. This seems to have dramatically increased the number of spinal INs labelled with substantial labelling of the dorsal horn originating from transsynaptic anterograde transfer from sensory afferents in addition to the retrograde infection of premotor INs.

8) In the discussion the authors propose that intracellular delivery of DNA for G and TVA followed by EnvA-coated RV could increase specificity and allow tracing from single functionally defined motoneurons. But I wonder how this experiment will work. The transsynaptic jump and expression of these genes takes days and therefore in vivo experiments allowing for enough survival time are necessary. Is motoneuron patch-clamping with in vivo experiments followed by a survival period possible? Are the authors proposing a feasible approach?

We added a note in the discussion (lines 526-534) to specify that with the currently available techniques, this is only a gedanken experiment, one that is theoretically possible, but has never been attempted. It is mentioned here because it has been successfully performed in the brain (the cited Marshel et al., 2010; Rancz et al., 2011), but the main limitation is that to our knowledge nobody has ever patched motoneurons in vivo, and sharp recordings (that are instead relatively straightforward) would not be ideal for the transfer of viral material and in all likelihood would not result in the survival of the recorded motoneurons.

9) The study is based on two muscles. Using LG or G as a model extensor and TA as a flexor. However, in reality LG is bifunctional by inserting in the femur and therefore also contributing flexion at the knee. I understand that current literature using genetic approaches routinely make these useful simplifications, but that does not make them true. It could be helpful to introduce the concept that these studies analyze the distribution of interneurons controlling two specific muscles with different roles at the ankle, and move away from strict extensor/flexors dichotomies that might not be exact.

We agree that the generic term ‘extensor’ is inappropriate for GS, that also contributes to knee flexion. We have now added the term ‘ankle extensor’ to the methods. We find that replacing all occurrences of ‘extensor’ with ‘ankle extensor’ would damage the flow of the text, but if required, we can do it on a second revision.

10) The important issue of how representative are the connections revealed at P1/P2 to the spinal cord in mature animals remains. This is a very significant limitation of the technique and affects all conclusions and should be carefully discussed.

See point 1: We agree that our map of premotor interneurons is a snapshot of the state of connectivity during the first 4-5 days following injection, when most of the jumps presumably occur and this does not necessarily reflect the actual connectivity in any given adult animal. This neonatal state in fact reflects the starting point used by spinal circuits, on which there are undoubtedly changes as mice develop, adapt to their environments, and learn new motor behaviours. As such, we have no evidence to support the idea that similar distributions of premotor interneurons would be observed at a more mature stage. We have now mentioned explicitly this caveat at the end of discussion (lines 553-561). Nonetheless, when it comes to methods comparisons and contrast with previously published results, the issue of age is not relevant, since all the rabies experiments are performed at similar ages in this study and the original study we are comparing our data to.

Reviewer #3 (Recommendations for the authors):– There are small differences which are not discussed that could provide potential explanations for discrepancies within single techniques but none will resolve the conflict since they each apply only to a single strategy. For example, the AAV promoters are different (Efa1 here and CMV in the Arber lab papers) and the ratio of Rab to G complement are different (3:1 here and 1:1 in the prior). These could fit well into the section of the Discussion (lines 449-454) because, where they could differentially affect the transmission in the co-injection to muscle experiments, they do not provide answers for discrepancies with other techniques, including the conditional virus experiments. (The only apparent difference that could apply to all is the distortion for normalization but I fail to see how that would differentially affect the flexor vs. extensor data.)

1) We thank the reviewer for noticing two further differences in the experiments. This has now been highlighted in the discussion (lines 482-489) in the suggested paragraph, but we agree that multiple lines of evidence indicate that this is not sufficient to explain the results. The normalization issue has been raised also by reviewer #2. In answer to their comment we have added an extra figure sowing the data before normalization (see comment number 1 to reviewer #2 above). We also wish to add that it is difficult to envisage a normalization issue that differentially affects injections from one muscle and not the other. Furthermore, many of our experiments were performed with double injections in the same animals (all individual experiments are reported in the supplementary figures) and in these cases the lack of segregation could clearly not be due to normalization issues.

– Restriction of cre-dependent G to ChAT or Olig neurons in the genetic strategy is not directly shown, for good reasons, but the leak in expression of Cre/tdTomato in the indirect experiment (Figure 13) is quite extensive and concerning. 25% of TdTomato neurons are ChAT-. Whether it be due to developmental downregulation of ChAT or ectopic cre expression, this is a significant number of interneurons that could initiate second transsynaptic jumps. The Olig2cre experiments are nice but not really helpful with this. If the leaky cre of the ChAT mouse marks V2 and V3 neurons, this would explain the similar results between the mice. The experiment in the RΦGT mouse without cre (Figure 13 supp 1) is necessary because it shows that the RΦGT mouse likely contributes little, if at all, to the issue but this does not get at the potential problem of the leaky cre. There should be a few lines devoted to this, given the results with the tomato reporter. However, again, this provides a potential confound or reason for a discrepancy in results which only applies to some of the experiments (those using the cre-dependent strategies), and the results and conclusions drawn from these cre-dependent strategies are consistent with the other non-genetic strategies used.

2) We agree that 25% of TdTomato neurons that are ChAT negative is concerning. However, the immunoreaction for ChAT is notoriously very delicate and heterogeneous between the very high expression in motor neurons and lower expression in other spinal IN populations that are known to express ChAT. This is notably shown by the high number of TdTomato^ON^ ChAT^OFF^ cells around the central canal in Fig13 that certainly depict either autonomic NOS+ motoneurons or V0c commissural INs that are known to express ChAT. Thus, the 25% should be considered as an upper limit, since some of the ChAT^OFF^ neurons were likely just below the threshold for detection. Furthermore, a large majority of the ChAT ^OFF^ and TdTomato ^ON^ neurons are located in the superficial laminae, where they could not contribute to di-synaptic jumps, since this area is devoid of projections from intermediate spinal interneurons. Hence the potential contribution of leaky Cre is limited, but we have now further acknowledged this issue when we describe the results of Figure 13 (lines 607-609). The observation of Cre leak is also what prompted us to use also alternative strategies (the two different AAVs and the PRV). The overlap of the distributions regardless of the methods used is evidence (albeit indirect) that while all methods have their associated error, the magnitude of such errors is not sufficient to significantly alter any of the observed distributions.

– The peaks of the contours in Figure 8C are indiscernible (tranverse section only, particularly the ipsilateral side). Since this is a key replication experiment, the results should be clear for comparison. Using darker shades of the colors for contrast may be enough.

3) We have re-plotted this dataset using different settings for the contours in order to make the representation clearer. The paper is now also available in executable form, so that any reader can alter the settings of the contour plots.

– Only ankle flexor/extensors tested. The wording throughout the text reflects this, except for the Abstract. The sentence in line 8-10 could include "motor pools controlling flexion and extension of the ankle".

This has been now corrected.